# Cultured meat with enriched organoleptic properties by regulating cell differentiation

Milae Lee [1], Sohyeon Park[1], Bumgyu Choi[1], Woojin Choi [1], Hyun Lee[2], Jeong Min Lee [3], Seung Tae Lee[2,3], Ki Hyun Yoo [4], Dongoh Han [4], Geul Bang [5], Heeyoun Hwang [5,6], Won-Gun Koh[1], Sangmin Lee [7] ✉ & Jinkee Hong [1] ✉

Research on cultured meat has primarily focused on the mass proliferation or differentiation of muscle cells; thus, the food characteristics of cultured meat remain relatively underexplored. As the quality of meat is determined by its organoleptic properties, cultured meat with similar sensory characteristics to animal-derived meat is highly desirable. In this study, we control the organoleptic and nutritional properties of cultured meat by tailoring the 2D differentiation of primary bovine myoblasts and primary bovine adipose-derived mesenchymal stem cells on gelatin/alginate scaffolds with varying stiffness. We assess the effect of muscle and adipose differentiation quality on the sensory properties of cultured meat. Thereafter, we fabricate cultured meat with similar sensory profiles to that of conventional beef by assembling the muscle and adipose constructs composed of highly differentiated cells. We introduce a strategy to produce cultured meat with enriched food characteristics by regulating cell differentiation with scaffold engineering.

Cultured meat is gaining attention as a sustainable and eco-friendly meat for the future;[1,2] accordingly, its demand has increased drastically.[3,4] However, research on cultured meat is still in its infancy because of the challenges on mimicking the biological and physical properties of slaughtered meat in vitro. Furthermore, cultured meat research on mimicking the food-related characteristics such as sensorial properties and nutritional value of meat is even fewer. To overcome these issues, cells can be supported with scaffolds, which provide them with biological or physical environment of natural tissues of animals.[5] Recently, researchers have reported strategies to produce cultured meat using various types of scaffolds, such as 3D printed bioink,[6] textured soy proteins,[7] and spinach,[8] to mimic the structure of meat. However, research on cultured meat scaffolds has focused on the evaluation of the occurrence of cell differentiation and the design of the scaffold shape to resemble meat. Consequently, factors that determine the various food characteristics of meat are generally overlooked. For example, the biological characteristics of muscle and fat tissues, such as myofiber dimensions and lipid content, determine the organoleptic properties of slaughtered meat,[9,10] and these tissue characteristics can be affected by the differentiation of cells (Fig. 1a). Therefore, it is crucial to develop cultured meat that mimics biological characteristics of livestock meat, which is composed of highly differentiated muscle and fat tissues, to achieve the natural sensorial characteristics of meat.

The mechanical properties of a scaffold can regulate cellular functions,[11] and myogenesis and adipogenesis rates can be regulated

[1]Department of Chemical & Biomolecular Engineering, College of Engineering, Yonsei University, 50 Yonsei-ro, Seodaemun-gu, Seoul 03722, Republic of Korea. [2]Department of Animal Life Science, Kangwon National University, 1 Kangwondaehak-gil, Chuncheon-si, Gangwon-do 24341, Republic of Korea. [3]Department of Applied Animal Life Science, Kangwon National University, 1 Kangwondaehak-gil, Chuncheon-si, Gangwon-do 24341, Republic of Korea. [4]Simple Planet, 805, 34, sangwan 12-gil, Seongdong-gu, Seoul 04790, Republic of Korea. [5]Research Center for Bioconvergence Analysis, Korea Basic Science Institute, Cheongju 28119, Republic of Korea. [6]Critical Diseases Diagnostics Convergence Research Center, Korea Research Institute of Bioscience and Biotechnology, Daejeon 34141, Republic of Korea. [7]School of Mechanical Engineering, Chung-ang University, 84, Heukseok-ro, Dongjak-gu, Seoul 06974, Republic of Korea. ✉e-mail: slee98@cau.ac.kr; jinkee.hong@yonsei.ac.kr

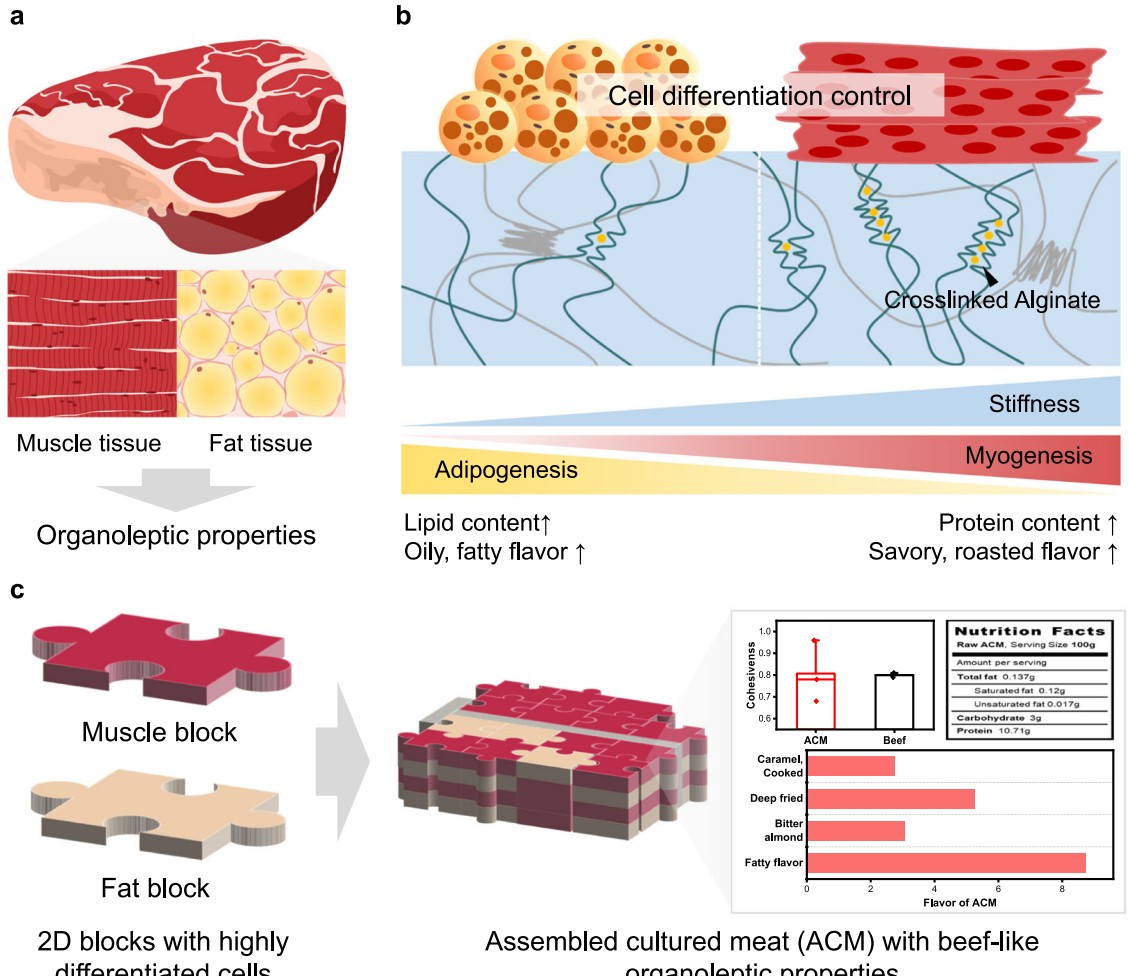

**Fig. 1 | Schematic illustration of the proposed strategy. a** Muscle and fat tissues consisting slaughtered beef determine the organoleptic properties of beef. **b** Scaffold engineering for controlling myogenesis and adipogenesis affects the nutritional and sensorial properties of cultured meat. **c** Assembled cultured meat composed of the 2-dimensional meat blocks with highly differentiated cells owns beef-like organoleptic properties.

by the stiffness of the scaffold. Particularly, Young's moduli of ~11[12] and ~3 kPa[13] are required to stimulate myogenesis and adipogenesis, respectively. These mechanical properties can be achieved by controlling the crosslinking degree of polymer networks in the scaffold. For example, the stiffness of an alginate hydrogel depends on the crosslinking density of alginate, which affects the adipogenesis of mesenchymal stem cells.[14] The biochemical and physical properties of the muscle fibers and lipid contents of the adipose tissue are determined by the degree of each cellular differentiation. To mimic the structure of meat, both adipogenic and myogenic differentiation should occur significantly in one mass of the cultured meat. This simultaneous differentiation can be achieved by controlling the stiffness of scaffolds by varying the crosslinking degree of polymers.

Gelatin, a helical protein derived from collagen, is now widely used in cultured meat research as well as in tissue engineering field owing to its cell-adhesive RGD sequence.[15,16] Further, it is a widely known food additive owing to its easy dissolution at elevated temperatures and solidification with an elastic texture at room temperature.[17] However, its low melting point makes it less stable under cell culture conditions. To address this poor stability, cytotoxic chemical crosslinkers, such as glutaraldehyde and genipin have been employed.[18] To prevent cytotoxicity, the use of microbial transglutaminase (mTG), which forms peptide bonds between the lysine and glutamine residues of gelatin chains, has been demonstrated.[19,20] This non-toxic crosslinker is widely used in the food industry, particularly

for meat bonding, indicating that mTG is an attractive crosslinker for achieving a stable gelatin-based scaffold. Alginate, a brown algae-derived polysaccharide, lacks cell adhesive capacity, but can strengthen scaffolds to support cells when crosslinked with cations. Particularly, the mechanical properties of alginate hydrogel can be significantly varied by varying the concentration of the crosslinked alginate.[21] For this reason, alginate has also been widely used for scaffold component to regulate the physical properties of scaffold.

In this study, we suggest a strategy of developing cultured meat that embodies the organoleptic properties of conventional beef by only regulating cellular differentiation. To investigate the effect of cellular differentiation on the sensorial characteristics of cultured meat, we firstly fabricate 2D hydrogel scaffolds composed of fish gelatin and alginate, and the stiffness of the hydrogels is controlled by varying the content of the crosslinked alginate. Subsequently, we regulate the differentiation of primary bovine myoblasts and primary bovine adipose-derived mesenchymal stem cells (adMSCs). We verify the variation in the sensory properties with a change in the cellular differentiation degree (Fig. 1b). Using this approach, we develop 2D muscle blocks and fat blocks. Lastly, the two meat block types are assembled to produce a cultured meat with beef-like organoleptic characteristics (Fig. 1c). In many cultured meat studies, efforts have been made to fabricate large construct of cultured meat through 3D printing technique or assembly methods with crosslinking agents to mimick large dimensions and appearance of meat.[22–24] Partially

inspired by these previous works, we developed the assembled cultured meat structure similar to slaughtered meat, in terms of organoleptic characteristics after cell differentiation. Furthermore, the food-related characteristics such as sensorial properties and nutritional values of the final assembled cultured meat were thoroughly assessed using the evaluation method commonly used in food and meat studies.[25–29] We expect that this strategy can be used to produce cultured meat that embodies the organoleptic properties of various conventional meat parts.

## Results

### Fabrication of the gelatin/alginate hydrogels with different mechanical properties

The hydrogel scaffold was composed of two biopolymers: fish gelatin and alginate. To control the scaffold stiffness, the concentration and crosslinking degree of alginate were adjusted. The concentration of alginate in the hydrogel was classified into two groups (LA and HA; Fig. 2a). Both the hydrogels contained the same amount of crosslinked gelatin, but with low (0.25% (w/v)) and high (2% (w/v)) alginate concentrations, respectively. Thereafter, the alginate network of each group was electrostatically crosslinked with calcium ions, and the crosslinking degree of the alginate was varied. For the highly crosslinked (HC) alginate, the alginate:calcium weight ratio was 1:1, whereas the ratio for the hydrogel with a low alginate crosslinking (LC) degree was 1:0.5. The hydrogels were named according to the alginate concentration and crosslinking degree (Supplementary Table 1). The microstructure of the polymers can be investigated using Raman spectroscopy.[30] In Fig. 2b, the calcium concentration-dependent changes in the alginate crosslinking degree were verified using Raman spectroscopy. The interaction of the carboxylates of alginate with calcium cations resulted in structural changes in the alginate network, as evident by the Raman shift in the range from 1400 to 1430 cm$^{-1}$.[31] Generally, the exchange of the Na$^+$ cations of sodium alginates

with Ca$^{2+}$ results in the shift of the COO$^-$ peaks at 1410–1420 cm$^{-1}$ to higher wavelengths, as well as the broadening of these peaks.[32] Similarly, this peak shift was confirmed in the Raman spectra of the hydrogels prepared in this study. As the crosslinking degree increased, the intensity of the Raman peaks at ~1400 cm$^{-1}$ decreased and became broader, whereas the intensity of those at ~1420 cm$^{-1}$ increased, and the peaks became sharp (Fig. 2b(i) and Supplementary Fig. 1). The peaks at ~1400 and 1420 cm$^{-1}$ can be assigned to Na–alginate and Ca–Alginate, respectively. Figure 2b(ii) shows that the percentage of Ca–alginate in the LAHC is approximately 1.5 times higher than that in the LALC, and this difference is similar to the fold difference in the percentage of the HALC and HAHC. These results confirmed that the crosslinking degree of an alginate network can be controlled by changing the calcium ion concentration of the respective alginate group (LA and HA).

The stiffness of the hydrogels was measured using compression test (Fig. 2c). A linear stress–strain curve is observed until the compressive strain reached 50% (Fig. 2c(i)). The compressive moduli of the hydrogels were compared at a compressive strain of 0–50% (Fig. 2c(ii)). The stiffness of the hydrogel increased with an increase in the concentration and crosslinking degree of alginate. Among the hydrogel groups, LALC and HALC exhibited similar Young's modulus of the adipose tissue (3–4.5 kPa)[33] and skeletal muscle tissue (10–12 kPa),[34,35] respectively. Based on these results, the LALC and HALC, which exhibited stiffnesses of 3 and 11 kPa, respectively, were selected for the adipogenic and myogenic differentiations. The dimensions of the two hydrogels were measured, and the LALC and HALC were confirmed to exhibit the same thickness (Supplementary Table 2). Next, the weight loss of the LALC and HALC was measured by immersing the respective scaffold in the bovine myoblast-conditioned medium for 12 days at 37 °C (Supplementary Fig. 2). The degradation rate in the LALC was higher but was not significantly different from that of the HALC. As a result, we confirmed that both the scaffolds did not

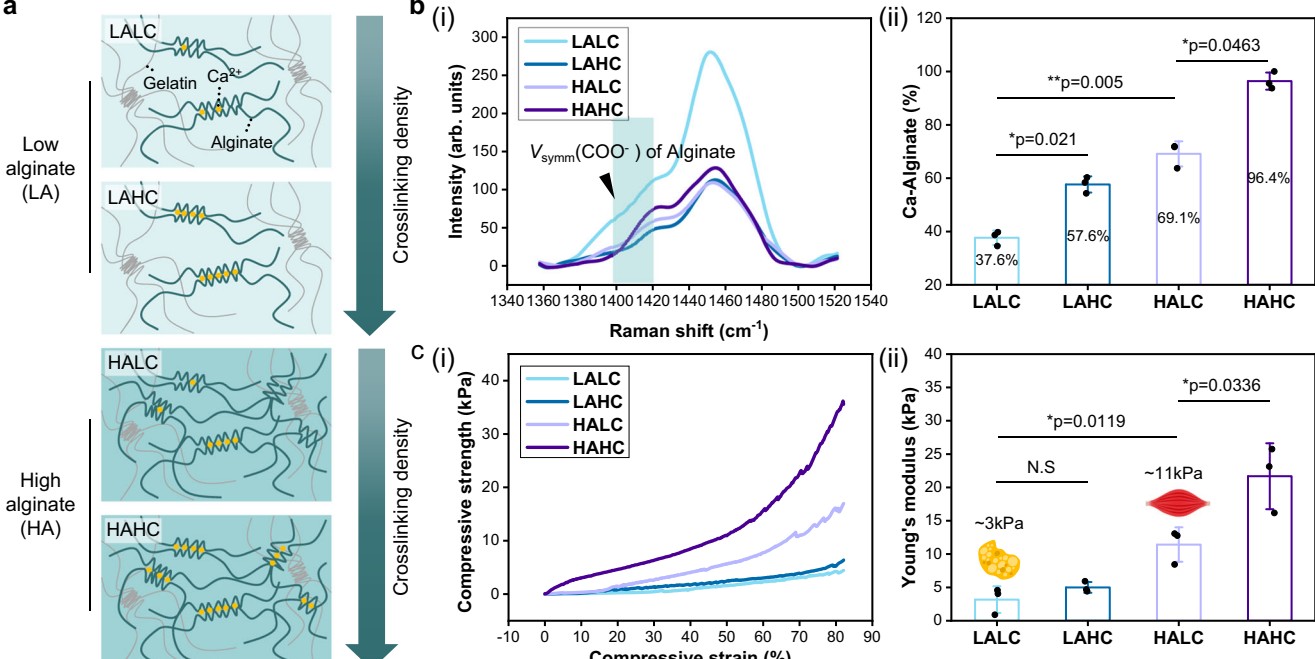

**Fig. 2 | Characterization of the gelatin/alginate scaffold. a** Illustration of the change in the alginate concentration and crosslinking degree of the alginate network of the scaffolds. **b** (i) Raman spectra of the low alginate ratio with low crosslinking hydrogel (LALC, light blue), low alginate ratio with high crosslinking hydrogel (LAHC, dark blue), high alginate ratio with low crosslinking hydrogel (HALC, light purple), and high alginate ratio with high crosslinking hydrogel (HAHC, dark purple), (ii) Percentage of the calcium-crosslinked alginate structures calculated from the Raman shifted peak area between 1390 and 1420 cm$^{-1}$ (mean ± SD, $n = 3$ independent experiments, two-tailed $t$-test). **c** (i) Stress–strain curves obtained during the compression test of the scaffolds, (ii) Young's modulus of each scaffold at a compressive strain of 0–50% (mean ± SD, $n = 3$ independent experiments, two-tailed $t$-test). Source data are provided as a Source Data file.

fully degrade under the cell culture environment, implying that the LALC and HALC can stably support the cells during proliferation and differentiation periods.

## Optimization of the muscle block: Enriching the texture and flavors through myogenesis

The difference in the myogenic differentiation of bovine myoblasts in the HALC and LALC was investigated. Before the differentiation, CCK-8 assay was performed to confirm the cell proliferation in the scaffolds (Supplementary Fig. 3). There was no significant difference in cell viability and proliferation behavior of the HALC and LALC scaffolds throughout the proliferation period. As the amounts of the cells proliferated on the two scaffolds were confirmed to be similar, factors other than differences in cell differentiation can be excluded. Additionally, there was no significant difference in the cell viability of the HALC and LALC compared to that of the culture well group, confirming the non-cytotoxicity of the HALC and LALC scaffolds.

After 4 days of proliferation, the medium was changed to initiate myogenic differentiation. The expression of the myosin heavy chains

(MHCs) was investigated by obtaining the confocal images as shown in Fig. 3a. On the 5 day of differentiation, nuclei fusion occurred only in the HALC. On day 8, the myogenic differentiation was accelerated in the HALC, and branched myotubes were observed, whereas only few cells were stained red in the LALC. Proteomic analysis was performed on the last day of myogenic differentiation using a liquid chromatography–tandem mass chromatography (LC-MS/MS) system to verify the effect of the scaffold type on the cellular behavior (Fig. 3b). The up-regulated and down-regulated proteins in the HALC were compared to those in the LALC (Fig. 3b(i) and Supplementary Fig. 4). Among the up-regulated proteins in the HALC, the proteins which were expressed by more than 1.5 times higher in HALC were listed and specifically analyzed by performing gene annotation (Fig. 3b(ii) and Supplementary Table 3). The results confirmed that the proteins significantly up-regulated in the HALC were related to the muscle and actin-binding proteins. Furthermore, the proteins composing cytoplasm and cytoskeleton were up-regulated in the HALC, indicating the cell elongation according to the myogenic differentiation. The proteomic analysis of the two groups before the induction of

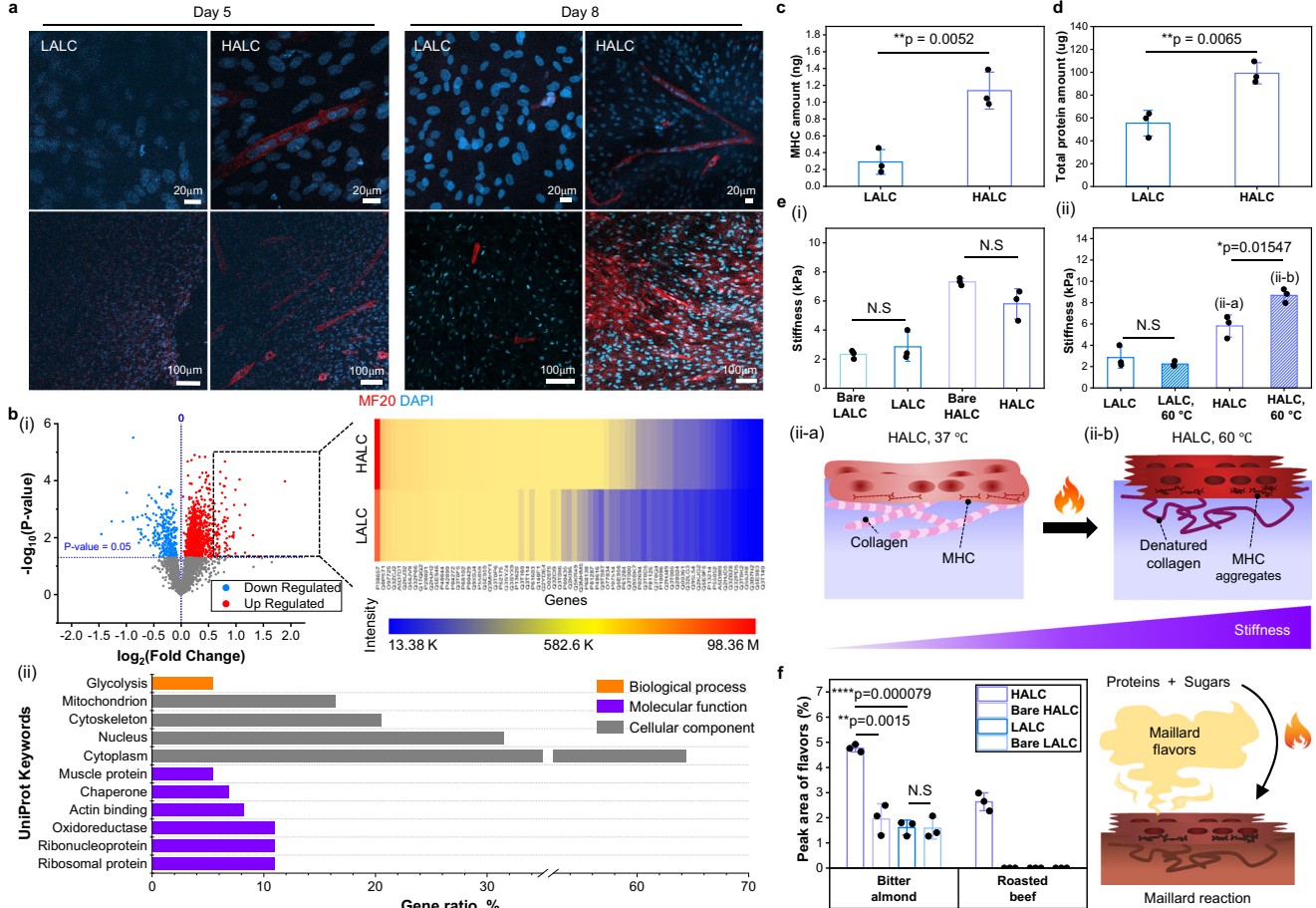

**Fig. 3 | Optimization of the muscle block. a** Myosin heavy chains (MHC) and nuclei immunostained with MF20 (red) and DAPI (blue) on differentiation day 5 and day 8. LALC represents the low alginate ratio with low crosslinking hydrogel cultured with myoblasts whereas HALC is the high alginate ratio with low crosslinking hydrogel cultured with myoblasts. Scale bars: 20 μm (first row) and 100 μm (second row). **b** (i) Volcano plot showing the up-regulated (red dots) and down-regulated (blue dots) proteins in the HALC ($n = 3$ independent experiments, two-tailed $t$-test). Fold change indicates the intensity ratio of the protein expression in the HALC to that in the LALC. Proteins expressed by more than 1.5 folds in the HALC were sorted in the heatmap with the intensity of the protein expression. (ii) Gene annotation of the proteins in the heatmap. **c** Quantification of the MHC of differentiated bovine myoblasts in LALC (blue graph) and HALC (purple graph)

(mean ± SD, $n = 3$ independent experiments, two-tailed $t$-test). **d** Bicinchoninic acid (BCA) assay results showing the total protein amount of the cells per scaffold (mean ± SD, $n = 3$ independent experiments, two-tailed $t$-test). **e** Stiffness of the raw (ungrilled) groups (i) and grilled groups (ii) measured using a rheometer (mean ± SD, $n = 3$ independent experiments, two-tailed $t$-test). Grilled groups are represented as hatched graphs. Illustration of the stiffness change in the HALC with cells owing to an increase in the temperature from 37 °C (ii-a) to 60 °C (ii-b). N.S means non-significant. **f** Peak area ratio of the flavors detected from each grilled sample (left) (mean ± SD, $n = 3$ independent experiments, two-tailed $t$-test). Illustration of the flavor enrichment in the cooked HALC group (right). Source data are provided as a Source Data file.

differentiation was also performed on the last day of proliferation (Supplementary Fig. 5). In the undifferentiated groups, the proteins up-regulated in the HALC were significantly lower than those in the differentiated groups, as shown in the volcano plots. In addition, the expression of myogenesis-related proteins was higher in the HALC than in the LALC with undifferentiated cells, suggesting that the cells were affected by the scaffold stiffness even before differentiation was induced.

For the quantitative comparison of the MHC of the groups, MHC enzyme-linked immunosorbent assay (ELISA) was performed (Fig. 3c). The results revealed the expression of MHC by the cells in the LALC and HALC, implying that myotube formation occurred in both groups. However, the degree of myogenic differentiation was higher in the HALC because of the similarity in its stiffness to that of a natural muscle tissue. The total content of the proteins extracted from the differentiated cells in each group is compared in Fig. 3d. Results revealed that the amount of proteins in the HALC was significantly higher than that in the LALC owing to the higher myogenesis. Both MHC ELISA and total protein analysis were conducted for the undifferentiated groups, and there was no significant difference in the results of the LALC and HALC (Supplementary Fig. 6 and Supplementary Fig. 7). This confirmed that the difference in MHC and total protein amount of the LALC and HALC can be attributed to the different degree of myogenic differentiation owing to the different mechanical properties of the scaffolds. The proteomic analysis demonstrated that the expression of the genes related to myogenesis was also affected by scaffold stiffness before the induction of differentiation. However, as there was no difference in the MHC and total protein amount of the undifferentiated groups with a change in the scaffold stiffness, it indicated that the effect of the scaffold on the cells increased significantly after induction of differentiation.

After confirming the change in the myogenesis with a change in the stiffness of the scaffold, the texture and flavor of the two groups were analyzed to evaluate the influence of the different myogenic differentiation degrees on the food characteristics. Figure 3e(i) shows the comparison of the stiffness of the bare scaffolds (bare LALC and bare HALC) and the scaffolds with differentiated cells (LALC and HALC). The stiffness change according to the cell culture was not significant. However, the stiffness of the HALC group increased significantly after heating at 60 °C owing to the abundance of muscle proteins (Fig. 3e(ii)). Muscle proteins, such as MHCs, which are the main components of muscle tissue, are expressed during myogenic differentiation. These proteins significantly affect the meat texture as they denature and stiffen the meat muscle at high temperatures above 50 °C.[36–39] To clearly confirm the stiffness change upon heating due to the myogenic differentiation, the same analysis was conducted for the bare scaffolds and undifferentiated groups. There was no significant change in the stiffness of the bare scaffold groups after heating (Supplementary Fig. 8). In contrast, the stiffness of the HALC of the undifferentiated groups decreased upon heating due to the factors such as thermal denaturation of the ECM proteins, such as collagen[40] (Supplementary Fig. 9). The triple helices of the collagens can unfold as the temperature increased to 60 °C, thus decreasing the Young's modulus of the collagen fibrils.[41] As demonstrated in Fig. 3e(ii-a) and Fig. 3e(ii-b), the denaturation of collagen and muscle proteins occurs with an increase in temperature. In the HALC with differentiated cells, the effect of MHC aggregation was dominant over that of collagen destruction owing to the high amount of expressed MHC, thus increasing the stiffness.

The flavors of the grilled scaffolds were analyzed using gas chromatography (GC), and the corresponding results are shown in Fig. 3f and Supplementary Table 4. As described in the illustration in Fig. 3f, Maillard reaction occurs when proteins and sugars react under high temperatures, and this is accompanied with the formation of various flavored volatile compounds. Owing to the difference in the alginate

contents in LALC and HALC, GC analysis was also performed for the bare scaffolds to identify the effect of cell myogenesis on flavor. All the samples were prepared by grilling at 180 °C for 5 min, and the Maillard compounds with savory and meat flavors were detected. Benzaldehyde is known as a Maillard reaction product with a bitter almond flavor, and it is found in cooked beef.[42] This compound was detected in all the samples. Because gelatin and alginate can participate in Maillard reactions, Maillard compounds can also be detected in the bare scaffolds. However, the peak area of benzaldehyde was approximately 2.2 and 3 times higher in the HALC than in the bare HALC and LALC, respectively. However, there was no significant change in the peak area of this compound in the LALC groups according to the myogenic differentiation. In addition, 2,5-Dimethylpyrazine is a volatile compound with a roasted beef-like flavor, and it is formed by the Maillard reaction in cooked beef.[43,44] Pyrazine was only detected in the HALC with differentiated muscle cells. To investigate the relationship between the cell differentiation and the flavor analysis results, GC analysis was performed for the undifferentiated samples (Supplementary Fig. 10). Benzaldehyde was detected in both HALC and LALC with undifferentiated myoblasts, but there was no difference in the peak area ratio of the benzaldehyde in the HALC and LALC. In contrast, pyrazine was not detected in either group. These results imply that the degree of myogenesis significantly affects the flavors of cultured meat, as it increases the content of the Maillard reaction products. In summary, the HALC scaffold promoted myogenic differentiation, resulting in a protein-rich muscle block with a meat-like texture and appetizing flavors.

## Optimization of the fat block: enhancing the lipid content and oily flavors

The effect of adipogenic differentiation on the sensory properties was investigated to optimize the fat block before fabricating the assembled cultured meat (ACM).

As illustrated in Fig. 4a(i), the bovine adipose tissue-derived mesenchymal stem cells (adMSCs) cultured on the LALC are expected to exhibit a higher adipogenic differentiation rate than cells on the HALC. Before investigating this rate difference, we confirmed the cell viability on the scaffolds using CCK-8 assay (Supplementary Fig. 11). During 3 days of proliferation, the cells in both groups exhibited similar viability. After 3 days of proliferation, the culture medium was changed to initiate the adipogenic differentiation, and the differentiation progressed for another 4 days. When the adipogenesis began, the morphology of the fibroblast-like shaped adMSCs changed into a round shape and abundant lipid droplets were formed inside their cytosols. The Oil Red O staining assay was performed to examine the lipid droplet formation in each group. Figure 4a(ii) shows the quantification results of the red-stained lipids in bar graph with the images of the red-stained lipid droplets. Evidently, there was a significant difference in the amount of lipid droplets in the two groups. In addition, the formation of lipid droplets was investigated for the undifferentiated adMSCs in both scaffolds and no significant difference was observed (Supplementary Fig. 12). Confocal images of the neutral lipid-stained cells were obtained for a more precise investigation of the viable adipocytes (Fig. 4b(i) and Fig. 4b(ii)). The results revealed that a higher amount of neutral lipids was deposited around the nuclei of the cells in the LALC. The lipid droplet coverage and number of lipid droplet per cell were significantly higher in LALC. However, it was only confirmed that the difference in the lipid droplet formation occurred depending on the scaffold stiffness, rather than precisely evaluating the adipocyte maturity. Therefore, it cannot be said that the adipocytes are fully mature on the scaffold, but it was confirmed that the degree of adipogenic differentiation was regulated by the scaffold stiffness. Furthermore, proteomic analysis was performed for a clearer verification of the difference in the degree of adipogenic differentiation of two groups (Supplementary Fig. 13).

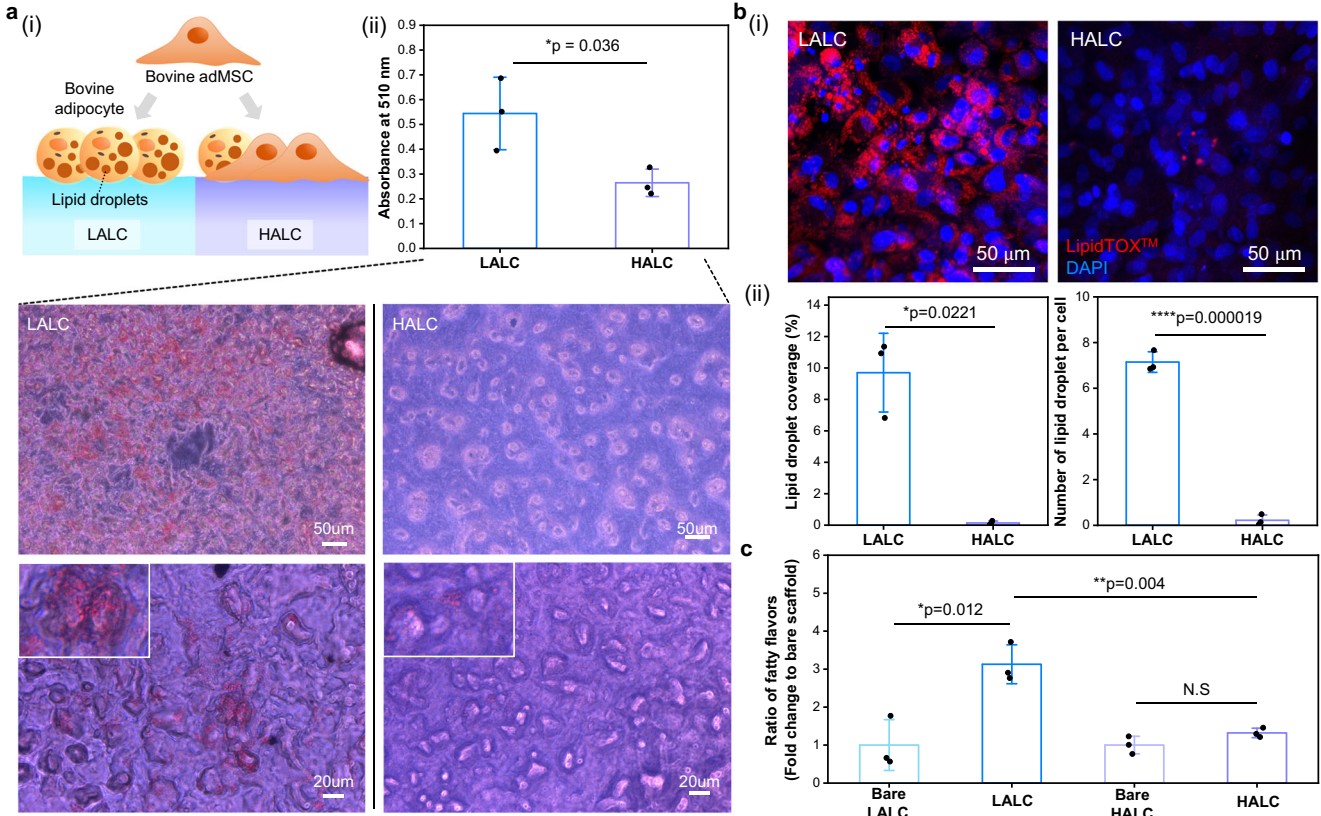

**Fig. 4 | Optimization of the fat block. a** (i) Illustration of the adipogenic differentiation of the adipose tissue-derived mesenchymal stem cells (adMSCs) depending on the scaffold, and (ii) the quantification of the adipogenic differentiation degree of LALC (blue graph) and HALC (purple graph) based on the lipid droplet staining performed using Oil Red O staining (mean ± SD, n = 3 independent experiments, two-tailed t-test). Scale bars: 50 μm (first row) and 20 μm (second row). **b** (i) Confocal images showing the lipid droplet formation analyzed by immunostaining using HCS LipidTOX™ Red Neutral Lipid Stain (red) and DAPI (blue). Scale bars: 50 μm. (ii) Quantitative assessments of the lipid droplet formation are shown in the graph based on the confocal images using Image J software (mean ± SD, n = 3 independent experiments, two-tailed t-test). **c** Peak area ratios of nonanal and 2-ethyl-1-hexanol are summed and normalized to that of the bare scaffolds (mean ± SD, n = 3 independent experiments, two-tailed t-test). Source data are provided as a Source Data file.

After differentiation, the proteins with higher expression in the LALC were selected and classified by performing gene annotation. The genes involved in lipid and fatty acid metabolism were confirmed to be up-regulated in the LALC. Furthermore, the expression of the adipogenesis-related proteins was confirmed to be higher in the LALC compared to that of HALC (Supplementary Fig. 14). Specifically, the representative marker for adipocyte differentiation, ADIPOQ, was expressed significantly higher in LALC. These results indicate the significantly different degree of adipogenesis depending on the scaffolds. For the undifferentiated groups, the amount of the proteins whose expression varied depending on the scaffold was fewer than that of the differentiated groups (Supplementary Fig. 15). However, the up-regulation of the proteins related to lipid synthesis was higher in the LALC with undifferentiated adMSCs than that in the HALC, suggesting that lipid metabolism-related pathway was influenced by scaffold stiffness even before differentiation was induced.

Next, the changes in the stiffness and flavor upon cell differentiation were investigated. The change in the stiffness of the differentiated group was not significant compared to those of the bare and undifferentiated groups (Supplementary Fig. 16). However, the effect of adipogenesis on the physical properties of cultured meat should be investigated in further studies since adipocyte maturation was not evaluated at this stage. Although the stiffness did not vary depending on the scaffolds, the increment of the fatty flavor was higher in the LALC with differentiated adMSCs than in the HALC group (Fig. 4c and Supplementary Table 5). In the LALC group, the sum of the peak area ratio of nonanal and 2-ethyl-1-hexanol with the same fatty flavor note

increased significantly compared to that of the bare LALC. Both compounds are known as volatile compounds with fatty flavor resulting from lipid oxidation in meat fat.[45–47] This flavor enhancement only occurred in the LALC group, and no flavor change was observed in the undifferentiated groups (Supplementary Fig. 17). In short, the LALC had enriched lipid content and fatty flavors by stimulating the adipogenic differentiation of the bovine adMSCs. Therefore, we selected the LALC as a fat block for the final cultured meat structure.

## Assembly of the muscle and fat blocks: production of an appetizing cultured meat

After optimizing the muscle and fat blocks, we assembled the blocks to produce a small-sized cultured beef that embodies the beef-like sensorial properties derived from the differentiated cells. The assembly was performed using mTG, which can form peptide bonds between the gelatin-containing scaffolds. After applying the mTG solution to the surface of each block, we stacked the blocks vertically to prepare the ACM (Fig. 5a). Subsequently, we compared the nutritional values, textures, and flavors of the ACM and slaughtered beef. The ratio of muscle tissue to adipose tissue varies depending on the cutting parts of beef. Therefore, a beef brisket with a muscle:fat ratio of 3:1 was used as a reference, and the ACM was prepared by assembling the muscle and fat blocks at the same ratio as that of the beef cut (Fig. 5b(i)). Particularly, three muscle blocks were assembled with one fat block to fabricate the ACM with a muscle:fat ratio of 3:1. The nutritional composition of the ACM and beef are compared in Fig. 5b(ii). The results confirmed that the ACM contained all three essential nutrients, similar

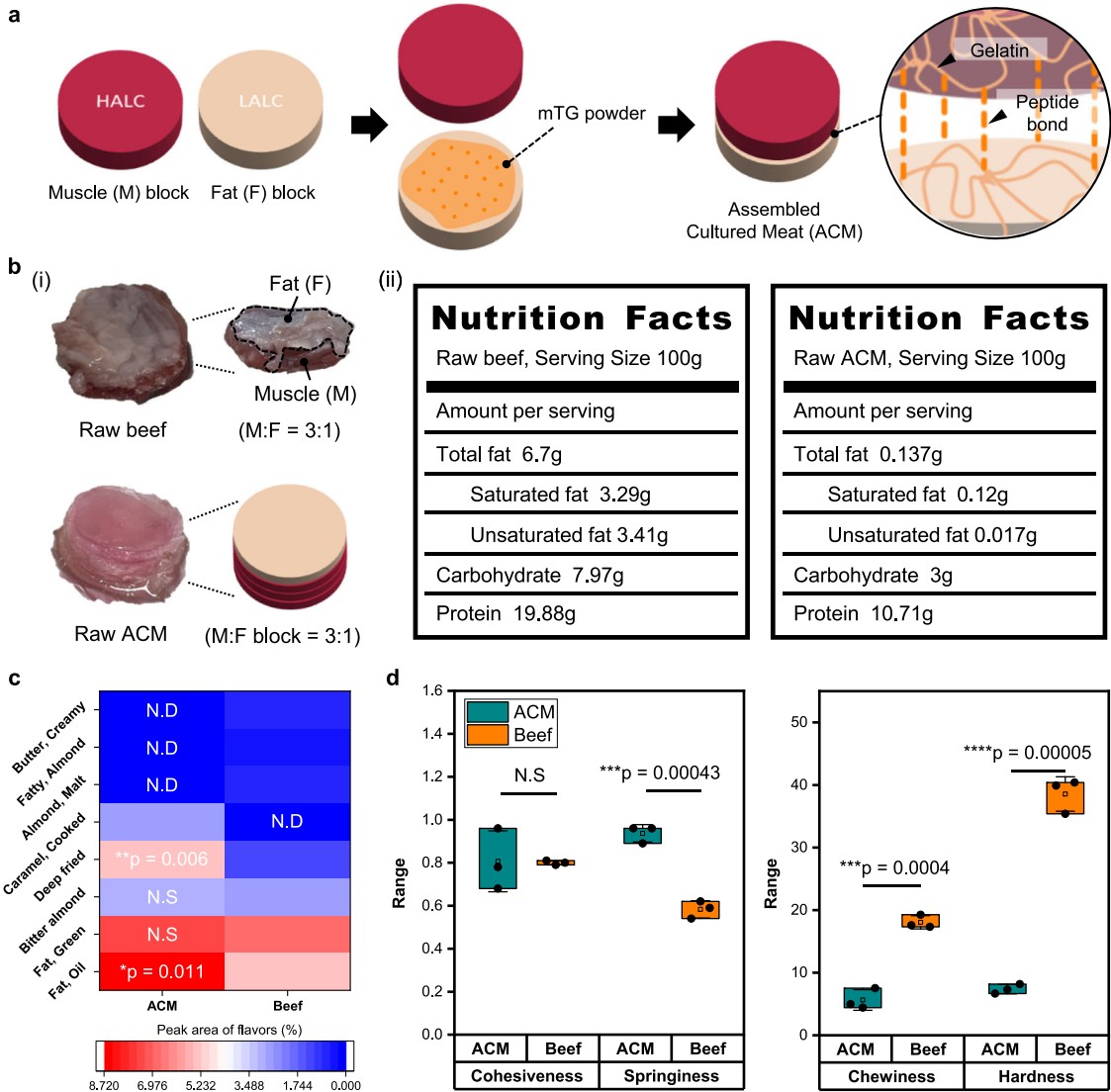

**Fig. 5 | Food analysis of the ACM. a** Illustration showing the production process of ACM. **b** (i) Images of raw beef and raw assembled cultured meat (ACM) and (ii) the nutritional analysis of each group. **c** Heatmap of the flavors detected in the ACM and beef after grilling at 180 °C (*n* = 3 independent experiments, two-tailed *t*-test). N.D indicates not detected and N.S indicates non-significant. **d** Texture profile analysis comparing the factors that determine the food texture of each group after grilling at 180 °C (mean(□) ± SD, *n* = 3 independent experiments, two-tailed *t*-test). Source data are provided as a Source Data file.

to beef. As the scaffolds are composed of gelatin and alginate, bare scaffolds also included proteins, fats, and carbohydrates. Therefore, the nutrition information of the bare ACM and undifferentiated ACM was evaluated (Supplementary Fig. 18). Compared to the bare ACM, the ACM with differentiated cells contained higher amount of proteins. This nutritional enrichment was not observed in the undifferentiated ACM. Although the protein content of the ACM was lower than that of beef brisket, the results confirmed that the control of the cell differentiation quality affected the nutritional value of the final cultured meat.

Texture and flavor analyses were performed to confirm if the ACM satisfies the sensory traits of beef. The samples were prepared by grilling at 180 °C for 10 min, and the color of both the samples was observed to change from red to yellowish brown owing to the occurrence of the Maillard reaction (Supplementary Fig. 19). The flavor analysis results (Fig. 5c and Supplementary Table 6) confirmed the presence of Maillard reaction compounds with assigned flavors in the ACM and beef. The results demonstrated that the flavor ratios of bitter almond and fatty flavors in the ACM were similar to those of beef.

These flavors were also detected in the bare and undifferentiated ACMs, but the peak ratios of these flavors were significantly higher in ACM owing to the abundant proteins and lipid droplets (Supplementary Fig. 20). Although some flavor compounds in beef were not detected in the ACM, the ACM and beef exhibited similar flavor notes, such as almond, fat, and fried-like flavors. These results indicate that the ACM was characterized by flavors that are similar to those of slaughtered beef because of its increased protein and lipid contents. To assess the texture of the ACM, texture profile analysis (TPA) was performed on the grilled ACM and grilled beef, and their cohesiveness, springiness, and hardness values were obtained (Fig. 5d). Subsequently, these values were multiplied to determine their chewiness. The results revealed that the ACM exhibited a more tender texture with higher springiness compared to beef, but their cohesiveness was similar. To investigate the effect of cell differentiation on the texture of ACM, TPA was performed for the bare ACM and undifferentiated ACM (Supplementary Fig. 21). The results revealed that the cohesiveness of the ACM was significantly higher than that of the bare and undifferentiated groups owing to the high amount of muscle proteins.

## Production of an engineered T-bone steak

To fabricate bigger size of cultured meat with beef-like properties, 2D muscle and fat blocks were assembled together. To increase the surface area of the blocks in contact with each other, the blocks were cut in the form of a puzzle. Thereafter, the blocks were assembled with a specific ratio of muscle and fat to produce an engineered T-bone steak. A T-bone steak is mainly composed of two parts of beef: strip loin and tenderloin. According to the United States Department of Agriculture (USDA), the protein:fat ratios of the strip loin and tenderloin are approximately 1.5–3:1 and 1–3.6:1, respectively.[48] Therefore, we prepared pieces of ACM by vertically bonding the muscle and fat blocks at a ratio of 1:1 and 3:1, respectively. Then, we horizontally assembled the pieces to prepare the engineered strip loin and tenderloin with muscle:fat ratios of 1.5:1 and 1:1, respectively (Fig. 6a). 19 muscle and 13 fat blocks were used for the strip loin ACM and 14 muscle and 14 fat blocks for used for tenderloin ACM. The cultured meat pieces composed of the muscle and fat blocks with a ratio of 3:1 were colored with red food coloring dyes to identify their locations. Finally, a cultured

T-bone steak composed of strip loin ACM and tenderloin ACM was obtained (Fig. 6b). When we grilled the steak ACM with olive oil at 180 °C for 10 min, browning was observed, indicating that the Maillard reaction occurred in the cultured meat, similar to a conventional steak.

Thus, we confirmed that the cultured meat that embodies the food-related properties of conventional beef can be fabricated with our 2D meat blocks. We also produced the bulk assembled cultured meat mimicking the different cuts of slaughtered beef, confirming the potential of our strategy to mimic both structural and sensorial characteristics of various cuts of slaughtered meat.

## Discussion

In this study, the organoleptic properties of the cultured meat were controlled by regulating the differentiation degree of muscle and fat cells with different scaffold stiffness. 2D hydrogels were used for scaffolds to control cell differentiation according to the physical properties of the scaffold without being affected by nutrition delivery or cell viability according to the porosity of the scaffold. Gelatin is a

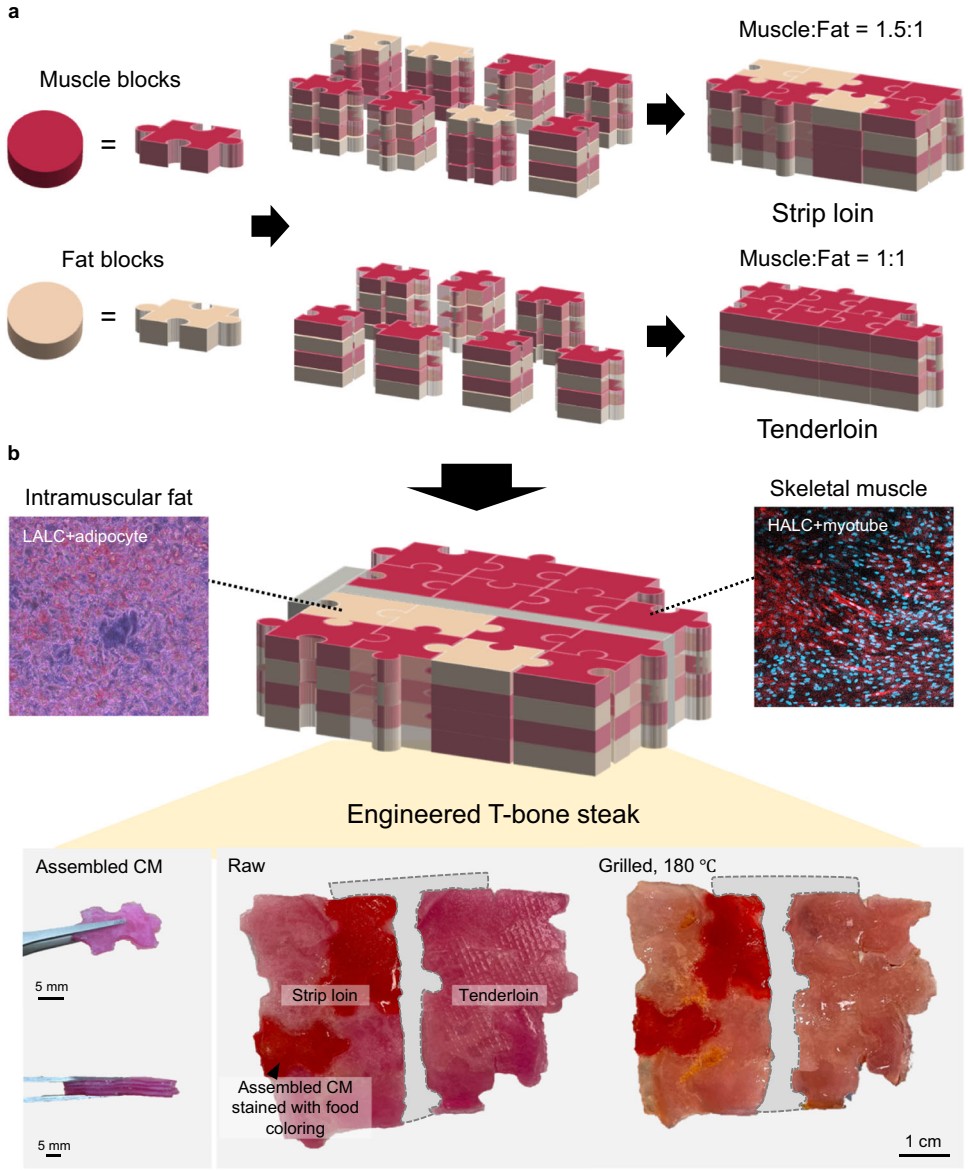

**Fig. 6 | Schematic showing the production of the engineered T-bone steak composed of muscle and fat blocks. a** Assembly process of muscle and fat blocks to produce a strip loin and tenderloin with different muscle:fat ratios. **b** Images of the raw and grilled engineered T-bone steaks consisting of the assembled strip loin and tenderloin. Scale bars: 5 mm (left) and 1 cm (right).

common biopolymer used in biomaterials which exhibits an inherently high cytoaffinity. Although there are recent studies reporting plant-derived scaffolds for cultured meat fabrication, only few studies have clearly confirmed whether plant-derived materials can provide the stable cell adhesion and cell differentiation environment as much as animal-derived biopolymers. As our study focused on the relationship between the different cell differentiation behavior and the organoleptic properties of cultured meat, a scaffold composed of biopolymers that can provide a stable cell differentiation environment is required. Therefore, gelatin was used in this study, but we expect that our strategy can be extended to various plant-derived scaffolds in the future.

The mechanical properties of the scaffold were optimized for the adipogenesis and myogenesis by controlling the crosslinking degree of alginate. Then, the degree of cellular differentiation of which regulated by the scaffolds affected both the nutritional and sensory properties of the cultured meat. As myoblasts differentiate, the nuclear fusion of the cells occurs, and the cell length increases along with the expression of proteins, including ECM.[49–51] From the perspective of cultured meat, we expect that the biological changes in the cells during the myogenesis would not only determine the amount of differentiated muscle cells but also the food quality of the meat, such as texture and flavor. The application of heat results in the denaturing of myosin heavy chains and actins, and they form globular aggregates as protein–protein interaction increases at temperature above 50 °C. In addition, it is known that the binding ability of myosin linearly increases as the heating temperature increases.[52] Therefore, the high amount of muscle proteins due to the myogenic differentiation affected the texture profile of the ACM as well as the stiffness of the muscle block. In our previous study, we discovered that cell proliferation increases the content of the protein participating in the Maillard reaction, thus affecting the flavor of the cultured meat.[53] Because protein synthesis increased during the myogenic differentiation, we expected higher amounts of Maillard reaction products in the HALC than in the LALC. We confirmed this using GC-MS analysis (Fig. 3f).

Fat is mainly composed of adipocytes with abundant lipid droplets, which determine the quality of meat, such as juiciness, flavor, and nutrition.[54,55] By optimizing the stiffness of the scaffold for adipogenesis, we expected to develop a fat block composed of highly differentiated adipocytes. The effect of the scaffold stiffness on the degree of adipogenic differentiation was investigated in Fig. 4. It was confirmed that the degree of adipogenesis was regulated depending on the stiffness of scaffold. However, the maturity of adipocyte was not specifically evaluated in this study. Therefore, the results showing non-significant effect of adipogenic differentiation on the stiffness of cultured meat should be further verified in future studies with fully matured adipocytes. Although stiffness was not dependent on the differentiation of adMSCs at such early differentiation stage, the formation of fatty flavor molecules was affected. Figure 4b revealed that lipid droplet formation was significantly higher in the LALC scaffold, which has similar stiffness to adipose tissue. Lipid droplets contain neutral lipids, consisting of triacylglycerols (TAGs).[56] As TAGs exhibit a structure in which three fatty acid molecules are ester-bonded to one glycerin molecule, TAG can be hydrolyzed to fatty acids by the lingual lipase in the mouth.[57] In addition, fat can be tasted via the interaction of these fatty acids with CD36, which is known as a fat taste receptor expressed by the taste bud cells on the tongue.[58] Therefore, the different amount of lipids formed in each scaffold implies the ability of the scaffolds to control the nutritional components and the taste of the cultured meat. Furthermore, fatty acids contribute to the flavors of slaughtered meat by creating volatile compounds upon oxidation. In Fig. 4c, we confirmed that a higher degree of adipogenic differentiation resulted in flavor enrichment of the fat block owing to the higher amount of lipid droplets.

These results suggest that the degree of differentiation of both muscle and fat cells contributed to the quality of the cultured meat.

Consequently, the cultured meat with characteristics similar to that of the conventional meat can be produced by assembling the muscle and fat blocks with enriched nutritional and sensory properties. We believe that this study provides a different perspective from previous studies in cultured meat production, and thus contribute to the development of cultured meat.

## Methods

### Inclusion and ethics
All the experimental procedures for animal slaughter were conducted in compliance with the Animal Care and Use Guidelines of Kangwon National University and were approved by the Institutional Animal Care and Use Committee (IACUC) of Kangwon National University (IACUC approval no. KW-220714-1).

### Synthesis of the gelatin/alginate hydrogel
For the commercial information of all the reagents in this study is provided in the Supplementary Table 7.

To prepare the hydrogel, first, 40% (w/v) fish gelatin solution was prepared by dissolving fish gelatin (GELTECH, Korea) in distilled water at 65 °C, and alginate was simultaneously dissolved in distilled water at 85 °C to prepare 8% (w/v) and 1% (w/v) alginate solution. To prepare the 2% (w/v) alginate hydrogel, 40% (w/v) of the gelatin solution was mixed with 8% (w/v) of the alginate solution at a volume ratio of 1:1 at 65 °C for 2 h. Subsequently, 2% (w/v) mTG solution was mixed with the gelatin/alginate solution at a volume ratio of 1:1 at 40 °C, and the solution was stirred vigorously. Lastly, the high alginate (HA) solution composed of 10% (w/v) fish gelatin, 2% (w/v) alginate, and 1% (w/v) microbial transglutaminase (mTG; AJINOMOTO, Japan) was obtained. To prepare the low alginate (LA) solution containing 10% (w/v) gelatin, 0.25% (w/v) alginate, and 1% (w/v) mTG, 40% (w/v) gelatin solution was mixed with 1% (w/v) alginate solution at a volume ratio of 1:1. Subsequently, 2% (w/v) mTG solution was added to the gelatin/alginate solution at a volume ratio of 1:1, and the solution was mixed vigorously. The hydrogel precursors were poured into 12-well plates (SPL) and gelled at 4 °C for 1 h. Thereafter, the hydrogels were incubated overnight at 37 °C to activate the mTG to achieve the gelatin crosslinking. After incubation, the hydrogels were washed with distilled water and immersed in calcium chloride (CaCl$_2$) solution for 1 h at room temperature for the alginate crosslinking. For LALC and LAHC, 0.125% (w/v) and 0.25% (w/v) CaCl$_2$ solutions were used, respectively. For HALC and HAHC, the hydrogels were immersed in 1% (w/v) and 2% (w/v) CaCl$_2$ solutions, respectively. Subsequently, the obtained hydrogels were washed with distilled water. Before cell culture, the hydrogels were immersed in 70% ethanol for 15 min and exposed to UV radiation for 1 h for sterilization.

### Characterization of the hydrogels
The crosslinking degree of the alginate network of the prepared hydrogels was compared using Raman spectroscopy (XploRA™ PLUS, HORIBA, Japan).

The degradation degrees of the LALC and HALC was quantified by measuring the weight loss of the hydrogels. The samples were immersed in the sterilized bovine myoblast conditioned medium for 12 days at 37 °C to mimic the cell culture condition. The conditioned medium was replaced with fresh conditioned medium every other day. The conditioned medium was the growth medium collected from the subcultured bovine myoblast at passage 2–3 on the 2 day of cell culture. Subsequently, the conditioned medium was filtered using 0.2 μm cellulose filter (Advantec) for sterilization and stored at 4 °C until usage. The weight loss (%) was calculated using the Eq. (1).

$$\text{Weight loss (\%)} = [(\text{Initial weight of the hydrogel} \\ - \text{Weight of the hydrogel on day 12}) \\ \times 100(\%)]/\text{Initial weight of the hydrogel} \tag{1}$$

Compressive tests were performed using a universal tensile machine (Model 3366, Instron, USA). Briefly, the scaffolds were prepared with the same size of 3.46 cm,[2] and a sled weight of 10 N was applied to the scaffolds at a constant displacement rate of 1 mm/min until disruption.

## Cell isolation

The bovine primary myoblasts were isolated from harvested gluteobiceps or semitendinosus muscle tissues, and the bovine adipose tissue-derived mesenchymal stem cells (adMSCs) were extracted from the intramuscular or subcutaneous adipose tissues. All the tissues were harvested randomly from 29–31-month-old male or female Hanwoo cattle slaughtered at a local slaughterhouse (Kwell LPC, Hongcheon, Korea). The retrieved skeletal muscle and adipose tissues were used to isolate bovine primary myoblasts and adipose tissue-derived mesenchymal stem cells (adMSCs), respectively. For the myoblasts retrieval, the harvested muscle tissues were washed once in 70% (v/v) ethanol (Samchun Chemical, Korea, E0220) and twice in 2% (v/v) antibiotic-antimyocotic solution diluted in Dubecco's phosphate-buffered saline. The muscle tissues were cut into small pieces followed by a digestion process with 0.2% (w/v) collagenase type II (Worthington Biochemical Corporation, Lakewood, NJ, USA) dissolved in high glucose-Dulbecco's modified eagle medium (HG-DMEM) at 37 °C for 30 min. Thereafter, the muscle tissue fragments were incubated at 37 °C for 5 min in HG-DMEM supplemented with 1% (w/v) pronase (Calbiochem, Darmstadt, Germany) for complete digestion. Subsequently, the digested muscle tissues were re-suspended in HG-DMEM supplemented with 2% (v/v) heat-inactivated fetal bovine serum. To obtain cell pellet, the digested tissue solution was centrifuged at 1500 x g for 4 min and re-suspended in red blood cells (RBCs) lysis buffer for 10 min at room temperature. The solution was filtered through a 70-µm cell strainer (SPL) and centrifuged at 1500 x g for 4 min. Subsequently, the pellet was re-suspended in the myoblast proliferation medium composed of HG-DMEM-supplemented 10% (v/v) FBS, 5 ng/ml basic fibroblast growth factor (bFGF), and 1% (v/v) antibiotic–antimycotic. To isolate the myoblasts from the muscle-derived primary cells, muscle-derived primary cells ($5 \times 10^5$) were seeded onto a 35-mm culture dish (SPL) using a myoblast proliferation medium. After 24 h, non-adherent cells were removed, and the remaining adherent cells were cultured at 37 °C in a humidified atmosphere of 5% $CO_2$ in air, and the medium was replaced at 2 days intervals. The adherent cells were collected at a cell confluency of 50–60% by treating the cells with 0.05% trypsin ethylene-diamine-tetraacetic acid (trypsin-EDTA) to obtain the bovine myoblasts.

To isolate the adMSC, the harvested adipose tissues were washed with 70% ethanol and 2% (v/v) antibiotic–antimycotic solution diluted in DPBS. Thereafter, the clean adipose tissues were cut into small pieces using surgical scissors. The tissue pieces were dispersed in 0.75% (w/v) collagenase type II dissolved in 0.25% trypsin-EDTA at 37 °C for 30 min. During the dispersion process, the small tissues-containing tubes were shaken intensively for 30 min at 37 °C, and the digested adipose tissues were re-suspended in low-glucose Dulbecco's modified Eagle's medium (LG-DMEM) supplemented with 10% (v/v) FBS. After filtering using a 100-µm cell strainer, the pellets descended by centrifuging at 415 x g for 5 min. Subsequently, the red blood cells were eliminated by re-suspending the pellet in RBC lysis buffer for 10 min at room temperature. The RBC-free primary cells were descended by centrifuging at 415 x g for 5 min followed by re-suspension in LG-DMEM supplemented with 10% (v/v) FBS and 1% (v/v) antibiotic–antimycotic (herein referred to as adMSC proliferation medium). The adipose-derived primary cells ($1 \times 10^4$) were seeded onto a 35-mm culture dish and cultured for 6–8 days in an adMSC proliferation medium at 37 °C in a humidified atmosphere of 5% $CO_2$ in air. Subsequently, cells that did not adhere to the culture dishes were discarded by replacing the adMSC proliferation medium at 2 days-

intervals, and the colony-forming units-fibroblast (CFU-F) formed on the culture dishes were harvested using 0.25% trypsin-EDTA. Finally, bovine adMSCs were obtained.

## Cell culture

Bovine myoblasts and bovine adMSCs were sub-cultured on TPP® tissue culture dishes (Sigma Aldrich). The cell subculture was conducted using 1X phosphate buffered saline (PBS), 0.025% trypsin-EDTA, and the growth medium of each cell group. The cultured bovine myoblast (Passage 3) and adMSC (Passage 3) were seeded on the hydrogels. The seeding density of the myoblast and adMSC were $3 \times 10^4$ cells/12 well molded scaffold and $2 \times 10^4$ cells/12 well molded scaffold, respectively. The myoblasts were proliferated for 4 days, after which the medium was replaced with the myoblast differentiation medium, which is composed of HG-DMEM supplemented with 5% (v/v) horse serum and 1% (v/v) penicillin-streptomycin-glutamine (PS). Further, the adMSCs were proliferated for 3 days to achieve a cell confluency of approximately 90%, after which the medium was replaced with the adipose differentiation medium. The adipose differentiation medium was composed of LG-DMEM supplemented with 5% (v/v) FBS, 10 µM insulin from bovine pancreas, 1 µM dexamethasone, 10 µM ciglitizone, and 100 µM oleic acid. The medium was replaced every other day.

## Cell viability analysis

The myoblast viability was assessed on day 2 and day 4 during the proliferation period by performing cell counting using the Cell Counting Kit-8 (CCK-8, D-Plus™ CCK cell viability assay kit, Dongin LS, Korea). Additionally, the cell viability of adMSCs was evaluated by performing the CCK-8 assay on day 2 and day 3 during the proliferation period.

## Cell differentiation analysis

Immunostaining was performed according to the following protocol to confirm the myotube formation. First, the cells were fixed with 10% neutral buffered formalin and thoroughly washed. Subsequently, the samples were treated overnight to inhibit any non-specific antibody binding using a blocking solution containing 2% (w/v) bovine serum albumin (BSA), 0.3% (v/v) Triton™ X-100 solution, and 10% (v/v) horse serum in 1X PBS. Myosin heavy chain antibody (MF 20, DSHB, ID:AB2147781) was diluted 100 folds in the diluent solvent composed of 10% (v/v) horse serum and 2% (w/v) BSA and was used to treat the samples for 2 h at room temperature. The samples were washed twice with 1X PBS and once with 0.025% (v/v) triton X-100. Thereafter, the samples were treated with the secondary antibody (Donkey anti-mouse Alexa flour 594, Thermo Fischer, #A21203), which was diluted to 400 folds in the same diluent solvent of MF 20, for 1 h at room temperature. After the washing procedure, the sample was treated with DAPI (Sigma Aldrich, #D9542), diluted to 250 folds in 1% (w/v) BSA solution, for 30 min to stain the cell nuclei in the scaffolds. The stained cells were observed using a confocal laser scanning microscope (LSM 980, Carl Zeiss).

For the adipogenic differentiation analysis, lipid staining was performed using the Oil Red O stain kit (abcam) and the associated staining protocol.[59] The fluorescent staining of the lipid droplets was performed using the 200-folds diluted LipidTOX™ (HCS LipidTOX™ Red Neutral Lipid Stain, Invitrogen, #H34476). The fluorescent staining protocol was the same as that of MF 20. For lipid droplet coverage data, the area percentage of the LipidTOX™ stained lipids in the total image area was calculated. The number of lipid droplet per cell was calculated by dividing the number of the LipidTOX™ stained lipid droplets by the number of DAPI.

For the rheological characterization of the cellular differentiation, frequency-sweep test was conducted using a rheometer (MCR 302, Anton Paar, Austria) at 37 °C. The angular frequency ranged from 0.1 to 10% (Shear strain at 1%). The Young's modulus (E) of the sample was

calculated using the Eqs. (2) and (3).

$$G = \sqrt{(G'^2 + G'^2)} \qquad (2)$$

$$E = 2G(1 + v) \qquad (3)$$

$G$ is the shear modulus calculated using the storage modulus ($G'$) and loss modulus ($G''$), and $v$ is Poisson's ratio; in this study, $v = 0.5$.

## Protein analysis

For the protein analysis, the cells were detached from the scaffolds by treating them with 0.025% trypsin-EDTA. The detached cells were centrifuged at 264 x g for 4 min, followed by the aspiration of the filtrate. To wash out the residual medium, the cells were resuspended in 1X PBS and centrifuged. This process was repeated to collect all the cells from the scaffold until there was no attached cells on the scaffold when observed through an optical microscope. The washing procedure was repeated three times. Finally, a cell pellet was prepared. Thereafter, radioimmunoprecipitation assay (RIPA) cell lysis buffer (RIPA Lysis and Extraction Buffer) was added to the cell pellet to lyse the cells to extract the proteins. After vortexing and pipetting the cell lysate solution, the solution was stored at 4 °C for 30 min. Subsequently, the solution was centrifuged at 9724 x g for 10 min to remove the cell debris, which sink after centrifugation. Finally, we obtained the pure solution of the extracted proteins for the BCA assay and LC-MS proteomic analysis.

The total protein concentration of the samples was quantified using the Pierce™ BCA Protein Assay Kit (Thermo Fishcer Scientific) according to the protocol included in the kit. The cellular expression of the MHC was analyzed by performing a bovine MYH1 ELISA (MyBioSource).

For the LC–MS-based proteomic analysis, protein solution was digested in an S-Trap mini spin column (Protifi, USA), according to the manufacturer's instructions. The samples were homogenized with 5% sodium dodecyl sulfate (SDS; Sigma-Aldrich) in 50 mM tetraethylammonium bromide (TEAB; Thermo Fisher scientific). The proteins were heated and alkylated with iodoacetamide, followed by the acidification procedure using phosphoric acid. Subsequently, the protein solution was mixed with binding buffer (90% methanol; 100 mM TEAB; pH 7.1) and loaded onto the filter, washed with a solution of 90:10 methanol:50 mM TEAB. The protein was digested using trypsin gold (Promega) at 37 °C at a protein-to-enzyme ratio of 10:1 (w/w) to obtain peptides. After eluting the peptides with elution buffers (50 mM TEAB in water, 0.2% formic acid in water, and 50% acetonitrile/0.2% formic acid in water), the peptides were labeled using TMTpro™ 16plex Label Reagent Set according to the manufacturer's instructions (Thermo Scientific). All the labeled peptides were combined, after which high pH reversed phase liquid chromatography (RPLC) fractionation was performed using NexeraXR HPLC system (Shimadzu). Analysis was performed on the prepared peptides using a LC-MS/MS system consisting of an UltiMate 3000 RSLCnano system (Thermo Fisher scientific) and a Orbitrap Eclipse Tribrid mass spectrometer (Thermo Fisher Scientific) equipped with a nano-electrospray source (EASY-Spray Sources, Thermo Fisher Scientific). The genes related to the analyzed proteins were classified according to the DAVID Bioinformatics Resources (2021 version). Raw data of protein expressions of myoblasts and adMSCs are available at Figshare database (https://doi.org/10.6084/m9.figshare.24712371.v3).

## Flavor analysis

The flavor compound was detected using a gas chromatography–mass spectrometer (GC-MS; Agilent 8890 GC system-Agilent 5677B MSD, Agilent Technologies). The specimens were prepared by following the protocol employed in our previous study using Headspace-Solid Phase Microextraction (HS-SPME).[53] First of all, volatile and semivolatile compounds from the samples were adsorbed to the carboxen/polydimethylsiloxane/divinylbenzene (CAR/PDMS/DVB) fibers for 40 min. Then, the fibers were injected to the GC-MS with split (20:1) mode. The inlet temperature was 250 °C whereas the temperature of the column oven was remained at 40 °C for the first 5 min and rasied to 240 °C at the raise speed of 4 °C/min. The column oven temperature was then remained at 240 °C for 20 min for analysis.

Thereafter, the specific flavors were assigned to each detected compound based on the flavor information obtained from the Flavor Extract Manufacturers Association of the United States (FEMA; https://www.femaflavor.org/flavor-library) lists and the Good Scents Company (TGSC) information system (https://www.thegoodscentscompany.com/).

## Characterization of the assembled cultured meat

Assembled cultured meat (ACM) was produced by assembling the muscle and fat blocks using the mTG solution (30% (w/v) mTG dissolved in distilled water). The mTG solution was spread on the surface of the blocks, after which the blocks were tightly wrapped with a PVC wrap and incubated at 37 °C for 1 h.

For the nutritional analysis, the ACM was prepared using larger meat blocks to meet the minimum test standards. The blocks were produced using the scaffolds molded in a 100-mm petri dish (SPL). The nutrition analysis was conducted by OATC INC, which is an institution certified by the Korean Laboratory Accreditation Scheme (KOLAS). To measure the protein content, the samples were prepared by digesting the ACM and beef with sulfuric acid. The digested samples were distilled with 40% NaOH solution and titrated with 0.1 N HCl solution. The protein content was calculated using the Eq. (4).

$$
\begin{aligned}
\text{Protein (g/100 g)} = & [(V_{\text{HCl,used}} \times V_{\text{HCl,blank}} \times M_{\text{HCl}} \times \text{N}) \\
& / (\text{Mass of the specimen (mg)})] \qquad (4) \\
& \times \text{Nitrogen coefficient} \times 100
\end{aligned}
$$

$V_{\text{HCl,used}}$: Volume of the HCl used (mL), $V_{\text{HCl,blank}}$: Volume of the HCl for blank (mL), $M_{\text{HCl}}$: Molarity of the HCl, N: Atomic weight of nitrogen

Carbohydrate content was calculated by subtracting the content of the protein, ash, fat, and water measured in the specimen from the total mass of the specimen. Fat content was measured using a flame ionization detector.

The texture profile analysis (TPA) was performed using the TXA™ Multi-axis Micro-Texture analyzer (YEONJIN S-Tech Corporation) at room temperature. A 3 kgf loadcell was used, and the ACM was compressed by 40% under ambient conditions (humidity).

## Software information

All graphs were produced by OriginPro (version 2018) and confocal images were produced by ZEISS ZEN (version 3.2). Image J software (version 1.52a) was used for quantification of the fluorescent images.

## Statistics and reproducibility

The data are reported as mean ± SD. Statistical analyses in all data were performed by two-tailed Student's $t$-test using Microsoft Excel. All experiments were repeated three times independently with similar results including confocal images.

## Reporting summary

Further information on research design is available in the Nature Portfolio Reporting Summary linked to this article.

# Data availability

All data is available in the main text or supplementary materials. Source data are provided with this paper. The protein expression data

generated in this study are available at "Figshare database [https://doi.org/10.6084/m9.figshare.24712371.v3]". Any additional requests for information can be directed to, and will be fulfilled by, the corresponding authors. Source data are provided with this paper.

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

## Acknowledgements

This work was supported by the Technology Innovation Program (Alchemist Project, 20012384, Jinkee Hong), funded by the Ministry of Trade, Industry & Energy (MOTIE, Korea) and by the Bio & Medical Technology Development Program of the National Research Foundation (NRF) funded by the Ministry of Science & ICT (2019M3A9H110378622, Jinkee Hong). This work is also supported by National Research Foundation of Korea (NRF) grant funded by the Korea government(MSIT)(No. 2021R1A4A3030268, Sangmin Lee) and also supported by Korea Research Institute of defense Technology planning and advancement (KRIT) grant funded by the Korea government (Defense Acquisition Program Administration (DAPA)) (No. KRIT-CT-21-034, Smart CBRNe Sensor Research Laboratory, Jinkee Hong).

## Author contributions

M.L., S.L., and J.H. contributed to design the study and performed experiments. S.P., B.C., and W.C. contributed to the in vitro experiments and analyzed the data. H.L., J.M.L., and S.T.L. provided primary cultured bovine cells and contributed to the biological experiments. K.H.Y., D.H., G.B., and H.H. analyzed the protein expression data. W.G.K. also contributed to the in vitro experiments.

## Competing interests

The authors declare no competing interests.
