## [Peer review file · Nature Communications]

REVIEWER COMMENTS

Reviewer #1 (Remarks to the Author):

The authors reported here an interesting block-assembling method for the construction of cultured meat. Gelatin-alginate scaffolds were crosslinked to obtain soft and hard characters for suitable differentiation properties of both adipocytes and myoblasts. After the differentiation of both cells on the suitable gel surfaces, the obtained gels were assembled by transglutaminase, as a piece of the block. Finally, they provided the engineered T-bone steak-like tissues even without bone. This method seems to be interesting and worthwhile for researchers in the specific cultured meat area. However, unfortunately, this reviewer cannot find a strong and broad scientific impact and a novelty interdisciplinary. For example, stiffness adjustment is currently a common way in the mechanobiology area and both soft gels for adipogenesis and hard gels for myogenesis are general knowledge. There was no author's effort or new finding here although they optimized the stiffness of the scaffolds by changing the ratio of gelatin and alginate. Assembling a piece of tissue to make larger meat-like tissues was also reported recently (Nat. Commun. 2021). Moreover, I cannot agree with calling it a "meat block" because of the very few amount of cells on the scaffolds. The 10^4 cells on the gels with 2.1 cm in diameter are very few as compared to the real meat. People will eat "scaffolds" mostly... In conclusion, due to low scientific novelty for the broad areas, no further review is recommended.

Reviewer #2 (Remarks to the Author):

Comments to authors: Block-Assembled Edible Scaffold for Cultured Meat

The authors fabricated a structured meat analogue by applying the Lego/puzzle blocks approach. The authors were also able to control the sensory characteristics of the muscle and fat blocks by regulating cell differentiation to tune the stiffness of the cultured meat product. Moreover, the authors also explored cultured meat's food characteristics, including texture, flavour and nutrition facts. This is good work and in-depth research on the individual building blocks of meats (fats and muscles). The authors rightly pointed out that sustainability is one of the main reasons behind the attention on cultured meat. Even though the microbial transglutaminase and alginate are non-animal based, fish gelatin - a biomaterial derived from an animal source, is still being used, which is contradictory. Regardless, the paper is a proof of concept for the feasibility of the technique for manufacturing cultured meat. However, this work still contains some concerns that need to be addressed. Please see the major and minor point-by-point comments below.

Comments to authors: Block-Assembled Edible Scaffold for Cultured Meat

The authors fabricated a structured meat analogue by applying the Lego/puzzle blocks approach. The authors were also able to control the sensory characteristics of the muscle and fat blocks by regulating cell differentiation to tune the stiffness of the cultured meat product. Moreover, the authors also explored cultured meat's food characteristics, including texture, flavour and nutrition facts. This is good work and in-depth research on the individual building blocks of meats (fats and muscles).

The authors rightly pointed out that sustainability is one of the main reasons behind the attention on cultured meat. Even though the microbial transglutaminase and alginate are non-animal based, fish gelatin - a biomaterial derived from an animal source, is still being used, which is contradictory. Regardless, the paper is a proof of concept for the feasibility of the technique for manufacturing cultured meat. However, this work still contains some concerns that need to be addressed. Please see the major and minor point-by-point comments below.

Major comments:

- Line 147-149: "The size of the pores decreases as the crosslinking degree and alginate concentration increase, and there are almost no micropores in the HAHC." The line is slightly confusing as contrasting points are brought up in the same line. Could the authors clarify if it meant no macropores or micropores in HAHC since the earlier part of the sentence implies that there will be fewer macropores and more micropores in HAHC?
- Line 387: the authors claimed that this assembled cultured meat (ACM) is scalable because it can be scaled into the cm size. However, this does not mean that this ACM can be scaled up for mass production. Scalability is one of the biggest limitations of some proposed methods for manufacturing cultured meat. As the proposed technique still involves a physical assembly of the muscle and fat blocks with microbial transglutaminase, there could be some potential concerns in the large-scale manufacturing setting as well. Please clarify this point or discuss the future plan on how to mass produce ACM.
- Why did the authors choose the 5th day and 8th day for the myogenic differentiation analysis study, despite the study leading to producing lab-grown meat from scratch of cells? Please clarify this point.
- Why is MHC the only protein of interest in studying myogenesis in LALC and HALC? Preferably should take more than one protein (Myosin with Myogenin or Myosin with Desmin)
- Time points of cell culture characterisations were unclear in the adipogenic differentiation study. When was oil red staining analysis performed to confirm the adipogenic differentiation? Please add this information
- Why did the authors choose the 12 days-conditioned medium for the degradation study? It would be more informative if the degradation study graph plots weight loss against the number of days.
- For weight loss calculation (in the supplementary file), should the equation be "Initial weight – the weight of hydrogel at day 12"? In the current equation format, this implies that the weight of the scaffold increased after 12 days which is not a degradation. Please recheck.
- The way authors show the relative proliferation in the supplementary file (Figures S3 and S7) is misleading. The normalised graph made it look like the cell proliferates faster/higher in LALC by 200% compared to the standard culture dish, which might not be the case. Is the relationship between CCK-8 and cell growth a linear relationship? Can authors provide a standard curve for that?

Minor comments:

- In Figure 2, what did the white arrow point to? Please explain it in the figure description

- Line 30: “different contents and forms of muscle tissues and adipose tissues” is a little unclear and awkward. Authors can consider paraphrasing it.
- Line 32: “these structural characteristics are highly desirable” What desirable structural characteristics are the authors referring to?
- Line 56: What defines/determines the “quality of each part” and what constitutes “food characteristics of the meat”?
- Line 63: The phrase “to implement the structure of meat” is a little unclear and awkward. Did the authors mean “to imitate” instead?
- Line 217: The use of “destruct” in this sentence is a little awkward. The authors can consider rephrasing it.
- Line 276-277: The authors may want to work on refining the captions for the subfigures in figure 4, especially for Figure 4a(ii) and (iii), as it is currently a little confusing.
- Line: 462 - 464. “The cells were seeded 12 well moulded on the scaffold”. Assuming the cells were seeded on the top layer of the scaffold. How do the authors ensure that the cells penetrate into the scaffold? Have the authors done H&E staining or Z-Stack confocal imaging?
- Line: 501 - 502. “For the protein analysis, the cells were detached from the scaffolds by treating with EDTA”. How do the authors collect all the cells from the scaffold? Please elaborate more on this point.

Reviewer #3 (Remarks to the Author):

The manuscript entitled "Block-Assembled Edible Scaffold for Cultured Meat" presents the attempt to fabricate large cultured meat constructs from smaller units of muscle and adipose tissues, where each was cultivated separately on alginate-gelatin based hydrogels and was later combined by applying enzymatic crosslinking with microbial transglutaminase. The differentiation of two bovine derived cell types, myoblast cells and mesenchymal stem cells, was assessed separately on two hydrogels with different stiffness and composition, as well as the resultant characteristics of the constructs, such as stiffness and presence of different volatile compounds. Multi-tissue constructs were then fabricated and analyzed in terms of nutritional, texture and flavor analysis. Horizontal and vertical assembly of muscle and adipose units was then applied to create larger marbled constructs.

While this work emphasizes the need to assess the crosstalk between cell behavior and resultant organoleptic and nutritional construct characteristics for cultured meat purposes, both novelty and quality are far from the required bar. The mentioning of previous works in the literature is extremely limited, and many of the tissue-engineering related concepts presented here were demonstrated before in other works. Both the used scaffolding materials and the technique used for larger structure assembly are not novel. Most of the presented conclusions are not supported by the currently shown DATA, either due to missing statistical analysis information or missing control groups, making further investigations mandatory. The description of several protocols and analysis are missing as well, preventing the reproduction of the presented research. Some parts of the paper should be reorganized. The comments below are extremely recommended for this article.

1. In general, one of the major limitations in thick-construct fabrication is the ability to support the growth of the cells in terms of sufficient material transfer. The method presented in this paper discusses cell seeding on-top of hydrogels, which are later stacked together by transglutaminase, to produce thicker constructs. However, one must ask: What was the thickness of each hydrogel layer? Were the cells capable of penetrating the hydrogels during cultivation? Were they evenly distributed throughout the hydrogels? The authors must provide 3D reconstructions of cell-seeded hydrogels, and analyze whether cell penetration occurred. This is relevant for both cell types, in the different hydrogel compositions.

If the cells could not infiltrate into the hydrogels, then the actual constructs produced in this work are hydrogels with thin cellular coatings – a configuration which is not conceptually beneficial for cultured meat.

2. In the abstract and introduction, please address previous works done in the field and be specific about what is the contribution of your work to the current knowledge. Some of the main concepts shown in your paper were already investigated previously. Moreover, some statements are inaccurate.

- line 26: "...only a small sized cultured meat with negligible similarity with conventional meat..." is not correct. Several works have already created large constructs for cultured meat, some of which included multiple cell-types:

<https://www.nature.com/articles/s41538-021-00090-7>

<https://www.nature.com/articles/s41467-021-25236-9>

<https://www.ncbi.nlm.nih.gov/pmc/articles/PMC9039213/>

<https://onlinelibrary.wiley.com/doi/full/10.1002/advs.202202877>

- lines 46-50: " Recently, researchers have reported strategies to produce cultured meat with various types of scaffolds, such as 3D printed bioink⁶, textured soy proteins⁷, and spinach⁸, to mimic the structure of meat. However, the complex technology and expensive extracellular matrix (ECM)-derived proteins involved in scaffold production limit the scalability of cultured meat." Please rephrase this.

More than one bioink was assessed. Here are more examples to mention:

<https://www.sciencedirect.com/science/article/abs/pii/S0142961222001260?via=ihub>

<https://pubmed.ncbi.nlm.nih.gov/36271729/>

Other decellularized plant tissues than spinach were also investigated for muscle tissue engineering:

<https://www.biorxiv.org/content/10.1101/2020.02.23.958686v1.abstract>

Moreover, some of these examples show the adherence, growth and differentiation of cells without any

use of ECM proteins.

- Lines 67-71: The paper suggests alginate-fish gelatin mixtures to create hydrogels as cultivation platforms for cultured meat. Why were these mixture of materials chosen? This is not elaborated enough. Moreover, such materials aren't novel for muscle or adipose tissue engineering, as they were assessed in previous works, such as:

<https://www.ncbi.nlm.nih.gov/pmc/articles/PMC5744339/>

<https://bmcbiotechnol.biomedcentral.com/articles/10.1186/1472-6750-12-35>

<https://pubmed.ncbi.nlm.nih.gov/22556122/>

<https://journals.sagepub.com/doi/10.1177/0885328216634057>

Additionally : please explain this choice, as animal-derived materials such as fish gelatin are not desired for cultured meat purposes and are unsustainable.

- Lines 60-66: The concept of tailoring hydrogel stiffness by changing composition or crosslinker concentration to promote adipogenesis or myogenesis was demonstrated widely in the literature. Please address previous attempts in your introduction and discussion.

- The paper suggests the assembly of several alginate-gelatin constructs to create larger multi-tissue type constructs, by using protein crosslinking with microbial transglutaminase. Previous works have shown the assembly of smaller scaffolds to achieve larger and complex structures (part of them also used transglutaminase):

<https://www.nature.com/articles/s42003-022-03852-5>

<https://www.nature.com/articles/s41467-021-25236-9>

<https://www.frontiersin.org/articles/10.3389/fbioe.2022.875069/full>

The use of microbial transglutaminase to create gelatin-based constructs was also described previously:

<https://www.ncbi.nlm.nih.gov/pmc/articles/PMC7825108/>

<https://www.nature.com/articles/s41538-019-0054-8>

It is crucial that the authors will address these aspects in the paper, and emphasize the novelty of their work in comparison to the current literature.

3. Statistical analysis to determine the significance between different groups is often missing or not properly described. Many graphs are shown without presenting the distribution of individual values around a mean value, some bars are with no STD or SEM markings, in the legends there is often no mention of number of repeats (n=?). Moreover, if there is no significance, please add "ns" in the graphs. Although the authors make various conclusions throughout the paper, one cannot rely on those without being given the supporting statistical analysis in a clear and full manner. Address the following:

Figure 2 bii, cii

Figure 2S,

Figure 3 b, d-f

Figure 3S,

Figure 4 c-d,

Figure S7,

Figure 5.

4. The authors make conclusions regarding the porosity of the 4 hydrogels according to Figure 2d. However, no qualitative assessment with statistical analysis was done to support them. Please perform a quantitative analysis if conclusions regarding the porosity/density of the constructs are to be made. Mind that the conditions under which SEM images are taken – such as performing freeze-drying – can alter the results. Moreover, add details regarding the SEM imaging and performed drying process (temperature, time), as those are missing in the materials and methods section.

5. The adipogenesis degree of the mesenchymal stem cells (Figure 4) is not evident according to the LipidTox staining. No lipid droplets were observed in a clear way. We wish to refer the authors to the following paper, where successful LipidTox staining is observed:

<https://pubmed.ncbi.nlm.nih.gov/35297273/> Additionally, the analysis of Oil-Red-O staining (Figure 4a_{ii}) is not clear, as no details of it are present in the paper. No control groups such as undifferentiated constructs are present.

Moreover, no additional biochemical quantitative assessments were made to verify it.

Please address this issue by performing additional qualitative (such as different/repeated staining) and quantitative (such as q-PCR) assessments to verify the adipogenic maturation degree on the two hydrogels, similarly to what you have done for myogenic differentiation experiments. Importantly: add control hydrogels seeded with cells yet not differentiated, as references. The verification of this issue is crucial for the entirety of this paper, as many subsequent conclusions and assumptions are derived from it by the authors.

6. Throughout the paper, observations regarding construct characteristics are concluded to have stemmed directly from the differentiation of seeded cells – a central concept the authors aim to demonstrate in this paper. However, important controls to rule out alternative explanations which could have affected the observed results were not performed. Therefore, the authors must perform additional studies and analyze them more carefully, with appropriate repeats and control groups, to differentiate between effects from alternative factors and the effect from cellular differentiation:

A. No statistical analysis was done, or a conclusive conclusion drawn, regarding the difference in construct degradation rates of the two hydrogels. This must be further repeated and analyzed (Figures S2), as it could affect results such as total protein content.

B. When two hydrogel types were analyzed for cell cultivation – either for myoblast cells (Figure S3, Figure 3) or mesenchymal stem cells (Figure S7, Figure 4) – construct properties were checked by the authors after the differentiation phase.

- No conclusions were drawn regarding the difference in cell proliferation rates on the two hydrogels, in both cell types. These must be statistically analyzed and discussed (Figures S3, S7).

- If there is indeed a difference in cellular density at the end of the proliferation phase, it could affect cell behavior and overall construct characteristics. Controls of hydrogels seeded with proliferated but non-differentiated cells should be analyzed and compared, to rule out alternative factors other than cell maturation. Please add such controls to the stiffness assessments (Figure 3 d, Figure 4c), total protein content assessment (Figure 3c), GC analysis (Figure 3e, Figure 4e), and protein expression analysis (Figure 3f) of the constructs.

- Different ECM secretion degree by seeded cells on the two hydrogel types could also have affected the obtained results, yet it wasn't checked at the end of cultivation of either cell type. Add ECM secretion analysis for each cell type (mesenchymal and myoblast cells), to better understand cell behavior on the different constructs (two hydrogel compositions). The results of these studies should be included in your discussion on final construct characteristics.

C. Controls of empty scaffolds of the two compositions assessed for cell cultivation should be analyzed to distinguish the effect of the composition itself on the obtained results, in the following assessments:

- Stiffness with/without cooking (Figure d_{ii}): the properties of the scaffolding materials can change significantly under such different treatments:

<https://pubmed.ncbi.nlm.nih.gov/31731744/>

D. Figure 5: controls of 1. empty, and 2. cell-seeded but not differentiated assembled cultured meat constructs, should both be analyzed and compared to cell seeded and differentiated constructs, as references to the obtained results. This is crucial for all 3 assessments : nutritional value, flavor, and texture.

7. Some parts of the paper are not well placed/organized.

- Please add a table describing the 4 types of prepared hydrogels, clarifying their compositions (%w/v of alginate and gelatin) and calcium crosslinking concentration.

- Conceptual explanations appear in the results section instead of the introduction. For example: lines 90-105, lines 164-166, lines 267-273, and lines 296-303 should be in the introduction, with

supporting references.

- Statements regarding the authors' hypothesis appear in the results, instead of the introduction/discussion, and do not mention supporting references. For example: lines 160-162. Pay attention to such phrases throughout the results section.
- The description of experiments and analysis should be in the materials & methods section. Please rearrange the legends of figures S2, S3, S7 accordingly.

8. There is missing or unclear information regarding performed protocols. Please provide the following in the materials and methods section, or other relevant places:

- Please provide detailed cell isolation protocols.
- Please provide detailed media used for myogenic and adipogenic differentiation.
- In construct fabrication: please state what were the dimensions (diameter and thickness) of individual hydrogels. Please clarify how you prepared the 2% -alginate containing hydrogels, in a similar manner to described for the 0.25% ones.
- SEM imaging: add details regarding the performed drying process (temperature, time) to the constructs, as those are missing.
- Scaffold degradation analysis: what was the exact medium used to incubate the scaffolds? was it collected during a different experiment each day, then frozen for subsequent use? Please elaborate as these are specific conditions where the described degradation occurred.
- What is the "fatty flavor" detected and measured by the GC? This is missing, as well as a detailed protocol.
- When analyzing cuts with different ratios between fat and muscle content: are the ratios calculated via mass? Volume? It must be mentioned in the relevant places in the paper.

Response letter

Journal: *Nature Communications*

Manuscript ID: NCOMMS-22-41801-T

Original Title: Block-Assembled Edible Scaffold for Cultured Meat

Revised Title: Block-Assembled Cultured Meat with Enriched Organoleptic Properties

Authors: Milae Lee, Sohyeon Park, Bumgyu Choi, Woojin Choi, Hyun Lee, Jeong Min Lee, Seung Tae Lee, Ki Hyun Yoo, Dongoh Han, Geul Bang, Heeyoun Hwang, Won-Gun Koh, Sangmin Lee, Jinkee Hong

Response to Reviewer #1

Reviewer(s)'s Comments to Author:

Reviewer 1.

Comments

The authors reported here an interesting block-assembling method for the construction of cultured meat. Gelatin-alginate scaffolds were crosslinked to obtain soft and hard characters for suitable differentiation properties of both adipocytes and myoblasts. After the differentiation of both cells on the suitable gel surfaces, the obtained gels were assembled by transglutaminase, as a piece of the block. Finally, they provided the engineered T-bone steak-like tissues even without bone.

This method seems to be interesting and worthwhile for researchers in the specific cultured meat area. However, unfortunately, this reviewer cannot find a strong and broad scientific impact and a novelty interdisciplinary. For example, stiffness adjustment is currently a common way in the mechanobiology area and both soft gels for adipogenesis and hard gels for myogenesis are general knowledge. There was no author's effort or new finding here although they optimized the stiffness of the scaffolds by changing the ratio of gelatin and alginate. Assembling a piece of tissue to make larger meat-like tissues was also reported recently (Nat. Commun. 2021). Moreover, I cannot agree with calling it a "meat block" because of the very few amount of cells on the scaffolds. The 10^4 cells on the gels with 2.1 cm in diameter are very few as compared to the real meat. People will eat "scaffolds" mostly...

In conclusion, due to low scientific novelty for the broad areas, no further review is recommended.

Response: We appreciate the reviewer's comments. Upon reflecting on the reviewer's comment, we realized that our description and the figures are insufficient to demonstrate the novelty of our findings. Therefore, we have revised the title, abstract, introduction, main text, and discussion in the manuscript, and performed additional experiments to clarify the novelty of our study. As the manuscript was revised thoroughly, the title of this study was changed to include a more specific expression. In the revised manuscript and figures, we changed Figure 2b(ii), Figure 3b-f, Figure 4a-c, Figure 5b(ii)-d, Supplementary Figure 3, Supplementary Figure 11, and Supplementary Table 3. In addition, we added supplementary Figure 4, Supplementary Figure 5, Supplementary Figure 6, Supplementary Figure 7, Supplementary Figure 8, Supplementary Figure 9, Supplementary Figure 10, Supplementary Figure 12, Supplementary Figure 13, Supplementary Figure 14, Supplementary Figure 15, Supplementary Figure 16, Supplementary Figure 17, Supplementary Figure 19, Supplementary Figure 20, Supplementary Table 1, Supplementary Table 2, Supplementary Table 4, Supplementary Table 5, and Supplementary Table 6. The p-values and the number of experiment repeats were added in all the figures and figure descriptions.

As the reviewer pointed out, the regulation of cell differentiation by scaffold engineering has been investigated for a long time. However, in this study, we investigated the effect of cell differentiation regulation on the organoleptic properties of cultured meat. Although various sensory characteristics, such as texture and flavor, are important factors that determine the quality of cultured meat, previous studies on cultured meat did not focused on these properties. In this study, the myogenic and adipogenic differentiation degree of bovine primary myoblasts and bovine adipose-derived mesenchymal stem cells (adMSCs) was regulated by controlling the mechanical properties of the edible scaffold with different ratio of calcium-crosslinked alginate. Additionally, we confirmed that the differentiation quality of cells affects the

nutritional and sensory properties of cultured meat. When the degree of myogenesis is high, the amount of the muscle proteins, as well as the total proteins synthesized by cells, increased, resulting in flavor and texture-enriched muscle block (Figure R1).

Figure R1 (Fig. 3). Optimization of the muscle block. **a** Immunostained myosin heavy chains (MHC) on differentiation day 5 and day 8. The nuclei stained with DAPI (blue), and the MHCs stained with MF20 (red). Scale bars: 20 μm (first row) and 100 μm (second row). **b (i)** Volcano plot showing the up-regulated (red dots) and down-regulated (blue dots) proteins in the HALC ($n = 3$ independent experiments, two-tailed t-test). Fold change indicates the intensity ratio of the protein expression in the HALC to that in the LALC. Proteins expressed by more than 1.5 folds in the HALC were sorted in the heatmap with the intensity of the protein expression. **(ii)** Gene annotation of the proteins in the heatmap. **c** Quantification of the MHC of differentiated bovine myoblasts in each group ($n = 3$ independent experiments, two-tailed t-test). **d** BCA assay results showing the total protein amount of the cells per scaffold ($n = 3$ independent experiments, two-tailed t-test). **e** Stiffness of the raw (ungrilled) groups (i) and grilled groups (ii) measured using a rheometer ($n = 3$ independent experiments, two-tailed t-test). Illustration of the stiffness change in the HALC with cells owing to an increase in the temperature from 37 °C (ii-a) to 60 °C (ii-b). N.S. means “non-significant”. **f** Peak area ratio of the flavors detected from each sample (left) ($n = 3$ independent experiments, two-tailed t-

test). Illustration of the flavor enrichment in the cooked HALC group (right). Error bars represent mean \pm s.d. Source data are provided as a Source Data file.

The differentiation of the myoblast in the HALC scaffold was higher than that in the LALC owing to the difference in the stiffness. The immunostaining, proteomic analysis, myosin heavy chain (MHC), and total protein evaluation (Figure R1a-d) confirmed that the degree of myogenesis varied with a change in the scaffold stiffness. Additionally, the amount of MHC increased with an increase in the differentiation of the myoblasts. This muscle protein is known to denature upon heating, thus affecting the texture of meat. We evaluated the changes in the texture of the experimental groups after heating and confirmed that the stiffness of the group with high amount of MHC increased upon heating (Figure R1e). Furthermore, proteins contributed to the flavor in meat upon heating as they participate in the Maillard reaction. We expected that HALC, the group with highly differentiated myoblasts, can produce more Maillard reaction products owing to the large amount of proteins. As expected, more savory and meat-like flavors were confirmed in the HALC group (Figure R1f). To rule out other factors except the cell differentiation degree, we performed same experiments for the undifferentiated groups. We confirmed that there was no significant difference in the amount of MHC and total protein in the different scaffold types (Figure R2 and R3).

Figure R2 (Supplementary Fig. 6). Amount of myosin heavy chain (MHC) expressed in the undifferentiated myoblasts in each scaffold. The expression of the myosin heavy chain in the undifferentiated samples was compared ($n = 3$ independent experiments, two-tailed t-test).

Figure R3 (Supplementary Fig. 7). Amount of total protein expressed in the undifferentiated myoblasts in each scaffold. Protein expression of the undifferentiated samples was evaluated using bicinchoninic acid (BCA) assay ($n = 3$ independent experiments, two-tailed t-test).

The results revealed that the scaffold stiffness exerted no notable effect on the sensory properties, including texture and flavor, of the undifferentiated groups (Figure R4 and R5).

Figure R4 (Supplementary Fig. 9). Change in the stiffness of the undifferentiated samples before and after cooking at 60 °C. The stiffness of the undifferentiated group was measured by frequency sweep at the angular frequency range from 0.1 to 1 (rad/s) ($n = 3$ independent

experiments, two-tailed t-test).

Figure R5 (Supplementary Fig. 10). Flavor analysis of the undifferentiated myoblast samples performed using GC-MS. Peak area ratio of the flavor detected in the undifferentiated samples are shown ($n = 3$ independent experiments, two-tailed t-test).

Additionally, we confirmed that the degree of adipogenesis affected the lipid droplet contents and the fatty flavor in the fat block (Figure R6). In meat fat, fatty acids create volatile compounds upon oxidation, which contributes to the flavor in meat. Here, we confirmed the higher amount of nonanal and 2-ethyl-1-hexanol in LALC. These compounds with the same fatty flavor are known as the volatile compounds detected from meat fat.

Figure. R6 (Fig. 4): Optimization of the fat block. **a (i)** Illustration of the adipogenic differentiation of the adipose tissue-derived mesenchymal stem cells (adMSCs) depending on the scaffold, and **(ii)** the quantification of the adipogenic differentiation degree of each group based on the lipid droplet staining performed using Oil Red O staining ($n = 3$ independent experiments, two-tailed t-test). Scale bars: 50 μm (first row) and 20 μm (second row). **b (i)** Confocal images showing the lipid droplet formation analyzed by immunostaining. Scale bars: 50 μm (first and second column) and 20 μm (third column). **(ii)** The quantitative assessment of the lipid droplet amount is shown in the graph based on the confocal images using Image J software ($n = 3$ independent experiments, two-tailed t-test). **c** Peak area ratios of nonanal and 2-ethyl-1-hexanol are summed and normalized to that of the bare scaffolds ($n = 3$ independent experiments, two-tailed t-test). Error bars represent mean \pm s.d. Source data are provided as a Source Data file.

Subsequently, the same analysis was performed the samples with undifferentiated adMSCs. The Oil Red O staining of the undifferentiated groups confirmed that there was no significant difference in the lipid droplet formation (Figure R7).

Figure R7 (Supplementary Fig. 12): Adipogenic differentiation degrees of the undifferentiated samples. Quantification of the Oil Red O staining of the undifferentiated adMSC cultured on each scaffold was performed on the proliferation day 3 ($n = 3$ independent experiments, two-tailed t-test). Error bars represent mean \pm s.d. Source data are provided a Source Data file.

Next, the fatty flavor of the undifferentiated groups was evaluated, and the results revealed that there was no significant difference between the groups (Figure R8). In addition, the flavor enrichment in LALC owing to the higher degree of adipogenesis was confirmed.

Figure R1-R7 indicated that the degree of cellular differentiation affected the texture and flavor of the cultured.

Figure R8 (Supplementary Fig. 16): Flavor analysis of the undifferentiated groups. Fatty flavors detected in the undifferentiated groups were normalized to that of bare scaffolds ($n = 3$ independent experiments, two-tailed t-test). Error bars represent mean \pm s.d. Source data are provided a Source Data file.

Finally, assembled cultured meat with improved flavor, texture, and nutritional value was fabricated by assembling the blocks with highly differentiated muscle and fat cells (Figure R9).

Figure R9 (Fig. 5): Food analysis of the ACM. **a** Illustration showing the production process of ACM. **b (i)** Images of raw beef and raw ACM and **(ii)** the nutritional analysis of each group. **c** Heatmap of the flavors detected in the ACM and beef after grilling at 180 °C ($n = 3$ independent experiments, two-tailed t-test). N.D indicates “not detected” and N.S indicates “non-significant”. **d** Texture profile analysis comparing the factors that determine the food texture of each group after grilling at 180 °C ($n = 3$ independent experiments, two-tailed t-test). Error bars represent mean \pm s.d. Source data are provided as a Source Data file.

We compared the food characteristics of the assembled cultured meat (ACM) to that of beef. It can be seen that the nutritional value, flavor, and texture. However, some flavors and texture factors of the ACM were similar to that of beef, and were not observed in the bare scaffold and

undifferentiated group. Although the nutrition facts of the ACM are lower than that of beef, the protein amount of the ACM was significantly higher than that of bare scaffold (Figure R10). As this behavior was not observed in the undifferentiated ACM, the assembled blocks with undifferentiated cells, this increase in the protein amount was attributed to the myogenic differentiation.

Figure R10 (Supplementary Fig. 17): Nutrition factors of the ACM, bare ACM, and undifferentiated ACM. Nutritional analysis of carbohydrates, fats, and proteins was conducted for the three experimental groups ($n = 3$ independent experiments, two-tailed t-test). Fats include saturated and unsaturated fats. Error bars represent mean \pm s.d. Source data are provided a Source Data file.

Furthermore, the flavor and texture of the undifferentiated groups were analyzed, and the organoleptic properties were observed to be significantly enriched in the ACM compared to the undifferentiated groups (Figure R11 and R12).

Figure R11 (Supplementary Fig. 19): Comparison of the flavors detected in each group. Flavors were detected in ACM, bare ACM (assembled scaffolds without cells), and undifferentiated ACM (assembled scaffolds with undifferentiated cells) by GC-MS. Significance was indicated as: a: non-significant (N.S) to bare ACM, b: $0.01 < p\text{-value} < 0.05$ to bare ACM, c: $0.001 < p\text{-value} < 0.01$ to bare ACM, d: $0.0001 < p\text{-value} < 0.001$ to bare ACM, e: N.S to undifferentiated ACM, f: $0.01 < p\text{-value} < 0.05$ to undifferentiated differentiated ACM, g: $0.001 < p\text{-value} < 0.01$ to undifferentiated ACM, h: $0.0001 < p\text{-value} < 0.001$ to undifferentiated ACM ($n = 3$ independent experiments, two-tailed t-test). Error bars represent mean \pm s.d. Source data are provided a Source Data file.

Figure R12 (Supplementary Fig. 20): Results of the texture profile analysis of each group. Factors that determine the texture (chewiness, hardness, cohesiveness, springiness) were measured for ACM, bare ACM, and undifferentiated ACM after grilling the samples at 180 °C

to confirm the effect of the cell differentiation quality control on the cultured meat texture ($n = 3$ independent experiments, two-tailed t-test). Error bars represent mean \pm s.d. Source data are provided a Source Data file.

In conclusion, we found out that controlling the differentiation quality of the cells significantly affected the food properties of the cultured meat. To the best of our knowledge, our study is the first research addressing the relationship between cell differentiation and the food properties of cultured meat. We believe that this study is the cornerstone research that suggests the importance of regulating cell differentiation for the production of high-quality cultured meat with the food properties of slaughtered meat. To emphasize this novelty, we revised the entire manuscript and added figures. In addition, we performed additional experiments on undifferentiated groups to rule out factors other than cell differentiation. Furthermore, we added the results of the statistical analysis to all the figures for clearer results.

As the reviewer pointed out, the muscle and fat blocks contain cells only on the surface of the scaffolds. However, as it was confirmed that various food properties of cultured meat are affected even by the cells on the scaffold surface, we believe that our findings can also be applied to various cultured meat scaffolds capable of containing cells therein. By carefully reflecting on the reviewer's comment, we could improve the quality of our manuscript. We hope that our revised manuscript will be positive for the reviewer.

Response to Reviewer #2

Reviewer(s)'s Comments to Author:

Reviewer 2.

Comments

The authors fabricated a structured meat analogue by applying the Lego/puzzle blocks approach. The authors were also able to control the sensory characteristics of the muscle and fat blocks by regulating cell differentiation to tune the stiffness of the cultured meat product. Moreover, the authors also explored cultured meat's food characteristics, including texture, flavour and nutrition facts. This is good work and in-depth research on the individual building blocks of meats (fats and muscles). The authors rightly pointed out that sustainability is one of the main reasons behind the attention on cultured meat. Even though the microbial transglutaminase and alginate are non-animal based, fish gelatin - a biomaterial derived from an animal source, is still being used, which is contradictory. Regardless, the paper is a proof of concept for the feasibility of the technique for manufacturing cultured meat. However, this work still contains some concerns that need to be addressed. Please see the major and minor point-by-point comments below.

Response: We sincerely appreciate been given the opportunity to improve the reliability and quality of our manuscript through the valuable comments from the reviewer. By carefully reflecting on the reviewer's comments, we were able to review our manuscript once again, and revised it to emphasize the novelty of our study. As the reviewer pointed out, fish gelatin is from an animal source, which can be contradictory to the sustainability of the cultured meat.

Thus, it would be ideal to use a plant-based protein that can stably provide focal adhesion for mammalian cells. Although there are recent studies reporting plant-derived scaffolds for the fabrication of cultured meat, only few studies have confirmed whether plant-derived materials can provide the stable cell adhesion and cell differentiation environment provided by animal-derived biopolymers. Gelatin is derived from collagen, an extracellular matrix (ECM) component, and it exhibits an inherently high cytoaffinity which can stably provide environment for cell differentiation. As our study focuses on the effect of the cellular differentiation behavior on the organoleptic characteristics of the cultured meat, a scaffold material with high cytoaffinity is required to rule out the effect of cell viability on the scaffolds. Therefore, gelatin was used for this work, but we expect that our strategy can be extended to various plant-derived scaffolds in the future.

Upon reflecting on the reviewer's comments, we realized that our manuscript had so many things to improve. Therefore, we revised the title and changed some of our data to address the novelty of our work. In addition, we reorganized both the main and supporting figures after reperforming our experiments and evaluating the significance of the results. We also added the undifferentiated groups for all the experimental data to rule out factors other than cell differentiation. As the manuscript was revised thoroughly, the title of this study was changed to include a more specific expression. In the revised manuscript and figures, we changed Figure 2b(ii), Figure 3b-f, Figure 4a-c, Figure 5b(ii)-d, Supplementary Figure 3, Supplementary Figure 11, and Supplementary Table 3. In addition, we added supplementary Figure 4, Supplementary Figure 5, Supplementary Figure 6, Supplementary Figure 7, Supplementary Figure 8, Supplementary Figure 9, Supplementary Figure 10, Supplementary Figure 12, Supplementary Figure 13, Supplementary Figure 14, Supplementary Figure 15, Supplementary Figure 16, Supplementary Figure 17, Supplementary Figure 19, Supplementary Figure 20,

Supplementary Table 1, Supplementary Table 2, Supplementary Table 4, Supplementary Table 5, and Supplementary Table 6. The p-values and the number of experiment repeats were added in all the figures and figure descriptions.

By performing additional experiments of the undifferentiated groups, we confirmed that the increase in the muscle protein expression (Figure R1) and stiffness of the undifferentiated group upon heating (Figure R2) were not dependent on the scaffold type. In addition, there was no significant difference in the flavor of the HALC and LALC group with the undifferentiated myoblasts after the Maillard reaction (Figure R3).

Figure R1 (Supplementary Fig. 6): Amount of myosin heavy chain (MHC) expressed in the undifferentiated myoblasts in each scaffold. The expression of the myosin heavy chain in the undifferentiated samples was compared ($n = 3$ independent experiments, two-tailed t-test). Error bars represent mean \pm s.d. Source data are provided a Source Data file.

Figure R2 (Supplementary Fig. 9): Change in the stiffness of the undifferentiated samples before and after cooking at 60 °C. The stiffness of the undifferentiated group was measured by frequency sweep at the angular frequency range from 0.1 to 1 (rad/s) ($n = 3$ independent experiments, two-tailed t-test). Error bars represent mean \pm s.d. Source data are provided a Source Data file.

Figure R3 (Supplementary Fig. 10): Flavor analysis of the undifferentiated myoblast samples performed using GC-MS. Peak area ratio of the flavor detected in the undifferentiated samples are shown ($n = 3$ independent experiments, two-tailed t-test). Error bars represent mean \pm s.d. Source data are provided a Source Data file.

In addition, the lipid droplet formation (Figure R4) and fatty flavor intensity (Figure R5) of the undifferentiated groups were also evaluated. The results confirmed that there was no significant difference in the lipid droplet formation and fatty flavor intensity of the LALC and HALC groups.

Figure R4 (Supplementary Fig. 12): Adipogenic differentiation degrees of the undifferentiated samples. Quantification of the Oil Red O staining of the undifferentiated adMSC cultured on each scaffold was performed on the proliferation day 3 ($n = 3$ independent experiments, two-tailed t-test). Error bars represent mean \pm s.d. Source data are provided a Source Data file.

Figure R5 (Supplementary Fig. 16): Flavor analysis of the undifferentiated groups. Fatty flavors detected in the undifferentiated groups were normalized to that of bare scaffolds ($n = 3$

independent experiments, two-tailed t-test). Error bars represent mean \pm s.d. Source data are provided a Source Data file.

Furthermore, the nutrition values (Figure R6), flavors (Figure R7), and texture (Figure R8) of the bare assembled scaffold (Bare ACM), ACM with undifferentiated cells (Undifferentiated ACM), and ACM with differentiated cells (ACM) were compared. The results confirmed the significant enrichment of all the organoleptic factors (nutrition, flavor, and texture) in the ACM compared to the bare ACM, whereas there was no difference in the organoleptic factors of the bare ACM and that of the undifferentiated ACM.

Figure R6 (Supplementary Fig. 17): Nutrition factors of the ACM, bare ACM, and undifferentiated ACM. Nutritional analysis of carbohydrates, fats, and proteins was conducted for the three experimental groups ($n = 3$ independent experiments, two-tailed t-test). Fats include saturated and unsaturated fats. Error bars represent mean \pm s.d. Source data are provided a Source Data file.

Figure R7 (Supplementary Fig. 19): Comparison of the flavors detected in each group. Flavors were detected in ACM, bare ACM (assembled scaffolds without cells), and undifferentiated ACM (assembled scaffolds with undifferentiated cells) by GC-MS. Significance was indicated as: a: non-significant (N.S) to bare ACM, b: $0.01 < p\text{-value} < 0.05$ to bare ACM, c: $0.001 < p\text{-value} < 0.01$ to bare ACM, d: $0.0001 < p\text{-value} < 0.001$ to bare ACM, e: N.S to undifferentiated ACM, f: $0.01 < p\text{-value} < 0.05$ to undifferentiated differentiated ACM, g: $0.001 < p\text{-value} < 0.01$ to undifferentiated ACM, h: $0.0001 < p\text{-value} < 0.001$ to undifferentiated ACM ($n = 3$ independent experiments, two-tailed t-test). Error bars represent mean \pm s.d. Source data are provided a Source Data file.

Figure R8 (Supplementary Fig. 20): Results of the texture profile analysis of each group. Factors that determine the texture (chewiness, hardness, cohesiveness, springiness) were measured for ACM, bare ACM, and undifferentiated ACM after grilling the samples at $180\text{ }^{\circ}\text{C}$ to confirm the effect of the cell differentiation quality control on the cultured meat texture ($n = 3$ independent experiments, two-tailed t-test). Error bars represent mean \pm s.d. Source data are provided a Source Data file.

Furthermore, we changed the bar graphs to bars with overlapping dots to show the mean and variance of our data. We highlighted the revised text and figures in the revised manuscript. Also, the part that has been corrected in response to the reviewer's comment is highlighted in blue.

We sincerely appreciate the reviewer's comments again. By carefully reflecting the reviewer's comments, the manuscript has been a more complete and comprehensive manuscript. We corrected our errors and carefully revised our manuscript by responding to the reviewer's valuable comment point by point as presented below.

Major comments:

Comment 1. Line 147-149: “The size of the pores decreases as the crosslinking degree and alginate concentration increase, and there are almost no micropores in the HAHC.” The line is slightly confusing as contrasting points are brought up in the same line. Could the authors clarify if it meant no macropores or micropores in HAHC since the earlier part of the sentence implies that there will be fewer macropores and more micropores in HAHC?

Response: We agree with the reviewer’s comment that the sentence in line 147-149 could be confusing to the readers. As shown in Fig. 2d, micropores and macropores are observed in the scaffolds with lower crosslinking degree, such as LALC and HALC. However, with an increase in the crosslinking degree, the number of micropores in LAHC and HAHC was lesser than that in LALC and HALC. Particularly, almost no micropores was observed in the SEM image of HAHC. In Fig. 2d, we wanted to address the effect of the crosslinked alginate content on the porosity of the scaffolds, which eventually affects the scaffold stiffness. However, as the reviewer pointed out, the description does not sufficiently address our meaning. Thus, we modified the sentence with clearer expressions and words to explain our meaning. The corrected sentence is below:

In the revised manuscript (Lines 146 - 150): The scanning electron microscopy (SEM) images revealed the presence of numerous **micro scale (<100 μm) pores** in the scaffolds (Fig. 2d). **These pores disappeared as the content of the crosslinked alginate increased, and there were almost no micro scale pores in the HAHC.** This result implies that the density of the alginate network increased with an increase in the content of the crosslinked alginate, thus increasing the stiffness of the scaffold.

Comment 2. Line 387: the authors claimed that this assembled cultured meat (ACM) is scalable because it can be scaled into the cm size. However, this does not mean that this ACM can be scaled up for mass production. Scalability is one of the biggest limitations of some proposed methods for manufacturing cultured meat. As the proposed technique still involves a physical assembly of the muscle and fat blocks with microbial transglutaminase, there could be some potential concerns in the large-scale manufacturing setting as well. Please clarify this point or discuss the future plan on how to mass produce ACM.

Response: We appreciate the reviewer for pointing this out. As the reviewer pointed out, we used the word “scalable” to express that the size of our ACM can be controlled using the assembly method, rather than suggesting the mass production of ACM. The cultured meat can be fabricated with various shapes, sizes, and thicknesses through an easy and fast assembly method in which crosslinking agent, microbial transglutaminase (mTG), was applied to the contact surface of the block to be assembled. Upon reading the reviewer’s comments, we realized that our explanation could be misleading. We are very sorry for the confusion. We have revised the manuscript and replaced the sentence with clearer explanation (Lines 409 - 412).

In the revised manuscript (Lines 409 - 412): Thus, we confirmed that different muscle to fat ratios can be implemented for mimicking each meat part through a simple assembly of the meat blocks. Furthermore, we produced the cultured meat with different sizes and shapes, confirming the potential of our strategy to mimic the structural and sensory characteristics of various cuts of slaughtered meat.

In addition, we thought about the mass production of ACM as we read through the reviewer’s

comment. We agree with the reviewer's comment that scalability is one of the biggest limitations of cultured meat. For large-scale manufacturing, the production step should be simple and cost-effective. Our scaffold was prepared using fish gelatin and alginate, which are cheap and easy to obtain. In addition, the scaffolds were fabricated by simply mixing the biopolymer solution and crosslinkers. Therefore, we believe that the scaffold fabrication process can be automated for large-scale manufacturing. However, we agree with the reviewer that the physical assembly method could have some potential concerns in large-scale manufacturing setting. Because mTG crosslinks the gelatin rapidly, it can be difficult to handle the assembly of the blocks in large scale. However, as the activation of mTG depends on the reaction temperature, we believe that the large-scale assembly can be performed in a precise temperature control facility.

From the perspective of mass production of cultured meat, our strategy is not a perfect solution. However, we believe that our study focusing on the relationship between various sensory characteristics of cultured meat and cell differentiation is fundamental research that can be applied to the mass production of cultured meat as a base technology in the future.

Comment 3. Why did the authors choose the 5th day and 8th day for the myogenic differentiation analysis study, despite the study leading to producing lab-grown meat from scratch of cells? Please clarify this point.

Response: We thank the reviewer for the comment. For myoblast, we performed MHC immunostaining on the 5th and 8th day of differentiation induction for two reasons: 1. To show the progress of the myogenic differentiation, 2. To compare the differentiation rates of LALC and HALC. In HALC, the myoblasts started to form myotube-like shape from the 5th day differentiation, as shown in Figure R9.

Figure R9. Optical microscopic images of the bovine myoblasts on the HALC scaffold after differentiation induction. Image taken on the 4th day of the myogenic differentiation is shown on the left and the image on the 5th day of the myogenic differentiation is shown on the right. Scale bars: 100 µm.

In Figure R10a, the myoblasts on LALC do not differentiate into a myotube on day 5. However, they undergo myogenic differentiation stage on day 8, but much less than that of HALC. These results suggest that myogenic differentiation occurs on both LALC and HALC scaffolds, but the degree of differentiation differed significantly depending on the scaffold stiffness. To emphasize this different degree of myogenic differentiation, we showed the MHC

immunostained results of day 5 and day 8. However, all the subsequent experiments were conducted on the 8th day of myogenic differentiation.

Figure R10 (Fig. 3): Optimization of the muscle block. **a** Immunostained myosin heavy chains (MHC) on differentiation day 5 and day 8. The nuclei stained with DAPI (blue), and the MHCs stained with MF20 (red). Scale bars: 20 μm (first row) and 100 μm (second row). **b (i)** Volcano plot showing the up-regulated (red dots) and down-regulated (blue dots) proteins in the HALC ($n = 3$ independent experiments, two-tailed t-test). Fold change indicates the intensity ratio of the protein expression in the HALC to that in the LALC. Proteins expressed by more than 1.5 folds in the HALC were sorted in the heatmap with the intensity of the protein expression. **(ii)** Gene annotation of the proteins in the heatmap. **c** Quantification of the MHC of differentiated bovine myoblasts in each group ($n = 3$ independent experiments, two-tailed t-test). **d** BCA assay results showing the total protein amount of the cells per scaffold ($n = 3$ independent experiments, two-tailed t-test). **e (i)** Stiffness of the raw (ungrilled) groups and **(ii)** griled groups measured using a rheometer ($n = 3$ independent experiments, two-tailed t-test). Illustration of the stiffness change in the HALC with cells owing to an increase in the temperature from 37 $^{\circ}\text{C}$ (ii-a) to 60 $^{\circ}\text{C}$ (ii-b). N.S. means “non-significant”. **f** Peak area ratio of the flavors detected from each sample (left) ($n = 3$ independent experiments, two-tailed t-test). Illustration of the flavor enrichment in the cooked HALC group (right). Error bars represent mean \pm s.d. Source data are provided as a Source Data file.

Comment 4. Why is MHC the only protein of interest in studying myogenesis in LALC and HALC? Preferably should take more than one protein (Myosin with Myogenin or Myosin with Desmin)

Response: We appreciate the reviewer for pointing this out. As pointed out by the reviewer, there are several immunomarkers that can be used to investigate the myogenic differentiation stage. Therefore, we considered various myogenic differentiation markers, including myogenin, desmin, myoD, and myosin heavy chain (MHC) before the immunostaining analysis. However, after studying the myogenic markers from previous research¹⁻³, we concluded that MHC is the most appropriate marker for investigating the mature myogenic differentiation of the cells. According to previous research, desmin is known to be expressed in undifferentiated myosatellite cells, so we excluded desmin. In addition, we found out that MyoD and myogenin are the markers expressed at the myocyte stage or at the beginning of the myocyte fusion stage. In our study, we wanted to compare the mature stage of myogenic differentiation of the cells depending on the scaffold stiffness. As MHC is the marker which is expressed only at the terminated stage of the myogenic differentiation (myotube stage), we selected MHC as a mature myogenic differentiation indicator in our study. In addition, the expression of MHC indicates that the cells were differentiated into mature myotubes after the expressions of myogenin and desmin, so we concluded that MHC is sufficient for the myogenic differentiation analysis.

Comment 5. Time points of cell culture characterisations were unclear in the adipogenic differentiation study. When was oil red staining analysis performed to confirm the adipogenic differentiation? Please add this information

Response: We are very sorry that the information on the adipogenic differentiation study was unclear in Fig. 4a. We performed the oil red staining analysis on the 4th day of differentiation. Particularly, bovine adipose-derived mesenchymal stem cells (adMSCs) were seeded on the scaffold and proliferated for 3 days. Subsequently, we replaced the growth medium with the adipogenic differentiation medium on the 3rd day of proliferation. Then the differentiation progressed for another 4 days, and we conducted the fixation process on the last day of differentiation before performing the Oil red O staining analysis.

We added this information in the revised manuscript as shown below (Lines 292 – 294):

In the revised manuscript (Lines 292 - 294): After three days of proliferation, the culture medium was changed to initiate the adipogenic differentiation, and the differentiation progressed for another 4 days.

Comment 6. Why did the authors choose the 12 days-conditioned medium for the degradation study? It would be more informative if the degradation study graph plots weight loss against the number of days.

Response: We thank the reviewer for this comment. We performed the scaffold degradation test for 12 days to test whether the scaffolds can stably support cells during the entire cell culture period. For myogenic differentiation, cells proliferate for 4 days, after which they differentiate for another 8 days. Therefore, myoblasts are cultured for a total of 12 days. For adipogenic differentiation, cells were cultured for 7 days in total (3 days of proliferation and another 4 days of differentiation). The degradation result revealed that our scaffold can remain in the conditioned medium for the entire cell culture period, suggesting that it can support the cells until full differentiation. The conditioned medium contains various enzymes and proteins secreted from the cells, which mimics the cell culture condition. During the cell culture period, the cells secrete the enzymes that degrade the substrate when they proliferate. Therefore, we used the conditioned medium to mimic the degradation behavior of the scaffold in the culture condition. As the main purpose of the degradation test was to confirm the stability of the scaffold for the entire culture period rather than to specifically analyze the degradation behavior, we calculated the weight change of the scaffold on the last day of the culture period.

Comment 7. For weight loss calculation (in the supplementary file), should the equation be “Initial weight – the weight of hydrogel at day 12”? In the current equation format, this implies that the weight of the scaffold increased after 12 days which is not a degradation. Please recheck.

Response: We are very sorry that our information was incorrectly written in the supporting data. As pointed out by the reviewer, the equation should be “Initial weight – the weight of the hydrogel on day 12”. Our data was calculated using the correct equation, but the equation was incorrectly written in the unrevised manuscript. We thank the reviewer for this comment. We have rechecked and corrected our errors. We corrected the equation and included it in the method section in lines 524 – 525 in the revised manuscript.

In the revised manuscript (Lines 524 - 525):

Weight loss (%)

$$= \frac{\text{Initial weight of the hydrogel} - \text{Weight of the hydrogel on day 12}}{\text{Initial weight of the hydrogel}}$$

Comment 8. The way authors show the relative proliferation in the supplementary file (Figures S3 and S7) is misleading. The normalised graph made it look like the cell proliferates faster/higher in LALC by 200% compared to the standard culture dish, which might not be the case. Is the relationship between CCK-8 and cell growth a linear relationship? Can authors provide a standard curve for that?

Response: We thank the reviewer for this comment. We compared the relative cell viability of the groups using the O.D value obtained from the CCK-8 results to demonstrate that our scaffolds exhibited no cytotoxicity. However, as the reviewer pointed out, the CCK-8 assay data would be misleading to the readers. The proliferation rate was simply obtained by comparing the average O.D values on day 2 and day 4, so we could not judge the significance of the data. For a clearer information, we reperformed the viability assay, and investigated the accurate cell numbers rather than just showing the O.D value. At this point, we obtained the O.D value of each sample and subtracted the O.D value of bare sample to obtain reliable data. For the culture well sample, we subtracted the O.D value of the culture well group from the O.D value of the blank. At the same time, we drew a standard curve to relate the cell numbers and O.D value of the CCK-8 assay results. Finally, we could accurately compare the cell viability of each group (Figure R11). The cell viabilities of HALC and LALC were lower than that of the culture well group on day 2, but became similar on day 4, confirming the non-cytotoxicity of the two scaffolds. We included this data as Supplementary Fig. 3 in the supporting material.

Figure R11 (Supplementary Fig. 3): Evaluation of the cell proliferation of the bovine myoblasts. Viabilities of the bovine myoblasts of the culture well, HALC, and LALC group were compared on day 2 and day 4 ($n = 3$ independent experiments, two-tailed t-test). Error bars represent mean \pm s.d. Source data are provided a Source Data file.

Additionally, we performed the same procedure for the adMSCs, and confirmed that the scaffold had no effect on the cell viability (Figure R12).

Figure R12 (Supplementary Fig. 11): Evaluation of cell viability of the bovine adMSCs. Viability of the bovine adMSCs in each group was analyzed using CCK-8 assay. The cell numbers on proliferation day 2 and day 3 were compared ($n = 3$ independent experiments, two-tailed t-test). Error bars represent mean \pm s.d. Source data are provided a Source Data file.

The results confirmed that there was no significant difference in the cell viability of LALC and HALC over the proliferation period, indicating that factors other than the cell differentiation can be excluded.

We thank the reviewer again for this comment. By responding to the reviewer's comment, we were able to obtain clearer results.

Minor comments:

Comment 9. In Figure 2, what did the white arrow point to? Please explain it in the figure description

Response: We appreciate the reviewer for pointing this out. We could recheck the insufficient description of our figure. In Fig. 2d, the white arrows were drawn to indicate the pores in our scaffolds. However, we realized that the white arrows also indicate some cracks of the scaffolds, which can be confusing to the readers. Therefore, we have redrawn the arrows to correctly indicate the micro-scale pores of the scaffold (Figure R13d). We have added this information in the figure description as follows:

Figure R13 (Fig. 2): Characterization of the gelatin/alginate scaffold. **a** Illustration of the change in the alginate concentration and crosslinking degree of the alginate network of the scaffolds. **b** (i) Raman spectra of the LALC, LAHC, HALC, and HAHC, (ii) Percentage of the calcium-crosslinked alginate structures calculated from the Raman shifted peak area between 1390 and 1420 cm^{-1} ($n = 3$ independent experiments, two-tailed t-test). **c** (i) Stress–strain curves obtained during the compression test of the scaffolds, (ii) Young’s modulus of each scaffold at a compressive strain of 0–50 % ($n = 3$ independent experiments, two-tailed t-test). **d** Scanning electron microscopy (SEM) images showing the porous structures of the scaffolds with **white arrows indicating the micro-size pores**. Scale bars: 500 μm in the first row and 100 μm in the second row. Error bars represent mean \pm s.d. Source data are provided as a Source Data file.

Comment 10. Line 30: “different contents and forms of muscle tissues and adipose tissues” is a little unclear and awkward. Authors can consider paraphrasing it.

Response: We thank the reviewer for pointing this out. We agree that the sentence is unclear and does not convey the meaning. We wanted to explain that the composition of muscle and adipose tissues in slaughtered meat varies depending on the slaughtered parts, and these compositions can affect the organoleptic properties of meat. However, as we have revised the abstract to clarify the novelty of our work, the sentence in Line 30 was removed. The revised abstract is shown below:

In the revised manuscript (Lines 25 - 35): Research on cultured meat has primarily focused on the mass proliferation or differentiation of muscle cells; thus, the food characteristics of cultured meat remain relatively underexplored. As the quality of meat is determined by its organoleptic properties, cultured meat with similar sensory characteristics to slaughter meat is highly desirable. In this study, we develop a gelatin/alginate scaffold with different stiffness to regulate the differentiation of muscle and fat cells. Thereafter, we investigate the effect of the cell differentiation quality on the sensory properties of cultured meat. Lastly, we develop the assembled cultured meat that embodies the food properties of slaughtered beef by stacking the muscle and fat parts with highly differentiated cells. In this study, we controlled the quality of the muscle and fat parts, which affects the sensory properties of the final cultured meat, by regulating the degree of cellular differentiation via scaffold engineering.

Comment 11. Line 32: “these structural characteristics are highly desirable” What desirable structural characteristics are the authors referring to?

Response: We are very sorry for the ambiguity of the explanation in Line 32. We tried to explain that the cultured meat composed of highly differentiated muscle and fat cells is highly desirable for mimicking the organoleptic properties of slaughtered meat. We replaced this sentence with clearer explanation in the revised manuscript.

In the revised manuscript (Lines 27 - 28): As the quality of meat is determined by its organoleptic properties, cultured meat with similar sensory characteristics to slaughter meat is highly desirable.

Comment 12. Line 56: What defines/determines the “quality of each part” and what constitutes “food characteristics of the meat”?

Response: We thank the reviewer for this comment. The quality of each part is defined by the muscle fiber dimensions and the lipid contents. The food characteristics of meat mainly include its flavor, texture, and nutrition. We have revised the sentence to reduce the ambiguity of the description and to emphasize the novelty of our work.

In the revised manuscript (Lines 48 - 51): For example, the biological characteristics of muscle and fat tissues, such as myofiber dimensions and lipid content, determine the organoleptic properties of slaughtered meat, and these tissues characteristics can be affected by the differentiation of cells.

Comment 13. Line 63: The phrase “to implement the structure of meat” is a little unclear and awkward. Did the authors mean “to imitate” instead?

Response: As pointed out by the reviewer, we meant “to imitate” when the word “to implement” was used. To reduce the awkwardness of the sentence, we replaced the word “implement” with “mimic” as shown below:

In the revised manuscript (Line 66 - 67): To **mimic** the structure of meat, both adipogenic and myogenic differentiation should occur significantly in one mass of the cultured meat.

Comment 14. Line 217: The use of “destruct” in this sentence is a little awkward. The authors can consider rephrasing it.

Response: We thank the reviewer for pointing this out. We agree with the reviewer’s comment that the word “destruct” is inappropriate in the sentence in Line 217. We meant that the unfolding of the triple helices of collagen occurs as the collagen fibrils denature under high temperature. To improve the specificity of the meaning, we corrected the sentence as follows:

In the revised manuscript (Lines 240 - 242): The triple helices of the collagens can **unfold** as the temperature increased to 60 °C, thus decreasing the Young’s modulus of the collagen fibrils.

Comment 15. Line 276-277: The authors may want to work on refining the captions for the subfigures in figure 4, especially for Figure 4a(ii) and (iii), as it is currently a little confusing.

Response: We agree with the reviewer’s comment. Data in Figure 4a are divided into too many subfigures which can confuse the readers, particularly Figure 4a(ii) and (iii). As Figure 4a(ii) shows the quantitative result based on the results in Figure 4a(iii), we combined Figure 4a(ii) and Figure 4a(iii) into one subfigure, Figure 4a(ii). Then we added black dotted lines to connect the results together to show that images of the Oil Red O staining analysis and the bar graph data belong to one subfigure (Figure R14).

Figure R14 (Fig. 4): Optimization of the fat block. a (i) Illustration of the adipogenic differentiation of the adipose tissue-derived mesenchymal stem cells (adMSCs) depending on the scaffold, and **(ii)** the quantification of the adipogenic differentiation degree of each group based on the lipid droplet staining performed using Oil Red O staining ($n = 3$ independent experiments, two-tailed t-test). Scale bars: 50 μm (first row) and 20 μm (second row). **b (i)**

Confocal images showing the lipid droplet formation analyzed by immunostaining. Scale bars: 50 μm (first and second column) and 20 μm (third column). (ii) The quantitative assessment of the lipid droplet amount is shown in the graph based on the confocal images using Image J software ($n = 3$ independent experiments, two-tailed t-test). **c** Peak area ratios of nonanal and 2-ethyl-1-hexanol are summed and normalized to that of the bare scaffolds ($n = 3$ independent experiments, two-tailed t-test). Error bars represent mean \pm s.d. Source data are provided as a Source Data file.

Comment 16. Line: 462 - 464. “The cells were seeded 12 well moulded on the scaffold”. Assuming the cells were seeded on the top layer of the scaffold. How do the authors ensure that the cells penetrate into the scaffold? Have the authors done H&E staining or Z-Stack confocal imaging?

Response: As the reviewer pointed out, cells are rather embedded in the surface of the scaffold than penetrating the scaffold. However, the main strategy in this study is to investigate the effect of cell differentiation control on the food properties of the final assembled cultured meat. Our scaffolds provide cells with 2D surfaces with different stiffnesses. Although our scaffold can be restructured to a 3D structure using the lyophilization method, but scaffolds with different stiffness will exhibit different pore sizes. In 2D hydrogels, the difference in porosity only affects the mechanical properties of the scaffolds. However, in 3D scaffolds, the pore size determines the area where cells can migrate and occupy, as well as the nutrient supply and gas diffusion to the cells. These factors can result in different cell viability and an uneven distribution of cells in the scaffold⁴, preventing the accurate comparison of the experimental groups. To rule out factors other than the effect of the scaffold stiffness on the cell differentiation, we used 2D scaffold type in this study.

To reflect the reviewer’s comment, we conducted both H&E staining and Z-stack confocal imaging (Figure R15). We apologize for the bad image quality of the H&E staining results. Some parts of the scaffolds were destroyed upon decalcification as the scaffolds contain calcium ions as crosslinkers. However, the presence of cells on the surface of the scaffolds was confirmed from the H&E staining results. To supplement the results, we obtained the z-stack confocal images of each group and confirmed the presence of the immunostained cells on the surface of the scaffold.

Figure R15: H&E staining and z-stack confocal images of the myoblast differentiated on HALC (a) and adMSC differentiated on LALC (b). Black arrows indicate the cells cultured on the scaffold.

Moreover, the surface of 3D porous scaffolds would eventually be perceived as 2D by cells owing to the difference in the scale of the cells and scaffolds. As our work is more of a fundamental study suggesting that cell differentiation behavior affects the sensory factors of cultured meat, we believe that our findings can also be applied to various 3D cultured meat scaffolds in the future.

Comment 17. Line: 501 - 502. “For the protein analysis, the cells were detached from the scaffolds by treating with EDTA”. How do the authors collect all the cells from the scaffold? Please elaborate more on this point.

Response: We apologize for the insufficient information. For the protein analysis in Figure 3f, we collected the cells by detaching the cells with 0.025 % trypsin EDTA. First, we washed the cells with 1X PBS, after which we treated the cells with 0.025 % trypsin EDTA to detach the cells from the scaffolds. Subsequently, we added 1X PBS to the detached cell suspension and collected the cells by centrifugation at $264 \times g$ for 4 min. Thereafter, we washed the cells three times with 1X PBS. This process was repeated to collect all the cells from the scaffold until no attached cell can be observed on the scaffold through an optical microscope. Thereafter, we obtained the cell pellet by centrifugation. Next, we added radioimmunoprecipitation assay (RIPA) cell lysis buffer to lysis the cells to extract the expressed proteins. After vortexing and pipetting the cell lysate solution, we stored the solution at 4 °C for 30 min. Thereafter, the solution was centrifuged at 10,000 rpm for 10 min to remove the cell debris, which sink after centrifugation. Finally, we obtained the pure solution of the extracted proteins for BCA assay and LC-MS proteomic analysis. We added this information in the method section in lines 630 – 640 in the revised manuscript.

In revised manuscript (Lines 630 - 640): For the protein analysis, the cells were detached from the scaffolds by treating them with 0.025% trypsin EDTA. The detached cells were centrifuged at $264 \times g$ for 4 min, followed by the aspiration of the filtrate. To wash out the residual medium, the cells were resuspended in 1X PBS and centrifuged. This process was repeated to collect all the cells from the scaffold until there was no attached cells on the scaffold

when observed through an optical microscope. The washing procedure was repeated three times. Finally, a cell pellet was prepared. Thereafter, radioimmunoprecipitation assay (RIPA) cell lysis buffer was added to the cell pellet to lysis the cells to extract the proteins. After vortexing and pipetting the cell lysate solution, the solution was stored at 4 °C for 30 min. Subsequently, the solution was centrifuged at 10,000 rpm for 10 min to remove the cell debris, which sink after centrifugation. Finally, we obtained the pure solution of the extracted proteins for the BCA assay and LC-MS proteomic analysis.

References:

1. Posont, Robert J. "The Role of Inflammatory Pathways in Development, Growth, and Metabolism of Skeletal Muscle in IUGR Offspring; Blood Gene Expression of Inflammatory Factors as Novel Biomarkers for Assessing Stress and Wellbeing in Exotic Species." (2019).
2. Iberite, Federica, Emanuele Gruppioni, and Leonardo Ricotti. "Skeletal muscle differentiation of human iPSCs meets bioengineering strategies: Perspectives and challenges." *NPJ Regenerative Medicine* 7.1 (2022): 23.
3. Dumont, Nicolas A., Yu Xin Wang, & Michael A. Rudnicki. "Intrinsic and extrinsic mechanisms regulating satellite cell function." *Development* 142.9 (2015): 1572-1581.
4. Yu, et al. "Effect of pore size on cell behavior using melt electrowritten scaffolds." *Front. Bioeng. Biotechnol.* **9** (2021): 629270.

Response to Reviewer #3

Reviewer(s)'s Comments to Author:

Reviewer 3.

Comments

The manuscript entitled “Block-Assembled Edible Scaffold for Cultured Meat” presents the attempt to fabricate large cultured meat constructs from smaller units of muscle and adipose tissues, where each was cultivated separately on alginate-gelatin based hydrogels and was later combined by applying enzymatic crosslinking with microbial transglutaminase. The differentiation of two bovine derived cell types, myoblast cells and mesenchymal stem cells, was assessed separately on two hydrogels with different stiffness and composition, as well as the resultant characteristics of the constructs, such as stiffness and presence of different volatile compounds. Multi-tissue constructs were then fabricated and analyzed in terms of nutritional, texture and flavor analysis. Horizontal and vertical assembly of muscle and adipose units was then applied to create larger marbled constructs.

While this work emphasizes the need to assess the crosstalk between cell behavior and resultant organoleptic and nutritional construct characteristics for cultured meat purposes, both novelty and quality are far from the required bar. The mentioning of previous works in the literature is extremely limited, and many of the tissue-engineering related concepts presented here were demonstrated before in other works. Both the used scaffolding materials and the technique used for larger structure assembly are not novel. Most of the presented conclusions are not supported by the currently shown DATA, either due to missing statistical analysis information or missing control groups, making further

investigations mandatory. The description of several protocols and analysis are missing as well, preventing the reproduction of the presented research. Some parts of the paper should be reorganized. The comments below are extremely recommended for this article.

Response: We sincerely appreciate been given the opportunity to improve the reliability and quality of our manuscript through the valuable comments from the reviewer. Through the reviewer's comments, we were able to thoroughly review our manuscript once again and revised it to emphasize the novelty of our work. By reflecting on the reviewer's comments, we realized that there are several areas needing improvements in our manuscript. Therefore, we changed some of our data and reorganized both the main and supporting figures after re-performing our experiments and evaluating the significance of the results. In addition, we added the undifferentiated group for all experimental data to rule out factors other than cell differentiation. We also revised the title of this work to include a more specific expression to address the novelty of our work. In the revised manuscript and figures, we changed Figure 2b(ii), Figure 3b-f, Figure 4a-c, Figure 5b(ii)-d, Supplementary Figure 3, Supplementary Figure 11, and Supplementary Table 3. In addition, we added supplementary Figure 4, Supplementary Figure 5, Supplementary Figure 6, Supplementary Figure 7, Supplementary Figure 8, Supplementary Figure 9, Supplementary Figure 10, Supplementary Figure 12, Supplementary Figure 13, Supplementary Figure 14, Supplementary Figure 15, Supplementary Figure 16, Supplementary Figure 17, Supplementary Figure 19, Supplementary Figure 20, Supplementary Table 1, Supplementary Table 2, Supplementary Table 4, Supplementary Table 5, and Supplementary Table 6. The p-values and the number of experiment repeats were added in all the figures and figure descriptions. Furthermore, we changed the bar graphs to bars with overlapping dots to show the mean and variance of our data. We highlighted the revised text

and figures in the revised manuscript. Also, the part that has been corrected in response to the reviewer's comment is highlighted in blue in the point-by-point response.

We sincerely appreciate the reviewer's comments again. By carefully reflecting the reviewer's comments, the manuscript has been a more complete and comprehensive manuscript. We corrected our errors and carefully revised our manuscript by reflecting the reviewer's valuable comment point by point as presented below.

Comments

Comment 1. In general, one of the major limitations in thick-construct fabrication is the ability to support the growth of the cells in terms of sufficient material transfer. The method presented in this paper discusses cell seeding on-top of hydrogels, which are later stacked together by transglutaminase, to produce thicker constructs. However, one must ask: What was the thickness of each hydrogel layer? Were the cells capable of penetrating the hydrogels during cultivation? Were they evenly distributed throughout the hydrogels? The authors must provide 3D reconstructions of cell-seeded hydrogels, and analyze whether cell penetration occurred. This is relevant for both cell types, in the different hydrogel compositions. If the cells could not infiltrate into the hydrogels, then the actual constructs produced in this work are hydrogels with thin cellular coatings – a configuration which is not conceptually beneficial for cultured meat.

Response: We thank the reviewer for this comment. The thickness of both LALC and HALC hydrogels was approximately 0.63 mm (Table R1). We are very sorry for the missing information, and we have added this information in Table S1 in the supporting data.

Hydrogel	Thickness (mm)
LALC	0.630±0.026458
HALC	0.633±0.020817

Table R1 (Supplementary Table 2). Thickness of the scaffolds. The thickness of each group was measured using a digital caliper (n = 3).

We appreciate the reviewer's comment about the cell penetration through the scaffold. As the reviewer pointed out, 3D scaffolds with cells evenly distributed throughout the entire scaffold would be more beneficial for cultured meat owing to the high cell contents. However,

our main strategy in this study is to investigate the effect of cell differentiation control on the food properties of the final assembled cultured meat. Our scaffolds provide cells with 2D surfaces with different stiffnesses. Although our scaffold can be restructured into 3D structure using the lyophilization method but scaffolds with different stiffness will have different pore sizes. In 2D hydrogels, different porosity only affects the mechanical properties of the scaffolds; however, in 3D scaffolds, the pore size determines the area where cells can migrate and occupy, as well as the nutrient supply and gas diffusion to the cells. These factors can result in different cell viability and an uneven distribution of cells in the scaffold¹, preventing the accurate comparison of the experimental groups. To rule out factors other than the effect of the scaffold stiffness on the cell differentiation, we used the 2D scaffold type in this study.

To investigate the cell deposition on the scaffold regarding the reviewer's valuable comment, we performed H&E staining for the muscle and fat blocks (Figure. R1). We apologize for the bad image quality of the H&E staining results. Some parts of the scaffolds were destroyed upon decalcification as the scaffolds contain calcium ions as crosslinkers. Nevertheless, we confirmed the presence of cells on the surface of the scaffolds from the H&E staining results. To supplement the results, we obtained z-stack confocal images of each group and confirmed the presence of immunostained cells on the surface of the scaffold.

Figure R1: H&E staining and z-stack confocal images of myoblast differentiated on HALC (a) and adMSC differentiated on LALC (b). Black arrows indicate the cells cultured on the scaffold.

Moreover, the surface of the 3D porous scaffolds would eventually be perceived as 2D by cells owing to the difference in the scale of the cells and scaffolds. As our work is more of a fundamental study suggesting that cell differentiation behavior affects the sensory factors of cultured meat, we believe that our findings can also be applied to various 3D cultured meat scaffolds in the future.

Comment 2. In the abstract and introduction, please address previous works done in the field and be specific about what is the contribution of your work to the current knowledge. Some of the main concepts shown in your paper were already investigated previously. Moreover, some statements are inaccurate.

- line 26: "...only a small sized cultured meat with negligible similarity with conventional meat...." is not correct. Several works have already created large constructs for cultured meat, some of which included multiple cell-types:

<https://www.nature.com/articles/s41538-021-00090-7>

<https://www.nature.com/articles/s41467-021-25236-9>

<https://www.ncbi.nlm.nih.gov/pmc/articles/PMC9039213/>

<https://onlinelibrary.wiley.com/doi/full/10.1002/advs.202202877>

- lines 46-50: " Recently, researchers have reported strategies to produce cultured meat with various types of scaffolds, such as 3D printed bioink⁶ , textured soy proteins⁷, and spinach⁸, to mimic the structure of meat. However, the complex technology and expensive extracellular matrix (ECM)-derived proteins involved in scaffold production limit the scalability of cultured meat." Please rephrase this.

More than one bioink was assessed. Here are more examples to mention:

<https://www.sciencedirect.com/science/article/abs/pii/S0142961222001260?via=ihub>

<https://pubmed.ncbi.nlm.nih.gov/36271729/>

Other decellularized plant tissues than spinach were also investigated for muscle tissue engineering:

<https://www.biorxiv.org/content/10.1101/2020.02.23.958686v1.abstract>

Moreover, some of these examples show the adherence, growth and differentiation of cells without any use of ECM proteins.

- Lines 67-71: The paper suggests alginate-fish gelatin mixtures to create hydrogels as cultivation platforms for cultured meat. Why were these mixture of materials chosen? This is not elaborated enough. Moreover, such materials aren't novel for muscle or adipose tissue engineering, as they were assessed in previous works, such as:

<https://www.ncbi.nlm.nih.gov/pmc/articles/PMC5744339/>

<https://bmcbiotechnol.biomedcentral.com/articles/10.1186/1472-6750-12-35>

<https://pubmed.ncbi.nlm.nih.gov/22556122/>

<https://journals.sagepub.com/doi/10.1177/0885328216634057>

Additionally : please explain this choice, as animal-derived materials such as fish gelatin are not desired for cultured meat purposes and are unsustainable.

- Lines 60-66: The concept of tailoring hydrogel stiffness by changing composition or crosslinker concentration to promote adipogenesis or myogenesis was demonstrated widely in the literature. Please address previous attempts in your introduction and discussion.

- The paper suggests the assembly of several alginate-gelatin constructs to create larger multi-tissue type constructs, by using protein crosslinking with microbial transglutaminase. Previous works have shown the assembly of smaller scaffolds to achieve larger and complex structures (part of them also used transglutaminase):

<https://www.nature.com/articles/s42003-022-03852-5>

<https://www.nature.com/articles/s41467-021-25236-9>

<https://www.frontiersin.org/articles/10.3389/fbioe.2022.875069/full>

The use of microbial transglutaminase to create gelatin-based constructs was also described previously:

<https://www.ncbi.nlm.nih.gov/pmc/articles/PMC7825108/>

<https://www.nature.com/articles/s41538-019-0054-8>

It is crucial that the authors will address these aspects in the paper, and emphasize the novelty of their work in comparison to the current literature.

Response: We sincerely appreciate the reviewer for this comment. We realize that our abstract and introduction are insufficient to address the novelty of our work. As the reviewer pointed out, there are previous research on developing cultured meat in mm to cm scales. In the introduction, we wanted to explain that we can fabricate cultured meat without limitations in size, shape, or tissue composition, just like conventional meat. However, as the reviewer pointed out, size controlling technique by assembly is not a novelty that stands out in our study compared to other previous research. Rather, the novelty of this research is that for the first time, it suggests the ability to control the flavor, texture, and nutritional value of cultured meat by varying the degree of cell differentiation. Previous studies in cultured meat only focus on the evaluation of the occurrence of cell differentiation and the design of the scaffold shape to resemble meat. However, organoleptic properties such as texture and flavor have not been considered importantly even though they determine the quality of meat. In slaughtered meat, physical and biochemical characteristics of muscle and fat tissues determine the sensorial and nutritional properties of meat. These characteristics of muscle and fat tissues are affected by myogenesis and adipogenesis, respectively. Therefore, we focused on controlling the myogenesis and adipogenesis degree of cells to enrich the organoleptic properties of cultured

meat.

By carefully reflecting on the reviewer's comment, we revised our abstract and introduction in a way that clearly presents the novelty of our study (Lines 25 – 35 and lines 45 – 57).

In the revised manuscript (Lines 25 – 35): Research on cultured meat has primarily focused on the mass proliferation or differentiation of muscle cells; thus, the food characteristics of cultured meat remain relatively underexplored. As the quality of meat is determined by its organoleptic properties, cultured meat with similar sensory characteristics to slaughter meat is highly desirable. In this study, we develop a gelatin/alginate scaffold with different stiffness to regulate the differentiation of muscle and fat cells. Thereafter, we investigate the effect of the cell differentiation quality on the sensory properties of cultured meat. Lastly, we develop the assembled cultured meat that embodies the food properties of slaughtered beef by stacking the muscle and fat parts with highly differentiated cells. In this study, we controlled the quality of the muscle and fat parts, which affects the sensory properties of the final cultured meat, by regulating the degree of cellular differentiation via scaffold engineering.

In the revised manuscript (Lines 45 – 57): However, research on cultured meat scaffolds has focused on the evaluation of the occurrence of cell differentiation and the design of the scaffold shape to resemble meat. Consequently, factors that determine the various food characteristics of meat are generally overlooked. For example, the biological characteristics of muscle and fat tissues, such as myofiber dimensions and lipid content, determine the organoleptic properties of slaughtered meat, and these tissue characteristics can be affected by the differentiation of cells. Furthermore, meat exhibits a marbled structure, in which muscle and fat tissues are randomly distributed, and the content ratio of these tissues varies even within a single piece of

meat (Fig. 1a). This content variation results in differences in the textures, tastes, and nutritional values of different meat parts. Therefore, it is crucial to develop cultured meat that mimics the marbled structure of livestock meat, which is composed of highly differentiated muscle and fat tissues, to achieve the natural sensory characteristics of meat.

We also addressed previous research on regulating cell differentiation by controlling the scaffold stiffness² in our revised introduction (Lines 58 - 64).

In the revised manuscript (Lines 58 - 64): The mechanical properties of a scaffold can regulate cellular functions, and the myogenic and adipogenic differentiation rates can be regulated by the stiffness of the scaffold. Particularly, Young's moduli of ~11 and ~3 kPa are required to stimulate myogenesis and adipogenesis, respectively. These mechanical properties can be achieved by controlling the crosslinking degree of polymer networks in the scaffold. For example, the stiffness of an alginate hydrogel depends on the crosslinking density of alginate, which affects the adipogenesis of mesenchymal stem cells.

In addition, we have included explanation on why fish gelatin and alginate were used in the introduction and discussion. To fabricate the hydrogel, gelatin was selected to provide cells a stable environment for the cell differentiation. Because gelatin is derived from collagen, an extracellular matrix (ECM) component, it exhibits an inherently high cytoaffinity. Although there are recent studies reporting plant-derived scaffolds for cultured meat fabrication, few studies have confirmed the ability of plant-derived materials to provide the stable cell adhesion and cell differentiation environment as much as animal-derived biopolymers. As our study focuses on the relationship between the different cell differentiation behavior and organoleptic

properties of cultured meat, a scaffold composed of biopolymers that can stably provide cell differentiation environment was required. In addition, considering recent studies using animal-derived materials such as collagen for cultured meat scaffold³, we thought more research is needed on non-animal derived scaffolds with high cytoaffinity to completely replace the animal-derived materials for cultured meat. However, we believe that our strategy can also be extended to various types of scaffolds including non-animal derived scaffolds in the future.

In addition, alginate was used as a stiffness-determining polymer because of the effect of the ion-crosslinking degree of alginate on the hydrogel stiffness. We included this explanation in the revised introduction and discussion (Lines 421 - 435):

In the revised manuscript (Lines 421 - 435): The organoleptic properties of the cultured meat were controlled by regulating the differentiation degree of each cell type with different scaffold stiffness. The scaffold was fabricated using fish gelatin and alginate. Gelatin is a common biopolymer used in biomaterials, and as it is derived from collagen, an extracellular matrix (ECM) component, it exhibits an inherently high cytoaffinity. Although there are recent studies reporting plant-derived scaffolds for cultured meat fabrication, only few studies have clearly confirmed whether plant-derived materials can provide the stable cell adhesion and cell differentiation environment as much as animal-derived biopolymers. As our study focused on the relationship between the different cell differentiation behavior and the organoleptic properties of cultured meat, a scaffold composed of biopolymers that can provide a stable cell differentiation environment is required. Therefore, gelatin was used in this study, but we expect that our strategy can be extended to various plant-derived scaffolds in the future.

In addition, alginate was used as a stiffness determining polymer. The mechanical properties of the scaffold were optimized for the adipogenic and myogenic differentiations by controlling

the crosslinking degree of alginate.

Comment 3. Statistical analysis to determine the significance between different groups is often missing or not properly described. Many graphs are shown without presenting the distribution of individual values around a mean value, some bars are with no STD or SEM markings, in the legends there is often no mention of number of repeats (n=?). Moreover, if there is no significance, please add “ns” in the graphs. Although the authors make various conclusions throughout the paper, one cannot rely on those without being given the supporting statistical analysis in a clear and full manner. Address the following:

Figure 2 bii, cii

Figure 2S,

Figure 3 b, d-f

Figure 3S,

Figure 4 c-d,

Figure S7,

Figure 5.

Response: We appreciate the reviewer’s comment. We agree with the reviewer that the significances of the data throughout the manuscript are missing. To improve the reliability of our study, we have indicated the significance of the data in the figures mentioned by the reviewer through repeated experiments. We have replaced all our bar graphs with bars with overlapping dots to show the mean and variance of the data. In addition, we added the number of repeated experiments in the figure descriptions. The corrected data and description are highlighted in blue.

The p-values have been added in Figure R2b(ii) and 2c(ii). In addition, the number of repeats was added in the figure description.

Figure R2 (Fig. 2): Characterization of the gelatin/alginate scaffold. **a** Illustration of the change in the alginate concentration and crosslinking degree of the alginate network of the scaffolds. **b** (i) Raman spectra of the LALC, LAHC, HALC, and HAHC, (ii) Percentage of the calcium-crosslinked alginate structures calculated from the Raman shifted peak area between 1390 and 1420 cm^{-1} ($n = 3$ independent experiments, two-tailed t-test). **c** (i) Stress–strain curves obtained during the compression test of the scaffolds, (ii) Young’s modulus of each scaffold at a compressive strain of 0–50 % ($n = 3$ independent experiments, two-tailed t-test). **d** Scanning electron microscopy (SEM) images showing the porous structures of the scaffolds with white arrows indicating the micro-size pores. Scale bars: 500 μm in the first row and 100 μm in the second row. Error bars represent mean \pm s.d. Source data are provided as a Source Data file.

The significance and number of experiments have been indicated in Figure R3 (Supplementary Fig. 2). In addition, the equation of weight loss was corrected and included in the method section in lines 524 - 525.

Figure R3 (Supplementary Fig. 2): Degradation rates of the LALC and HALC. The degradation degree was quantified using the weight loss of the hydrogels ($n = 3$ independent experiments, two-tailed t-test). N.S indicates “non-significant”. Error bars represent mean \pm s.d. Source data are provided a Source Data file.

In the revised manuscript (Lines 524 - 525):

$$\begin{aligned} & \text{Weight loss (\%)} \\ &= \frac{\text{Initial weight of the hydrogel} - \text{Weight of the hydrogel on day 12}}{\text{Initial weight of the hydrogel}} \end{aligned}$$

Fig. 3b, d-f have been reorganized, and the number of experiments and significance of the proteomic analysis, MHC and total protein analysis, stiffness analysis, and flavor analysis have been indicated, as shown below in Figure R4.

Figure R4 (Fig. 3): Optimization of the muscle block. **a** Immunostained myosin heavy chains (MHC) on differentiation day 5 and day 8. The nuclei stained with DAPI (blue), and the MHCs stained with MF20 (red). Scale bars: 20 μm (first row) and 100 μm (second row). **b** (i) Volcano plot showing the up-regulated (red dots) and down-regulated (blue dots) proteins in the HALC ($n = 3$ independent experiments, two-tailed t -test). Fold change indicates the intensity ratio of the protein expression in the HALC to that in the LALC. Proteins expressed by more than 1.5 folds in the HALC were sorted in the heatmap with the intensity of the protein expression. (ii) Gene annotation of the proteins in the heatmap. **c** Quantification of the MHC of differentiated bovine myoblasts in each group ($n = 3$ independent experiments, two-tailed t -test). **d** BCA assay results showing the total protein amount of the cells per scaffold ($n = 3$ independent experiments, two-tailed t -test). **e** Stiffness of the raw (ungrilled) groups (i) and grilled groups (ii) measured using a rheometer ($n = 3$ independent experiments, two-tailed t -test). Illustration of the stiffness change in the HALC with cells owing to an increase in the temperature from 37 $^{\circ}\text{C}$ (ii-a) to 60 $^{\circ}\text{C}$ (ii-b). N.S means “non-significant”. **f** Peak area ratio of the flavors detected from each sample (left) ($n = 3$ independent experiments, two-tailed t -

test). Illustration of the flavor enrichment in the cooked HALC group (right). Error bars represent mean \pm s.d. Source data are provided as a Source Data file.

We re-performed the experiments on the cell viability to obtain exact cell numbers rather than O.D values (Figure R5). We included the p-values and number of repeats in the figure description.

Figure R5 (Supplementary Fig. 3): Evaluation of the cell proliferation of the bovine myoblasts. Viabilities of the bovine myoblasts of the culture well, HALC, and LALC group were compared on day 2 and day 4 ($n = 3$ independent experiments, two-tailed t-test). Error bars represent mean \pm s.d. Source data are provided a Source Data file.

We re-organized Fig. 4 (Figure R6). We included the stiffness analysis results in the supporting figures and replaced the confocal images with images with better quality. In addition, we added the quantitative analysis results of the confocal images in Fig. 4b(ii) (Figure R6b(ii)). We repeated the fatty flavor analysis experiments and added the p-values in Fig. 4c (Figure R6c). We added number of repeats in the figure description.

Figure R6 (Fig. 4): Optimization of the fat block. **a (i)** Illustration of the adipogenic differentiation of the adipose tissue-derived mesenchymal stem cells (adMSCs) depending on the scaffold, and **(ii)** the quantification of the adipogenic differentiation degree of each group based on the lipid droplet staining performed using Oil Red O staining ($n = 3$ independent experiments, two-tailed t-test). Scale bars: $50\ \mu\text{m}$ (first row) and $20\ \mu\text{m}$ (second row). **b (i)** Confocal images showing the lipid droplet formation analyzed by immunostaining. Scale bars: $50\ \mu\text{m}$ (first and second column) and $20\ \mu\text{m}$ (third column). **(ii)** The quantitative assessment of the lipid droplet amount is shown in the graph based on the confocal images using Image J software ($n = 3$ independent experiments, two-tailed t-test). **c** Peak area ratios of nonanal and 2-ethyl-1-hexanol are summed and normalized to that of the bare scaffolds ($n = 3$ independent experiments, two-tailed t-test). Error bars represent mean \pm s.d. Source data are provided as a Source Data file.

The stiffness measurement data in Fig. 4c was changed to Supplementary Fig. 15 (Figure R7) with the significance indication as below.

Figure R7 (Supplementary Fig. 15): Evaluation of the stiffness change according to adipogenic differentiation. Stiffness of each group was measured by frequency sweep at the angular frequency range from 0.1 to 1 (rad/s) ($n = 3$ independent experiments, two-tailed t-test). Error bars represent mean \pm s.d. Source data are provided a Source Data file.

The viability results of the adMSC (Supplementary Fig. 7) was moved to Supplementary Fig. 11 (Figure R8). Statistical analysis results and number of repeats are included.

Figure R8 (Supplementary Fig. 11): Evaluation of cell viability of the bovine adMSCs. Viability of the bovine adMSCs in each group was analyzed using CCK-8 assay. The cell numbers on proliferation day 2 and day 3 were compared ($n = 3$ independent experiments, two-tailed t-test). Error bars represent mean \pm s.d. Source data are provided a Source Data file.

Re-experiments and statistical analysis were performed, and the results are shown in Figure

R9.

Figure R9 (Fig. 5): Food analysis of the ACM. **a** Illustration showing the production process of ACM. **b** (i) Images of raw beef and raw ACM and (ii) the nutritional analysis of each group. **c** Heatmap of the flavors detected in the ACM and beef after grilling at 180 °C ($n = 3$ independent experiments, two-tailed t-test). N.D indicates “not detected” and N.S indicates “non-significant”. **d** Texture profile analysis comparing the factors that determine the food texture of each group after grilling at 180 °C ($n = 3$ independent experiments, two-tailed t-test). Error bars represent mean \pm s.d. Source data are provided as a Source Data file.

Comment 4. The authors make conclusions regarding the porosity of the 4 hydrogels according to Figure 2d. However, no qualitative assessment with statistical analysis was done to support them. Please perform a quantitative analysis if conclusions regarding the porosity/density of the constructs are to be made. Mind that the conditions under which SEM images are taken – such as performing freeze-drying – can alter the results. Moreover, add details regarding the SEM imaging and performed drying process (temperature, time), as those are missing in the materials and methods section.

Response: We thank the reviewer for this comment. To quantitatively compare the porosity of the hydrogels, we analyzed the pore area of each lyophilized sample in same dimensional size using image J software based on the SEM images (Figure R10). The results revealed that the pore area and the content of crosslinked alginate were inversely proportional. The pore area of the scaffolds increased in the following order: HAHC < HALC < LAHC < LALC. This trend was inversely proportional to the stiffness of the samples. According to several previous studies, the crosslinking degree of polymer, as well as the polymer concentration, affects the pore size of the scaffold, which is inversely proportional to the hardness of the hydrogel⁴⁻⁶. The experimental results in Figure. R10, Figure. R11, and Figure. R12 confirmed that our scaffolds exhibited the same trend as those reported in previous studies. Therefore, we concluded that the stiffness difference between the four hydrogels was derived from the different pore sizes of the samples, which was due to the different crosslinking degree of alginate.

Figure. R10: Pore size assessment from the SEM images of each sample (n = 3).

Figure R11 (Fig. 2c): Stiffness (Young's modulus) of the hydrogel samples.

Figure R12 (Fig. 2d): SEM images comparing the pore shape and pore size of the samples.

To prepare the lyophilized samples, we stored the hydrogels at $-20\text{ }^{\circ}\text{C}$ for one day, after which they were placed in a lyophilizer for 5 days at $-50\text{ }^{\circ}\text{C}$ collector temperature before obtaining the SEM images. We are very sorry for omitting this information in the manuscript. We thank the reviewer for this comment. We have added this information in the method section in lines 530 – 534 of the revised manuscript.

In the revised manuscript (Lines 530 - 534): The inner geometry of the scaffolds was evaluated using field-emission scanning electron microscopy (FE-SEM, JEOS, IT-500HR, USA) analysis. The hydrogels were completely lyophilized before the FE-SEM observations. To prepare the lyophilized samples, the hydrogels were stored at $-20\text{ }^{\circ}\text{C}$ for one day, and then lyophilized at $-50\text{ }^{\circ}\text{C}$ collector temperature for 5 days before obtaining the SEM images.

Comment 5. The adipogenesis degree of the mesenchymal stem cells (Figure 4) is not evident according to the LipidTox staining. No lipid droplets were observed in a clear way. We wish to refer the authors to the following paper, where successful LipidTox staining is observed: <https://pubmed.ncbi.nlm.nih.gov/35297273/> Additionally, the analysis of Oil-Red-O staining (Figure 4a_{ii}) is not clear, as no details of it are present in the paper. No control groups such as undifferentiated constructs are present.

Moreover, no additional biochemical quantitative assessments were made to verify it.

Please address this issue by performing additional qualitative (such as different/repeated staining) and quantitative (such as q-PCR) assessments to verify the adipogenic maturation degree on the two hydrogels, similarly to what you have done for myogenic differentiation experiments. **Importantly: add control hydrogels seeded with cells yet not differentiated, as references. The verification of this issue is crucial for the entirety of this paper, as many subsequent conclusions and assumptions are derived from it by the authors.**

Response: We thank the reviewer's valuable comment. We replaced the lipid droplet confocal images to obtain clearer images in Figure R13 (Fig. 4), which are now marked as Fig. 4b(i) in the revised manuscript. Red stained lipid droplets can be clearly observed in LALC, whereas almost no droplets are shown in HALC. In addition, we performed quantitative analysis based on these confocal images using Image J Software (Fig. 4b(ii)). We confirmed that the LALC sample had significantly higher amount of lipid droplets, which were formed by adMSCs, than HALC.

Figure R13 (Fig. 4): Optimization of the fat block. **a** (i) Illustration of the adipogenic differentiation of the adipose tissue-derived mesenchymal stem cells (adMSCs) depending on the scaffold, and (ii) the quantification of the adipogenic differentiation degree of each group based on the lipid droplet staining performed using Oil Red O staining ($n = 3$ independent experiments, two-tailed t-test). Scale bars: 50 μm (first row) and 20 μm (second row). **b** (i) Confocal images showing the lipid droplet formation analyzed by immunostaining. Scale bars: 50 μm (first and second column) and 20 μm (third column). (ii) The quantitative assessment of the lipid droplet amount is shown in the graph based on the confocal images using Image J software ($n = 3$ independent experiments, two-tailed t-test). **c** Peak area ratios of nonanal and 2-ethyl-1-hexanol are summed and normalized to that of the bare scaffolds ($n = 3$ independent experiments, two-tailed t-test). Error bars represent mean \pm s.d. Source data are provided as a Source Data file.

In addition, we performed Oil Red O staining for the undifferentiated samples. We cultured adMSCs on each scaffold and allowed the cells to proliferate for 3 days. Thereafter, we performed Oil Red O staining, and included the quantitative results of the Oil Red O staining in the revised manuscript (Figure R14). The results showed no significant difference in the lipid droplet formation of the LALC and HALC before induction of adipogenic differentiation.

Figure R14 (Supplementary Fig. 12): Adipogenic differentiation degrees of the undifferentiated samples. Quantification of the Oil Red O staining of the undifferentiated adMSC cultured on each scaffold was performed on the proliferation day 3 ($n = 3$ independent experiments, two-tailed t-test). Error bars represent mean \pm s.d. Source data are provided a Source Data file.

In addition, we performed the proteomic analysis of both the differentiated and undifferentiated adMSCs (Figure R15 and Figure R16). The differentiated adMSCs in each scaffold were collected and the expressed proteins were analyzed (Figure R15). We sorted the proteins that were expressed significantly higher in the LALC group, and performed gene annotation to confirm which cell functions were up-regulated in the LALC samples. As expected, the genes involved in the fatty acid and lipid metabolism were up-regulated in the LALC group. Also, we performed the same process for the undifferentiated samples. As shown in Figure R16, both the up-regulated and down-regulated genes in the LALC samples were significantly lesser than those of the differentiated groups, indicating that the difference in the protein expression of the LALC and HALC groups was not significant before inducing adipogenic differentiation. In the undifferentiated groups, it was confirmed that the proteins

related to fatty acid and lipid metabolism were also up-regulated in LALC. This suggests that the mechanical property of scaffold affects the cell metabolism even before differentiation was induced. However, we confirmed that the lipid droplet formation and fatty acid flavor enrichment did not occur in the undifferentiated groups after performing experiments on the undifferentiated groups (Figure R14 and Figure R17). Therefore, we concluded that the degree of the influence of the scaffold was significantly higher after differentiation was induced.

Figure R15 (Supplementary Fig. 13): Proteins upregulated in the differentiated adMSCs in LALC. Volcano plot indicates the significantly down-regulated (blue dots) and up-regulated (red dots) proteins in LALC compared to that in HALC ($n = 3$ independent experiments, two-

tailed t-test). Fold change indicates the ratio of the protein expression intensity in LALC to the protein expression intensity in HALC. Bar graph below the volcano plot shows the gene annotation of the proteins that are up-regulated in LALC. The X-axis indicates the proportion of the genes with function corresponding to the gene annotation, whereas the Y-axis is the description of the gene functions. Source data are provided a Source Data file.

Figure R16 (Supplementary Fig. 14): Proteins expressed in the undifferentiated adMSCs. Volcano plot indicates the significantly down-regulated (blue dots) and up-regulated (red dots) proteins in LALC compared to that in HALC ($n = 3$ independent experiments, two-tailed t-test). Fold change indicates the ratio of the protein expression intensity in LALC to the protein expression intensity in HALC. Gene annotation of the up-regulated in HALC is shown below the volcano plot. The X-axis indicates the proportion of the genes with function corresponding to the gene annotation, whereas the Y-axis is the description of the gene functions. Source data are provided a Source Data file.

Figure R17 (Supplementary Fig. 16): Flavor analysis of the undifferentiated groups. Fatty flavors detected in the undifferentiated groups were normalized to that of bare scaffolds ($n = 3$ independent experiments, two-tailed t-test). Error bars represent mean \pm s.d. Source data are provided a Source Data file.

We thank the reviewer again. We supplemented the data for the adipogenic differentiation. We improved the reliability of our manuscript by adding more quantitative comparison results and performing statistical analysis.

Comment 6. Throughout the paper, observations regarding construct characteristics are concluded to have stemmed directly from the differentiation of seeded cells – a central concept the authors aim to demonstrate in this paper. However, important controls to rule out alternative explanations which could have affected the observed results were not performed. Therefore, the authors must perform additional studies and analyze them more carefully, with appropriate repeats and control groups, to differentiate between effects from alternative factors and the effect from cellular differentiation:

A. No statistical analysis was done, or a conclusive conclusion drawn, regarding the difference in construct degradation rates of the two hydrogels. This must be further repeated and analyzed (Figures S2), as it could affect results such as total protein content.

B. When two hydrogel types were analyzed for cell cultivation – either for myoblast cells (Figure S3, Figure 3) or mesenchymal stem cells (Figure S7, Figure 4) – construct properties were checked by the authors after the differentiation phase.

- No conclusions were drawn regarding the difference in cell proliferation rates on the two hydrogels, in both cell types. These must be statistically analyzed and discussed (Figures S3, S7).

- If there is indeed a difference in cellular density at the end of the proliferation phase, it could affect cell behavior and overall construct characteristics. Controls of hydrogels seeded with proliferated but non-differentiated cells should be analyzed and compared, to rule out alternative factors other than cell maturation. Please add such controls to the stiffness assessments (Figure 3 d, Figure 4c), total protein content assessment (Figure 3c), GC analysis (Figure 3e, Figure 4e), and protein expression analysis (Figure 3f) of the constructs.

- Different ECM secretion degree by seeded cells on the two hydrogel types could also

have affected the obtained results, yet it wasn't checked at the end of cultivation of either cell type. Add ECM secretion analysis for each cell type (mesenchymal and myoblast cells), to better understand cell behavior on the different constructs (two hydrogel compositions). The results of these studies should be included in your discussion on final construct characteristics.

C. Controls of empty scaffolds of the two compositions assessed for cell cultivation should be analyzed to distinguish the effect of the composition itself on the obtained results, in the following assessments:

- **Stiffness with/without cooking (Figure dii): the properties of the scaffolding materials can change significantly under such different treatments:**

<https://pubmed.ncbi.nlm.nih.gov/31731744/>

D. Figure 5: controls of 1. empty, and 2. cell-seeded but not differentiated assembled cultured meat constructs, should both be analyzed and compared to cell seeded and differentiated constructs, as references to the obtained results. This is crucial for all 3 assessments : nutritional value, flavor, and texture.

Response: We thank the reviewer for the valuable comments. We apologize for the insufficient information and the missing control groups. By carefully reflecting on the reviewer's comment, we added the undifferentiated groups as a control group and performed additional experiments. In addition, after re-performing the experiments, we performed statistical analysis and indicated the p-values in the figures.

A. We performed the statistical analysis for Figure R18 (Supplementary Fig. 2) and confirmed that the scaffold degradation rate was not significantly different from that of LALC

and HALC. Therefore, we could rule out the effect of scaffold degradability in our study. In addition, by confirming that both the scaffolds do not fully degrade for the entire cell culture period, we ensured that both LALC and HALC can support cells to fully differentiate.

Figure R18 (Supplementary Fig. 2): Degradation rates of the LALC and HALC. The degradation degree was quantified using the weight loss of the hydrogels ($n = 3$ independent experiments, two-tailed t-test). N.S indicates “non-significant”. Error bars represent mean \pm s.d. Source data are provided a Source Data file.

B. We re-performed the cell viability test to show the exact cell numbers in each group. Thereafter, we performed the statistical analysis to investigate the significant difference between the cell density of HALC and LALC (Figure R19 and Figure R20). The results confirmed that there is no difference in the cell density with a change in the scaffold stiffness. Therefore, we could rule out the effect of different cell viability when performing the subsequent experiments. In addition, the myoblast viability of the two groups was not significantly different from that of the culture well group on the proliferation day 4, confirming the non-cytotoxicity of the scaffolds.

Figure R19 (Supplementary Fig. 3): Evaluation of the cell proliferation of the bovine myoblasts. Viabilities of the bovine myoblasts of the culture well, HALC, and LALC group were compared on day 2 and day 4 ($n = 3$ independent experiments, two-tailed t-test). Error bars represent mean \pm s.d. Source data are provided a Source Data file.

Figure R20 (Supplementary Fig. 11): Evaluation of cell viability of the bovine adMSCs. Viability of the bovine adMSCs in each group was analyzed using CCK-8 assay. The cell numbers on proliferation day 2 and day 3 were compared ($n = 3$ independent experiments, two-tailed t-test). Error bars represent mean \pm s.d. Source data are provided a Source Data file.

In addition, we added undifferentiated groups for the protein expression analysis (Figure R21), myosin heavy chain (MHC) analysis (Figure R22), total protein content assessment (Figure R23), stiffness assessment (Figure R24), and GC analysis (Figure R25) to rule out factors other than cell maturation.

Figure R21 (Supplementary Fig. 5): Proteins expressed in the undifferentiated myoblasts.

Volcano plot indicates the significantly down-regulated (blue dots) and up-regulated (red dots) proteins in HALC compared to that of LALC ($n = 3$ independent experiments, two-tailed t-test). Fold change indicates the ratio of the protein expression intensity in HALC to the protein expression intensity in LALC. Gene annotation of the up-regulated proteins is shown below the volcano plot. X-axis indicates the proportion of the genes with function corresponding to the gene annotation whereas Y-axis is the description of the gene functions. Source data are provided a Source Data file.

Figure R22 (Supplementary Fig. 6): Amount of myosin heavy chain (MHC) expressed in the undifferentiated myoblasts in each scaffold. The expression of the myosin heavy chain in the undifferentiated samples was compared ($n = 3$ independent experiments, two-tailed t-test). Error bars represent mean \pm s.d. Source data are provided a Source Data file.

Figure R23 (Supplementary Fig. 7): Amount of total protein expressed in the undifferentiated myoblasts in each scaffold. Protein expression of the undifferentiated samples was evaluated using bicinchoninic acid (BCA) assay ($n = 3$ independent experiments, two-tailed t-test). Error bars represent mean \pm s.d. Source data are provided a Source Data file.

Figure R24 (Supplementary Fig. 9): Change in the stiffness of the undifferentiated samples before and after cooking at 60 °C. The stiffness of the undifferentiated group was measured by frequency sweep at the angular frequency range from 0.1 to 1 (rad/s) ($n = 3$ independent experiments, two-tailed t-test). Error bars represent mean \pm s.d. Source data are provided a Source Data file.

Figure R25 (Supplementary Fig. 10): Flavor analysis of the undifferentiated myoblast samples performed using GC-MS. Peak area ratio of the flavor detected in the undifferentiated samples are shown ($n = 3$ independent experiments, two-tailed t-test). Error bars represent mean \pm s.d. Source data are provided a Source Data file.

In Figure R21, the proteins up-regulated in HALC were significantly fewer compared to those in the differentiated groups in Fig. 3b, suggesting that the effect of scaffold stiffness on cell metabolism is significantly higher after differentiation induction. Even though myogenesis-related proteins were up-regulated in the HALC with undifferentiated cells, we concluded that the influence of the scaffold was significantly smaller before differentiation was induced, as we confirmed that there was no significant difference in the amount of MHC (Figure R22) and total proteins (Figure R23) in the HALC and LALC of the undifferentiated groups.

In addition, we performed the stiffness and flavor analysis of the undifferentiated adMSCs. The results confirmed that stiffness was not different between LALC and HALC with differentiated adMSCs as well as with undifferentiated cells (Figure R26). In flavor analysis, there was no significant difference between the undifferentiated groups and bare scaffold groups (Figure R27), confirming that the fatty flavor is derived from adipogenic differentiation.

Figure R26 (Supplementary Fig. 15): Evaluation of the stiffness change according to adipogenic differentiation. Stiffness of each group was measured by frequency sweep at the angular frequency range from 0.1 to 1 (rad/s) ($n = 3$ independent experiments, two-tailed t-test). Error bars represent mean \pm s.d. Source data are provided a Source Data file.

Figure R27 (Supplementary Fig. 16): Flavor analysis of the undifferentiated groups. Fatty flavors detected in the undifferentiated groups were normalized to that of bare scaffolds ($n = 3$ independent experiments, two-tailed t-test). Error bars represent mean \pm s.d. Source data are provided a Source Data file.

We thank the reviewer’s comment regarding the ECM secretion of the cells depending on the scaffold type. As the reviewer pointed out, ECM secretion can vary depending on the scaffold type. Previous research reported that ECM components are highly secreted during the myogenic differentiation to build up the cellular microenvironment along with myotube and myofiber formation⁷. In addition, it is known that ECM secretion increases during adipogenic differentiation⁸. In our study, we confirmed higher amount of total proteins, which include the ECM components in the highly differentiated myoblasts group (HALC) in Figure R28. Furthermore, we observed up-regulated proteins involved in the ECM components of the highly differentiated adMSCs group (LALC) in Figure R29. The cell-derived proteins, including ECM components, in muscle blocks can affect the flavor by participating in the Maillard reaction. Therefore, we added an illustration to describe the effect of increased protein amount in HALC on the flavor of the muscle block in Fig. 3f (Figure R30f). In addition, we

added the description in the discussion section of our revised manuscript (Lines 436 – 442 and lines 448 - 453).

In the revised manuscript (Lines 436 - 442): The degree of myogenesis in the HALC is expected to be greater than that in LALC. As myoblasts differentiate, the nuclear fusion of the cells occurs, and the cell length increases along with the expression of proteins, including ECM. From the perspective of cultured meat, we expect that the biological changes in the cells during the myogenesis would not only determine the amount of differentiated muscle cells but also the food quality of the meat, such as texture and flavor.

In the revised manuscript (Lines 448 - 453): In our previous study, we discovered that cell proliferation increases the content of the protein participating in the Maillard reaction, thus affecting the flavor of the cultured meat. Because protein synthesis, such as ECM secretion and muscle protein formation, increases during the myogenic differentiation, we expected higher amounts of Maillard reaction products in the HALC than in the LALC. We confirmed this using GC-MS analysis (Fig. 3f).

For fat blocks, lipid droplet is the main factor determining the sensory properties; thus, we focused on the different contents of lipids rather than analyzing the ECM secretion.

Figure R28 (Fig. 3d): BCA assay results showing the total protein amount of the differentiated myoblasts in each group (n = 3 independent experiments, two-tailed t-test).

Figure R29 (Supplementary Fig. 13): Proteins upregulated in the differentiated adMSCs in LALC. Volcano plot indicates the significantly down-regulated (blue dots) and up-regulated (red dots) proteins in LALC compared to that in HALC ($n = 3$ independent experiments, two-tailed t-test). Fold change indicates the ratio of the protein expression intensity in LALC to the protein expression intensity in HALC. Bar graph below the volcano plot shows the gene annotation of the proteins that are up-regulated in LALC. The X-axis indicates the proportion of the genes with function corresponding to the gene annotation, whereas the Y-axis is the description of the gene functions. Source data are provided a Source Data file.

Figure R30 (Fig. 3): Optimization of the muscle block. **a** Immunostained myosin heavy chains (MHC) on differentiation day 5 and day 8. The nuclei stained with DAPI (blue), and the MHCs stained with MF20 (red). Scale bars: 20 μm (first row) and 100 μm (second row). **b** (i) Volcano plot showing the up-regulated (red dots) and down-regulated (blue dots) proteins in the HALC ($n = 3$ independent experiments, two-tailed t-test). Fold change indicates the intensity ratio of the protein expression in the HALC to that in the LALC. Proteins expressed by more than 1.5 folds in the HALC were sorted in the heatmap with the intensity of the protein expression. (ii) Gene annotation of the proteins in the heatmap. **c** Quantification of the MHC of differentiated bovine myoblasts in each group ($n = 3$ independent experiments, two-tailed t-test). **d** BCA assay results showing the total protein amount of the cells per scaffold ($n = 3$ independent experiments, two-tailed t-test). **e** Stiffness of the raw (ungrilled) groups (i) and grilled groups (ii) measured using a rheometer ($n = 3$ independent experiments, two-tailed t-test). Illustration of the stiffness change in the HALC with cells owing to an increase in the temperature from 37 °C (ii-a) to 60 °C (ii-b). N.S means “non-significant”. **f** Peak area ratio of the flavors detected from each sample (left) ($n = 3$ independent experiments, two-tailed t-test). Illustration of the flavor enrichment in the cooked HALC group (right). Error bars represent mean \pm s.d. Source data are provided as a Source Data file.

C. We apologize for the missing information. We agree with the reviewer that bare HALC and LALC scaffold can have different behavior upon heating owing to the different alginate

content. Therefore, we performed additional experiments for the bare group. We immersed bare HALC scaffold and bare LALC scaffold in myoblast proliferation medium for 4 days and in myoblast differentiation medium for another 8 days at 37 °C, which is the same preparation method of the cell seeded groups. Then, we evaluated the stiffness change of the bare HALC and bare LALC upon heating (Figure R31, Supplementary Fig. 8). The results confirmed that there was no significant change in the stiffness of both HALC and LALC upon heating. Therefore, we concluded that the change in the stiffness of HALC after heating was due to the myogenic differentiation.

Figure R31 (Supplementary Fig. 8): Stiffness change of the bare scaffolds before and after cooking at 60 °C. The stiffness of each group was measured by frequency sweep at the angular frequency range from 0.1 to 1 (rad/s) ($n = 3$ independent experiments, two-tailed t-test). Error bars represent mean \pm s.d. Source data are provided a Source Data file.

D. Controls of empty scaffolds (Bare ACM) and the undifferentiated cells-seeded scaffold (Undifferentiated ACM) were added as control groups for the nutritional, flavor, and texture profile analyses (Figure R32, Figure R33, and Figure R34). We have included these results in the supporting figures (Supplementary Fig. 17, Supplementary Fig. 19, and Supplementary Fig. 20). Compared to the bare ACM, the enrichment in the nutrition, flavor, and texture was

observed in the differentiated ACM. This behavior was not observed in the undifferentiated group, suggesting that the control of the differentiation degree of the cells significantly affects the food properties of the final cultured meat.

Figure R32 (Supplementary Fig. 17): Nutrition factors of the ACM, bare ACM, and undifferentiated ACM. Nutritional analysis of carbohydrates, fats, and proteins was conducted for the three experimental groups ($n = 3$ independent experiments, two-tailed t-test). Fats include saturated and unsaturated fats. Error bars represent mean \pm s.d. Source data are provided a Source Data file.

Figure R33 (Supplementary Fig. 19): Comparison of the flavors detected in each group. Flavors were detected in ACM, bare ACM (assembled scaffolds without cells), and undifferentiated ACM (assembled scaffolds with undifferentiated cells) by GC-MS.

Significance was indicated as: a: non-significant (N.S) to bare ACM, b: $0.01 < p\text{-value} < 0.05$ to bare ACM, c: $0.001 < p\text{-value} < 0.01$ to bare ACM, d: $0.0001 < p\text{-value} < 0.001$ to bare ACM, e: N.S to undifferentiated ACM, f: $0.01 < p\text{-value} < 0.05$ to undifferentiated differentiated ACM, g: $0.001 < p\text{-value} < 0.01$ to undifferentiated ACM, h: $0.0001 < p\text{-value} < 0.001$ to undifferentiated ACM ($n = 3$ independent experiments, two-tailed t-test). Error bars represent mean \pm s.d. Source data are provided a Source Data file.

Figure R34 (Supplementary Fig. 20): Results of the texture profile analysis of each group. Factors that determine the texture (chewiness, hardness, cohesiveness, springiness) were measured for ACM, bare ACM, and undifferentiated ACM after grilling the samples at 180 °C to confirm the effect of the cell differentiation quality control on the cultured meat texture ($n = 3$ independent experiments, two-tailed t-test). Error bars represent mean \pm s.d. Source data are provided a Source Data file.

Comment 7. Some parts of the paper are not well placed/organized.

- Please add a table describing the 4 types of prepared hydrogels, clarifying their compositions (%w/v of alginate and gelatin) and calcium crosslinking concentration.
- Conceptual explanations appear in the results section instead of the introduction. For example: lines 90-105, lines 164-166, lines 267-273, and lines 296-303 should be in the introduction, with supporting references.
- Statements regarding the authors' hypothesis appear in the results, instead of the introduction/discussion, and do not mention supporting references. For example: lines 160-162. Pay attention to such phrases throughout the results section.
- The description of experiments and analysis should be in the materials & methods section. Please rearrange the legends of figures S2,S3, S7 accordingly.

Response: We thank the reviewer for this comment. To clarify the composition of the hydrogels, we added a table to show the concentration of the polymers and crosslinking molecules in the four hydrogels (Table R2 (Supplementary Table S1)).

Hydrogel	Gelatin concentration	Alginate concentration	Microbial transglutaminase concentration	CaCl ₂ concentration
LALC	10% (w/v)	0.25% (w/v)	1% (w/v)	0.125% (w/v)
LAHC	10% (w/v)	0.25% (w/v)	1% (w/v)	0.25% (w/v)
HALC	10% (w/v)	2% (w/v)	1% (w/v)	1% (w/v)
HAHC	10% (w/v)	2% (w/v)	1% (w/v)	2% (w/v)

Table R2 (Supplementary Table S1): Composition of the hydrogels. Concentration of the polymers and crosslinkers in the hydrogels are shown in the table.

As the reviewer pointed out, the conceptual explanations in lines 90-105, lines 164-166, lines 267-273, lines 296-303, and lines 160-162 are inappropriate in the result section. Therefore, we included the explanations in the introduction and discussion of the revised manuscript in lines 60 - 64, lines 70 - 82, lines 436 - 442, lines 454 - 472.

In the revised manuscript (Lines 60 - 64): Particularly, Young's moduli of ~ 11 and ~ 3 kPa are required to stimulate myogenesis and adipogenesis, respectively. These mechanical properties can be achieved by controlling the crosslinking degree of polymer networks in the scaffold. For example, the stiffness of an alginate hydrogel depends on the crosslinking density of alginate, which affects the adipogenesis of mesenchymal stem cells.

In the revised manuscript (Lines 70 - 82): Gelatin, a helical protein derived from collagen, is widely used in the tissue engineering field owing to its cell-adhesive RGD sequence. Further, it is a widely known food additive owing to its easy dissolution at elevated temperatures and solidification with an elastic texture at room temperature. However, its low melting point makes it less stable under cell culture conditions. To address this poor stability, cytotoxic chemical crosslinkers, such as glutaraldehyde and genipin have been employed. To prevent cytotoxicity, the use of microbial transglutaminase (mTG), which forms peptide bonds between the lysine and glutamine residues of gelatin chains, has been demonstrated. This non-toxic crosslinker is widely used in the food industry, particularly for meat bonding, indicating that mTG is an attractive crosslinker for achieving a stable gelatin-based scaffold. Alginate, a brown algae-derived polysaccharide, lacks cell adhesive capacity, but can strengthen scaffolds to support cells when crosslinked with cations. Particularly, the mechanical properties of alginate hydrogel can be significantly varied by varying the concentration of the crosslinked alginate.

In the revised manuscript (Lines 436 - 442): The degree of myogenesis in the HALC is expected to be greater than that in LALC. As myoblasts differentiate, the nuclear fusion of the cells occurs, and the cell length increases along with the expression of proteins, including ECM. From the perspective of cultured meat, we expect that the biological changes in the cells during the myogenesis would not only determine the amount of differentiated muscle cells but also the food quality of the meat, such as texture and flavor.

In the revised manuscript (Lines 454 - 472): Furthermore, fat is mainly composed of adipocytes with abundant lipid droplets, which determine the quality of meat, such as juiciness, flavor, and nutrition. Adipocytes are derived from mesenchymal stem cells which can be differentiated into several types of cells, such as adipocytes, myocytes, and chondrocytes. Therefore, it is critical to provide a specific environment for the bovine stem cells to differentiate only into adipocytes, while producing a cultured fat. By optimizing the stiffness of the scaffold for adipogenesis, we expected to develop a fat block composed of highly differentiated adipocytes. The effect of the scaffold stiffness on the degree of adipogenic differentiation was investigated in Fig. 4. The results revealed that lipid droplet formation was significantly higher in the LALC scaffold, which has similar stiffness to adipose tissue. Lipid droplets contain neutral lipids, consisting of triacylglycerols (TAGs). As TAGs exhibit a structure in which three fatty acid molecules are ester-bonded to one glycerin molecule, TAG can be hydrolyzed to fatty acids by the lingual lipase in the mouth. In addition, fat can be tasted via the interaction of these fatty acids with CD36, which is known as a fat taste receptor expressed by the taste bud cells on the tongue. Therefore, the different amount of lipids formed in each scaffold implies the ability of the scaffolds to control the nutritional components and the taste of the cultured meat. Furthermore, fatty acids contribute to the flavors of slaughtered

meat by creating volatile compounds upon oxidation. In Fig. 4c, we confirmed that a higher degree of adipogenic differentiation resulted in flavor enrichment of the fat block owing to the higher amount of lipid droplets.

We also removed the sentence addressing our hypothesis in lines 225 – 229 and included them in the discussion in the revised manuscript (Lines 448 - 453).

In the revised manuscript (Lines 448 - 453): In our previous study, we discovered that cell proliferation increases the content of the protein participating in the Maillard reaction, thus affecting the flavor of the cultured meat. Because protein synthesis, such as ECM secretion and muscle protein formation, increases during the myogenic differentiation, we expected higher amounts of Maillard reaction products in the HALC than in the LALC. We confirmed this using GC-MS analysis (Fig. 3f).

We apologize for the inappropriate description in the supporting figures. We included all the experimental descriptions and analysis methods in the method section in the revised manuscript. We rearranged the legends of Supplementary Figure 2, Supplementary Figure 3, and Supplementary Figure 11 (which was figure S7 before revision).

Comment 8. There is missing or unclear information regarding performed protocols.

Please provide the following in the materials and methods section, or other relevant places:

- Please provide detailed cell isolation protocols.
- Please provide detailed media used for myogenic and adipogenic differentiation.
- In construct fabrication: please state what were the dimensions (diameter and thickness) of individual hydrogels. Please clarify how you prepared the 2% -alginate containing hydrogels, in a similar manner to described for the 0.25% ones.
- SEM imaging: add details regarding the performed drying process (temperature, time) to the constructs, as those are missing.
- Scaffold degradation analysis: what was the exact medium used to incubate the scaffolds? was it collected during a different experiment each day, then frozen for subsequent use? Please elaborate as these are specific conditions where the described degradation occurred.
- What is the “fatty flavor” detected and measured by the GC? This is missing, as well as a detailed protocol.
- When analyzing cuts with different ratios between fat and muscle content: are the ratios calculated via mass? Volume? It must be mentioned in the relevant places in the paper.

Response: We apologize for the missing and unclear information, and we thank the reviewer for giving us opportunity to supplement it. For cell isolation, the following protocols were performed, which we have added in the method section in the revised manuscript (Lines 536 - 579).

- **Cell isolation procedure:** Gluteobiceps or semitendinosus muscle and intramuscular or

subcutaneous adipose tissues were harvested from 29–31 months old male or female Hanwoo cattles immediately after slaughtering. Thereafter, the retrieved skeletal muscle and adipose tissues were used to isolate bovine primary myoblasts and adipose tissue-derived mesenchymal stem cells (adMSCs), respectively. For the myoblasts retrieval, the harvested muscle tissues were washed once in 70% (v/v) ethanol (Samchun Chemical, Seoul, Korea) and twice in 2% (v/v) antibiotic-antimycotic solution (Welgene, Korea) diluted in Dubecco's phosphate-buffered saline (DPBS; Welgene). The muscle tissues were cut into small pieces followed by a digestion process with 0.2 % (w/v) collagenase type II (Worthington Biochemical Corporation, Lakewood, NJ, USA) dissolved in high glucose-Dulbecco's modified eagle medium (HG-DMEM; Welgene) at 37 °C for 30 min. Thereafter, the muscle tissue fragments were incubated at 37 °C for 5 min in HG-DMEM supplemented with 1 % (w/v) pronase (Calbiochem, Darmstadt, Germany) for complete digestion. Subsequently, the digested muscle tissues were re-suspended in HG-DMEM supplemented with 2% (v/v) heat-inactivated fetal bovine serum (FBS; Welgene). To obtain cell pellet, the digested tissue solution was centrifuged at 1500 × g for 4 min and re-suspended in red blood cells (RBCs) lysis buffer (Sigma-Aldrich, USA) for 10 min at room temperature. The solution was filtered through a 70-μm cell strainer (SPL) and centrifuged at 1500 × g for 4 min. Subsequently, the pellet was re-suspended in the myoblast proliferation medium composed of HG-DMEM-supplemented 10% (v/v) FBS, 5 ng/ml basic fibroblast growth factor (bFGF; Peprotech, USA), and 1% (v/v) antibiotic–antimycotic. To isolate the myoblasts from the muscle-derived primary cells, muscle-derived primary cells (5×10^5) were seeded onto a 35-mm culture dish (SPL) using a myoblast proliferation medium. After 24 h, non-adherent cells were removed, and the remaining adherent cells were cultured at 37 °C in a humidified atmosphere of 5% CO₂ in air, and the medium was replaced at 2 days intervals. The adherent cells were collected at a cell confluency of 50–60% by treating the cells

with 0.05% trypsin ethylene-diamine-tetraacetic acid (trypsin-EDTA, Welgene) to obtain the bovine myoblasts.

To isolate the adMSC, the harvested adipose tissues were washed with 70% ethanol and 2% (v/v) antibiotic–antimycotic solution diluted in DPBS. Thereafter, the clean adipose tissues were cut into small pieces using surgical scissors. The tissue pieces were dispersed in 0.75% (w/v) collagenase type II dissolved in 0.25% trypsin-EDTA at 37 °C for 30 min. During the dispersion process, the small tissues-containing tubes were shaken intensively for 30 min at 37 °C, and the digested adipose tissues were re-suspended in low-glucose Dulbecco’s modified Eagle’s medium (LG-DMEM; Welgene) supplemented with 10% (v/v) FBS. After filtering using a 100-µm cell strainer, the pellets descended by centrifuging at $415 \times g$ for 5 min. Subsequently, the red blood cells were eliminated by re-suspending the pellet in RBC lysis buffer for 10 min at room temperature. The RBC-free primary cells were descended by centrifuging at $415 \times g$ for 5 min followed by re-suspension in LG-DMEM supplemented with 10% (v/v) FBS and 1% (v/v) antibiotic–antimycotic (herein referred to as adMSC proliferation medium). The adipose-derived primary cells (1×10^4) were seeded onto a 35-mm culture dish and cultured for 6–8 days in an adMSC proliferation medium at 37 °C in a humidified atmosphere of 5% CO₂ in air. Subsequently, cells that did not adhere to the culture dishes were discarded by replacing the adMSC proliferation medium at 2 days-intervals, and the colony-forming units-fibroblast (CFU-F) formed on the culture dishes were harvested using 0.25% trypsin-EDTA. Finally, bovine adMSCs were obtained.

- Differentiation medium composition: The myoblast differentiation medium is composed of HG-DMEM supplemented with 5 % (v/v) horse serum (Gibco™ Horse Serum, Thermo Fischer Scientific) and 1 % (v/v) Penicillin-Streptomycin-Glutamine (PS; Gibco™, Thermo Fischer

Scientific). The adipose differentiation medium is composed of LG-DMEM supplemented with 5 % (v/v) FBS, 10 μ M insulin (insulin from bovine pancreas, Sigma Aldrich), 1 μ M dexamethasone (Sigma Aldrich), 10 μ M ciglitizone (Sigma Aldrich), and 100 μ M oleic acid (Sigma Aldrich).

We also included this content of detailed composition of the myoblast and adipose differentiation medium in the method section (Lines 587 - 596).

- Hydrogel preparation: To prepare the hydrogel, first, 40% (w/v) fish gelatin solution was prepared by dissolving fish gelatin in distilled water at 65 °C, and alginate was simultaneously dissolved in distilled water at 85 °C to prepare 8% (w/v) and 1% (w/v) alginate solution. To prepare the 2% (w/v) alginate hydrogel, 40% (w/v) of the gelatin solution was mixed with 8% (w/v) of the alginate solution at a volume ratio of 1:1 at 65 °C for 2 h. Subsequently, 2% (w/v) mTG solution was mixed with the gelatin/alginate solution at a volume ratio of 1:1 at 40 °C, and the solution was stirred vigorously. Lastly, the high alginate (HA) solution composed of 10% (w/v) fish gelatin, 2% (w/v) alginate, and 1% (w/v) mTG was obtained. To prepare the low alginate (LA) solution containing 10% (w/v) gelatin, 0.25% (w/v) alginate, and 1% (w/v) mTG, 40% (w/v) gelatin solution was mixed with 1% (w/v) alginate solution at a volume ratio of 1:1. Subsequently, 2% (w/v) mTG solution was added to the gelatin/alginate solution at a volume ratio of 1:1, and the solution was mixed vigorously. The hydrogel precursors were poured into 12-well plates (SPL) and gelled at 4 °C for 1 h. Thereafter, the hydrogels were incubated overnight at 37 °C to activate the mTG to achieve the gelatin crosslinking. After incubation, the hydrogels were washed with distilled water and immersed in calcium chloride (CaCl_2 ; Sigma Aldrich) solution for 1 h at room temperature for the alginate crosslinking. For LALC and LAHC, 0.125% (w/v) and 0.25% (w/v) CaCl_2 solutions were used, respectively. For

HALC and HAHC, the hydrogels were immersed in 1% (w/v) and 2% (w/v) CaCl₂ solutions, respectively. Subsequently, the obtained hydrogels were washed with distilled water. Before cell culture, the hydrogels were immersed in 70% ethanol for 15 min and exposed to UV radiation for 1 h for sterilization.

We included this detailed process description of the hydrogel preparation in the method section (Line 493 - 512).

As all the hydrogels were formed in the same 12-well plate mold, the diameters of the hydrogels were 21.9 mm, which is the diameter of a single well of the 12-well plate. The thicknesses of LALC and HALC were both measured as approximately 0.63 mm (Table R3).

We included this information in the revised manuscript (Lines 152 - 154).

Hydrogel	Thickness (mm)
LALC	0.630±0.026458
HALC	0.633±0.020817

Table R3 (Supplementary Table 2): Thickness of the scaffolds. The thickness of each group was measured using a digital caliper (n = 3).

In the revised manuscript (Lines 152 - 154): The dimensions of the two hydrogels were measured, and the LALC and HALC were confirmed to exhibit the same thickness (Supplementary Table 2).

- Sample preparation for the SEM images: We are very sorry for the missing information of the lyophilization process. For freeze-drying, first, we stored the hydrogels at -20 °C for one day. Subsequently, we lyophilized the frozen hydrogels at -50 °C collector temperature for

5 days to obtain the freeze-dried hydrogels.

We have included this information the method section of the revised manuscript (Line 530 - 534).

- Scaffold degradation analysis: For the degradation test, conditioned medium was used. The samples were immersed in the sterilized bovine myoblast conditioned medium for 12 days at 37 °C to mimic the cell culture condition. The conditioned medium was replaced with the fresh conditioned medium every other day. The conditioned medium was the growth medium collected from the subcultured bovine myoblast at passage 2-3 on the second day of cell culture. Thereafter, the conditioned medium was filtered using a 0.2 μm cellulose filter (Advantec) for sterilization and stored at 4 °C until usage.

We have included this information in the revised manuscript (Line 516 - 523).

- Fatty flavor compound: The fatty flavors were detected by performing gas chromatography-mass spectrometer (GC-MS) analysis. The fatty flavors were derived from 2-ethyl-1-hexanol and nonanal. Because these two compounds share the same fatty flavor notes, we summed the detected peak area ratios of the two compounds from each sample to evaluate the fatty flavor. Both 2-ethyl-1-hexanol and nonanal are known as flavor compounds formed by the oxidation of lipid in meat fat⁹⁻¹¹. We clarified the exact compound name in the Table R4 (Supplementary Table 5).

Cas Number	Compound Name	Flavors
124-19-6	Nonanal	Fat, floral, green
104-76-7	2-ethyl-1-hexanol	Fatty, green

Table R4 (Supplementary Table 5): Flavor compound detected from the adMSC groups.

- **Analyzing cuts with different ratios of fat and muscle:** To produce the ACM with different muscle and fat ratios in Figure 5 and Figure 6, we selected the single muscle block and single fat block optimized in Figure 3 and Figure 4 as the muscle and fat unit for the ACM. For example, we assembled three muscle blocks and one fat block to fabricate the ACM with muscle:fat ratio of 3:1 in Figure 5. In Figure 6, we assembled 19 muscle blocks and 13 fat blocks to produce the strip loin ACM with muscle:fat ratio of 1.5:1. For the tenderloin ACM, we assembled 14 muscle blocks and 14 fat blocks. We tried to describe this information visually in Figure 5 and Figure 6 through the illustration, but we realized that there should be description in the main text as the reviewer pointed out. We included this information in the revised manuscript (Lines 346 - 347, lines 399 - 400).

In the revised manuscript (Lines 346 - 347): Particularly, three muscle blocks were assembled with one fat block to fabricate the ACM with a muscle:fat ratio of 3:1.

In the revised manuscript (Lines 399 - 400): For the strip loin ACM, 19 muscle and 13 fat blocks were assembled together, whereas the tenderloin ACM was fabricated by assembling 14 muscle and 14 fat blocks.

References

1. Han, Yu, et al. "Effect of pore size on cell behavior using melt electrowritten scaffolds." *Frontiers in Bioengineering and Biotechnology* 9 (2021): 629270.
2. Freeman, Fiona E., and Daniel J. Kelly. "Tuning alginate bioink stiffness and composition for controlled growth factor delivery and to spatially direct MSC fate within bioprinted tissues." *Scientific reports* 7.1 (2017): 17042.
3. Yen, Feng-Chun, et al. "Cultured meat platform developed through the structuring of edible microcarrier-derived microtissues with oleogel-based fat substitute." *Nature Communications* 14.1 (2023): 2942.
4. Kumar, Sanjay. "Stiffness does matter." *Nature materials* 13.10 (2014): 918-920.
5. Guccini, Valentina, et al. "Tuning the Porosity, Water Interaction, and Redispersion of Nanocellulose Hydrogels by Osmotic Dehydration." *ACS Applied Polymer Materials* 4.1 (2021): 24-28.
6. Paradee, Nophawan, et al. "Effects of crosslinking ratio, model drugs, and electric field strength on electrically controlled release for alginate-based hydrogel." *Journal of Materials Science: Materials in Medicine* 23 (2012): 999-1010.
7. Ojima, Koichi, et al. "Proteomic analysis of secreted proteins from skeletal muscle cells during differentiation." *EuPA Open Proteomics* 5 (2014): 1-9.
8. Nakajima, Ikuyo, et al. "Extracellular matrix development during differentiation into adipocytes with a unique increase in type V and VI collagen." *Biology of the Cell* 94.3 (2002): 197-203.
9. Watanabe, Akira, et al. "Analysis of volatile compounds in beef fat by dynamic-headspace solid-phase microextraction combined with gas chromatography–mass spectrometry." *Journal of Food Science* 73.5 (2008): C420-C425.

10. Bi, Jicai, et al. "Effect of different cooking times on the fat flavor compounds of pork belly." *Journal of Food Biochemistry* 46.8 (2022): e14184.
11. Bleicher, Julian, Elmar E. Ebner, and Kathrine H. Bak. "Formation and Analysis of Volatile and Odor Compounds in Meat—A Review." *Molecules* 27.19 (2022): 6703.

REVIEWER COMMENTS

Reviewer #1 (Remarks to the Author):

The authors revised their manuscripts based on the reviewers comments. Especially, the novelty of this study has been changed to "organoleptic properties of their assembled cultured meats". However, there are still no scientific novelty because adipogenesis and myogenesis differentiation based on elastic moduli of materials are widely reported. For example, "D.-H. Kim et al., Adv. Biosys. 2020, 4, 2000247" and "X. Cai et al., J. Cell Physiol. 2018, 233, 3418.were reported stem cell differentiation to both adipocyte and myoblasts. There are many other reports which have been already showed the same material stiffness effect on the differentiation to myoblasts and adipocytes. Moreover, "organoleptic property" generally means the sensory characteristics of the detailed analyses of over 50-100 types of flavors components, taste substrates, amino acids, fatty acids, textures, and etc. Only 8 components of flavors are of course not enough to explain "organoleptic property"... sensory scoring by professional analysts is also necessary. This modified title and statement based on "organoleptic property" seems to be exaggeration. Accordingly, unfortunately this reviewer still cannot find scientific novelty of this revised manuscript. This reviewer recommends the publication of this manuscript to specific journal such as npj Sci. food.

Additionally, the authors have to clearly explain why they added additional corresponding authors, sangmin Lee. What is the contribution of this additional author and why this author needs to be one of the corresponding authors although this reviewer was not be the author before. He was not the member of this project for making a concept, strategy, experimental setup, discussion, and summary at the starting point. This reviewer also cannot agree with the being corresponding author of this additional person....Corresponding authors should have strong responsibility of this project, not additional...

Reviewer #2 (Remarks to the Author):

I am satisfied with the author's response to my comments.

Reviewer #3 (Remarks to the Author):

The authors have made significant effort to address most of the topics raised previously. The changes made in the paper's title, as well as additional experiments conducted and data provided by the authors, imply that evident work was done to improve the manuscript and solve previously risen queries. However, there are still several important issues that still must be addressed, as will be stated hereafter. Moreover, as many new results were not discussed in the results and discussion sections, the authors are asked to add these in the relevant parts of the manuscript.

1. After the authors clarified in their first response that the topic of using 2D hydrogel surfaces with different stiffnesses instead of porous 3D scaffolds is to single-out the effect of hydrogel stiffness alone, they must also include such clarification in the manuscript, in the introduction & discussion sections. It is crucial for readers to understand that the 2D cultivation method was used to single-out the effect of cell differentiation on the food-related properties of the final tissues.
2. Regarding the abstract and further explanations of the novelty, its important to be more concise and on-point about the goals of in the research, as was clearly stated in the second response: the novelty was in assessing the organoleptic and nutritional attributes of muscle or adipose constructs, either separately or combined, resulting from tailored 2D cell differentiation on scaffolding surfaces with varying stiffnesses. This should be stated in a well-defined manner in the abstract.
3. In the introduction, some topics are still unexplained throughly: What food-related and potential sensory characteristics were evaluated in cultured meat research until now? The stiffness of constructs

was evaluated before and compared with the values of natural tissues, to analyze their mechanical nature – refer to such assessments and to others that might be relevant. Moreover, what are the common assessments done for regular meat? Why did you choose your analysis - where they one of the most common ones for regular meat? Please also elaborate on the following topics:

- Were large cultured meat constructs fabricated before? In what ways?
- What materials were used as scaffolds? (bioinks, decellularized plant tissues..?)
- Were alginate-fish gelatin used to create hydrogels as cultivation platforms?

Another example (the topic of marbled construct fabrication by small tissue attachment): it was inspired by other works (which need to be stated and cited), to create constructs relevant for comparison to real meat, in terms of organoleptic properties after cell maturation (the gap).

4. According to the added information regarding the performed SEM and porosity assessments, it is now clear that conceptually, it is incorrect to relate the porosity results of freeze-dried constructs, to the stiffness of the hydrogels on which cells were later seeded. This is because the freeze-drying process itself creates pores. As described in your work, both the compression tests and all later cell seeding experiments in the whole paper were done with non-lyophilized hydrogels. Proper porosity analysis of the wet hydrogels would need other methods such Environmental Scanning Electron Microscopy (ESEM) – where the samples aren't dried. Thus, you cannot link the porosity results of lyophilized constructs with the performance of the hydrogels used in the rest of this work. Please change all related sections in the manuscript accordingly.

5. While the authors show results for control groups in R14, was the analysis done after performing an experiment with all 4 groups? If not, this was done incorrectly and therefore should be addressed again. The analysis of adipogenic behavior on both scaffold types with and without adipogenic induction, should have been repeated where all 4 groups were grown in-parallel, and later analyzed. Moreover, it is crucial to compare the negative control groups and their respective induced ones: LALC uninduced compared to LALC induced, and HALC uninduced compared to HALC induced. This is relevant both to Oil red O staining, and confocal imaging results in Figures R13 & R14. Additionally, please address in your manuscript the fact that confocal imaging results might show the presence of lipids, but do not show mature adipocytes (on the 1st review round, you were referred to another work in our comment, showing relevant staining as reference). What stain did you use for the confocal imaging?

6. The performed proteomics could indeed aid in achieving a more accurate evaluation of the state of the cells is (Figure R15 and Figure R16). However, the same comment arises here as before: did you analyze all 4 groups after being produced and grown in the same experiment? If not, the previous statement refers to this part as well. Moreover, your evaluation of the extent of the stiffness effect prior and after adipogenic induction is very important – please add all these new results and potential conclusions to the results and discussion sections properly.

7. Regarding Figures R21-25: While the authors show results for undifferentiated control groups in order to compare to differentiated samples, were all these analyses done after performing experiments with all 4 groups together? The analysis of cell behavior on both scaffold types with and without maturation induction, should have been repeated where all 4 groups were grown in-parallel, and later analyzed. It is crucial to compare the negative control groups and their respective induced ones: LALC uninduced compared to LALC induced, and HALC uninduced compared to HALC induced. Inaccuracies may have risen if results from different experiments were compared.

8. What is shown in R26 (i) are results of stiffness of bare scaffolds compared to ones seeded with cells, unclear at what state. For example, what is the meaning of "LALC" or HALC" in this graph? do you mean that there are induced cells there? If yes, then the part comparing the effect of adipogenic differentiation was assessed fully. Please address relevant results and conclusions in your manuscript.

9. Please check your description of the results and conclusions regarding potential ECM secretion: While a higher total protein content in differentiated myoblasts was shown, total protein content might be influenced both from cell maturation and (inner protein synthesis in adults muscle fibers) or from ECM secretion, or both. However, only the overall protein content was checked, while direct ECM secretion evaluation wasn't performed to unveil the exact reason. Only a general statement on higher protein content could be made, but not on higher ECM proteins.

Response letter

Journal: *Nature Communications*

Manuscript ID: NCOMMS-22-41801-T

Original Title: Block-Assembled Edible Scaffold for Cultured Meat

Revised Title: Block-Assembled Cultured Meat with Enriched Organoleptic Properties

Authors: Milae Lee, Sohyeon Park, Bumgyu Choi, Woojin Choi, Hyun Lee, Jeong Min Lee, Seung Tae Lee, Ki Hyun Yoo, Dongoh Han, Geul Bang, Heeyoun Hwang, Won-Gun Koh, Sangmin Lee, Jinkee Hong

Response to Reviewer #1

Reviewer(s)'s Comments to Author:

Reviewer 1.

Comments

The authors revised their manuscripts based on the reviewers comments. Especially, the novelty of this study has been changed to "organoleptic properties of their assembled cultured meats". However, there are still no scientific novelty because adipogenesis and myogenesis differentiation based on elastic moduli of materials are widely reported. For example, "D.-H. Kim et al., Adv. Biosys. 2020, 4, 2000247" and "X. Cai et al., J. Cell Physiol. 2018, 233, 3418.were reported stem cell differentiation to both adipocyte and myoblasts. There are many other reports which have been already showed the same material stiffness effect on the differentiation to myoblasts and adipocytes. Moreover, "organoleptic property" generally means the sensory characteristics of the detailed analyses of over 50-100 types of flavors components, taste substrates, amino acids, fatty acids, textures, and etc. Only 8 components of flavors are of course not enough to explain "organoleptic property" sensory scoring by professional analysts is also necessary. This modified title and statement based on "organoleptic property" seems to be exaggeration. Accordingly, unfortunately this reviewer still cannot find scientific novelty of this revised manuscript. This reviewer recommends the publication of this manuscript to specific journal such as npj Sci. food.

Additionally, the authors have to clearly explain why they added additional corresponding authors, sangmin Lee. What is the contribution of this additional author and why this author needs to be one of the corresponding authors although this reviewer

was not be the author before. He was not the member of this project for making a concept, strategy, experimental setup, discussion, and summary at the starting point. This reviewer also cannot agree with the being corresponding author of this additional person....Corresponding authors should have strong responsibility of this project, not additional...

Response: We thank the reviewer for reviewing our manuscript once more and providing objective and truthful comments about the manuscript. We regret the points regarding the novelty of the work raised by the reviewer. As the reviewer pointed out, stiffness control for cell cultivation and regulating differentiation has been conventionally used in tissue engineering field. However, we want to address that our work mainly focuses on revealing the effect of cell differentiation degree on various organoleptic properties of cultured meat. Controlling the matrix stiffness was used as a tool to investigate the relationship between the cell differentiation and sensorial characteristics of cultured meat in this study. To our knowledge, although food-related properties such as texture, flavor, and nutrition are important in cultured meat, there were only a few studies to control these properties. Recent cultured meat research have made an effort to evaluate the sensory properties of cultured meat compared to that of conventional meat by measuring single Young's modulus of cultured meat¹⁻⁴. Only a few research have addressed specific texture profile parameters or nutritional facts of cultured meat to evaluate the food-related characteristics of cultured meat^{5,6}. Other important organoleptic properties like flavor and taste were only assessed in our previous work⁷. Even though texture, flavor, and nutritional values are all important characteristics that cultured meat should have as a food product, to our knowledge, no cultured meat research has evaluated all of these characteristics. In our work, we evaluated texture profile parameters, flavors, and

nutritional values of cultured meat and compared them to that of slaughtered meat which have not investigated thoroughly in other previous works. Also, we confirmed that the differentiation degree of cells significantly affects the overall food-related characteristics (flavor, texture, and nutritional value) of cultured meat. By reflecting the reviewer's valuable comments, we felt the need to express the novelty of our research more concisely and clearly in our manuscript. Therefore, we revised the manuscript as below.

In the revised manuscript (Lines 25 - 35): Research on cultured meat has primarily focused on the mass proliferation or differentiation of muscle cells; thus, the food characteristics of cultured meat remain relatively underexplored. As the quality of meat is determined by its organoleptic properties, cultured meat with similar sensory characteristics to slaughter meat is highly desirable. In this study, we control the organoleptic and nutritional properties of the cultured meat by tailoring the 2D cell differentiation on scaffolds with varying stiffness. We assess the effect of muscle and adipose differentiation quality on the sensory properties of cultured meat. Thereafter, we finally fabricate cultured meat with similar sensory profiles to that of conventional beef by assembling the muscle and adipose constructs composed of highly differentiated cells. In short, we suggest a strategy to produce the cultured meat with enriched food characteristics by regulating cell differentiation with scaffold engineering.

We also appreciate the reviewer's comments on our research concept focusing on organoleptic properties. As the reviewer point out, more than 50 to 100 flavor components can be evaluated along with the professional sensory scoring⁸. However, there are also many previous studies that analyzed fewer flavor components in food⁹⁻¹¹. Even though a professional sensory scoring was not performed, we assigned the flavors of each flavor compound using the

Flavor and Extract Manufacturers Association (FEMA) library which provides flavor and safety information of the flavor compounds by flavor experts. In this study, rather than an in-depth analysis of flavor compounds, we intended to perform a comprehensive analysis of flavor, texture, and nutritional values, which are representative organoleptic characteristics of food. We regret that more evaluations were not performed as pointed by the reviewer. By reflecting the valuable comments from the reviewer, we humbly acknowledge that the sensorial evaluation we performed throughout this work still can be insufficient. However, we hopefully want to persuade that our study includes lab-scale cultured meat analysis and that the performed assessments can be more varied with a further developed cultured meat prototype based on this research in the future. We also want to address the potential of our study which can contribute to the fabrication of cultured meat with enhanced organoleptic properties comparable to those of slaughtered meat. Although our research is still insufficient to design perfect cultured meat with the identical properties as meat, we believe that based on our research, cultured meat that embodies various food-related characteristics of slaughtered meat can be developed in the near future.

We thank for the reviewer's comment on the corresponding author issue. Sangmin Lee was added as a corresponding author during the first revision for his contribution to the overall revision of the manuscript. During the first revision, we changed almost all figures and contents of the manuscript under guidance of both Jinkee Hong and Sangmin Lee. Furthermore, Sangmin Lee has provided funding resources in this work. Therefore, we added Sangmin Lee as a corresponding author in the revised manuscript.

We sincerely appreciate all the comments from the reviewer once again.

References

1. Kang, Dong-Hee, et al. "Engineered whole cut meat-like tissue by the assembly of cell fibers using tendon-gel integrated bioprinting." *Nature Communications* 12.1 (2021): 5059.
2. Ben-Arye, Tom, et al. "Textured soy protein scaffolds enable the generation of three-dimensional bovine skeletal muscle tissue for cell-based meat." *Nature Food* 1.4 (2020): 210-220.
3. Furuhashi, Mai, et al. "Formation of contractile 3D bovine muscle tissue for construction of millimetre-thick cultured steak." *npj Science of Food* 5.1 (2021): 6.
4. Jeong, Dayi, et al. "Efficient Myogenic/Adipogenic Transdifferentiation of Bovine Fibroblasts in a 3D Bioprinting System for Steak-Type Cultured Meat Production." *Advanced Science* 9.31 (2022): 2202877.
5. Chen, Yan, et al. "Gellan gum-gelatin scaffolds with Ca²⁺ crosslinking for constructing a structured cell cultured meat model." *Biomaterials* 299 (2023): 122176.
6. Liu, Ye, et al. "Engineered meatballs via scalable skeletal muscle cell expansion and modular micro-tissue assembly using porous gelatin micro-carriers." *Biomaterials* 287 (2022): 121615.
7. Lee, Milae, et al. "Tailoring a gelatin/agar matrix for the synergistic effect with cells to produce high-quality cultured meat." *ACS Applied Materials & Interfaces* 14.33 (2022): 38235-38245.
8. Ruan, Eric D., et al. "Analysis of volatile and flavor compounds in grilled lean beef by stir bar sorptive extraction and thermal desorption—gas chromatography mass spectrometry." *Food analytical methods* 8 (2015): 363-370.
9. Ko, Hye Sung, et al. "Aroma active compounds of bulgogi." *Journal of food science* 70.8 (2005): c517-c522.
10. Resconi, Virginia C., et al. "Relationship between odour-active compounds and flavour

perception in meat from lambs fed different diets." *Meat Science* 85.4 (2010): 700-706.

11. Lin, Fang M., and Walter F. Wilkens. "Volatile flavor components of coconut meat." *Journal of Food Science* 35.5 (1970): 538-539.

Response to Reviewer #3

Reviewer(s)'s Comments to Author:

Reviewer 3.

Comments

The authors have made significant effort to address most of the topics raised previously. The changes made in the paper's title, as well as additional experiments conducted and data provided by the authors, imply that evident work was done to improve the manuscript and solve previously risen queries. However, there are still several important issues that still must be addressed, as will be stated hereafter. Moreover, as many new results were not discussed in the results and discussion sections, the authors are asked to add these in the relevant parts of the manuscript.

Response: We sincerely appreciate the reviewer for reviewing our manuscript once more and providing objective and truthful comments about the manuscript. In responding to the reviewer's grateful comments and conducting additional experiments during the first revision period, we could improve the quality of our manuscript with clearer experimental results. We also appreciate the second opportunity to further develop our manuscript. We carefully revised our manuscript by reflecting the reviewer's valuable comments point-by-point as presented below. Changes made according to the reviewer's comments are highlighted in blue in the point-by-point responses.

Comment 1. After the authors clarified in their first response that the topic of using 2D hydrogel surfaces with different stiffnesses instead of porous 3D scaffolds is to single-out the effect of hydrogel stiffness alone, they must also include such clarification in the manuscript, in the introduction & discussion sections. It is crucial for readers to understand that the 2D cultivation method was used to single-out the effect of cell differentiation on the food-related properties of the final tissues.

Response: We thank the reviewer for pointing out this. We agree with the reviewer. To clarify that we used 2D hydrogel throughout the research, we added the description in the introduction and discussion sections as below:

In the revised manuscript (Lines 28 - 30): In this study, we control the organoleptic and nutritional properties of the cultured meat by tailoring the 2D cell differentiation on scaffolds with varying stiffness.

In the revised manuscript (Lines 86 - 88): First, we fabricate 2D hydrogel scaffolds composed of fish gelatin and alginate, and the stiffness of the hydrogels is controlled by varying the content of the crosslinked alginate.

In the revised manuscript (Lines 441 - 443): In this study, 2D hydrogel was used as a scaffold to control cell differentiation according to the physical properties of the scaffold without being affected by nutrition delivery or cell viability according to the porosity of the scaffold.

Comment 2. Regarding the abstract and further explanations of the novelty, its important to be more concise and on-point about the goals of in the research, as was clearly stated in the second response: the novelty was in assessing the organoleptic and nutritional attributes of muscle or adipose constructs, either separately or combined, resulting from tailored 2D cell differentiation on scaffolding surfaces with varying stiffnesses. This should be stated in a well-defined manner in the abstract.

Response: We thank the reviewer for pointing out this. As the reviewer pointed out, we need to convey the novelty of our work more concisely in the abstract. Therefore, we revised our abstract by carefully reflecting the reviewer's comment.

In the revised manuscript (Lines 25 - 35): Research on cultured meat has primarily focused on the mass proliferation or differentiation of muscle cells; thus, the food characteristics of cultured meat remain relatively underexplored. As the quality of meat is determined by its organoleptic properties, cultured meat with similar sensory characteristics to slaughter meat is highly desirable. In this study, we control the organoleptic and nutritional properties of the cultured meat by tailoring the 2D cell differentiation on scaffolds with varying stiffness. We assess the effect of muscle and adipose differentiation quality on the sensory properties of cultured meat. Thereafter, we finally fabricate cultured meat with similar sensory profiles to that of conventional beef by assembling the muscle and adipose constructs composed of highly differentiated cells. In short, we suggest a strategy to produce the cultured meat with enriched food characteristics by regulating cell differentiation with scaffold engineering.

Comment 3. In the introduction, some topics are still unexplained throughly: What food-related and potential sensory characteristics were evaluated in cultured meat research until now? The stiffness of constructs was evaluated before and compared with the values of natural tissues, to analyze their mechanical nature – refer to such assessments and to others that might be relevant. Moreover, what are the common assessments done for regular meat? Why did you choose your analysis - where they one of the most common ones for regular meat? Please also elaborate on the following topics:

- Were large cultured meat constructs fabricated before? In what ways?
- What materials were used as scaffolds? (bioinks, decellularized plant tissues..?)
- Were alginate-fish gelatin used to create hydrogels as cultivation platforms?

Another example (the topic of marbled construct fabrication by small tissue attachment): it was inspired by other works (which need to be stated and cited), to create constructs relevant for comparison to real meat, in terms of organoleptic properties after cell maturation (the gap).

Response: We appreciate the reviewer for this comment. Recent cultured meat research have made an effort to evaluate the sensory properties of cultured meat compared to that of conventional meat by measuring the single Young's modulus of cultured meat¹⁻⁴. Only a few research have addressed the texture profile or nutritional facts of cultured meat to evaluate the food-related characteristics of cultured meat^{5,6}. Other important organoleptic properties like flavor and taste were only assessed in our previous work⁷. Even though texture, flavor, and nutritional values are all important characteristics that cultured meat should have as a food

product, to our knowledge, no cultured meat research has evaluated all of these characteristics thoroughly.

Also, the compression test was performed to evaluate the stiffness of scaffolds, of which method is commonly used for stiffness measurement of living tissues⁸. The stiffness change of muscle and fat blocks were assessed with rheometer which is also widely used for the mechanical characterization of soft tissues⁹. We also used this rheological analysis to evaluate the change in mechanical properties due to the cell culture in the cell-sheet based cultured meat in our previous paper¹⁰. For conventional meat, texture profile analysis (TPA), single compression, puncture test, Warner-Bratzler test were used for texture analysis¹¹. Among these assessments, TPA and Warner-Bratzler test are the most popular methods. In this study, we used texture profile analysis (TPA) to compare the texture of cultured meat and conventional beef because various texture parameters such as hardness, chewiness, cohesiveness can be addressed by TPA method. Also, it is known that TPA is more useful for evaluating texture of cooked meat than the Warner-Bratzler method¹². Therefore, we used the TPA method to accurately compare the sensory texture of cooked cultured meat and cooked beef. Furthermore, flavors of meat are commonly assessed using gas chromatography-mass spectrometry which we used for cultured meat flavor analysis in this study^{13,14}. This method can detect volatile compounds with fragrance from meat. For nutrition assessment, we performed Kjeldahl digestion and gas chromatography analysis for protein and fat, respectively. Amount of carbohydrates was obtained by subtracting the amount of water, crude protein, crude fat, and ash in the sample. We chose these methods because they are generally used for nutritional evaluation of foods including meat^{15,16}. Also, we performed all the procedures following the legal food regulations according to the Ministry of Food and Drug Safety in Korea to accurately evaluate the nutritional values of cultured meat and slaughtered beef.

A few studies have reported a large construct of cultured meat using 3D printing technique or assembly method with enzymatic crosslinkers. Lingshan Su et al (2022) developed a 3D-printed cultured meat with diameter of 2.5 cm and thickness of 200 μm ¹⁷ and Dayi Jeong et al (2022) reported a steak-type cultured meat with 3.42 cm width using GelMA ink and 3D printing technique¹⁸. Also, Kang et al (2021) assembled the bioprinted cell-ECM fibers using an enzymatic crosslinker to fabricate cultured meat¹⁹. However, these studies only assess the stiffness of final cultured meat compared to that of slaughtered meat. They mostly focus on mimicking the appearance and shape of meat. Partially inspired by these previous works, we developed the large cultured meat construct by assembling small pieces of cultured muscle and fats. However, we focused on mimicking the organoleptic characteristics of meat rather than its appearance. We thank the reviewer for kindly providing an example of this perspective. By regarding the reviewer's valuable comments, we supplemented the introduction of the revised manuscript by adding more descriptions on the food analyzing methods and on the reported studies addressing large cultured meat constructs.

In the revised manuscript (Lines 98 - 100): Furthermore, the food-related characteristics such as sensorial properties and nutritional values of the final assembled cultured meat were thoroughly assessed using the evaluation method commonly used in food and meat studies.

In the revised manuscript (Lines 93 - 97): In many cultured meat studies, efforts have been made to fabricate large construct of cultured meat through 3D printing technique or assembly methods with crosslinking agents to mimic large dimensions and appearance of meat. Partially inspired by these previous works, we developed the assembled cultured meat structure similar to slaughtered meat, in terms of organoleptic characteristics after cell differentiation.

In previous studies in cultured meat, ECM-derived bioinks or coatings were used for stable cell culture^{2,19}. Gelatin and polysaccharides such as alginate and chitosan were also used for scaffold ingredient in the form of hydrogels in cultured meat studies^{20,21}. Since these materials have conventionally used as scaffold materials in tissue engineering field due to high biocompatibility and high cytoaffinity, cultured meat studies have also utilized them. We have included the information of scaffold components, fish gelatin and sodium alginate, in the introduction (Lines 70-73 and lines 79-82 in the manuscript). However, we supplemented the introduction of the revised manuscript to explain that gelatin and alginate are now commonly used for cell scaffold in cultured meat research as well as tissue engineering field.

In the revised manuscript (Lines 70 - 73): Gelatin, a helical protein derived from collagen, is now widely used in cultured meat research as well as in tissue engineering field owing to its cell-adhesive RGD sequence. Further, it is a widely known food additive owing to its easy dissolution at elevated temperatures and solidification with an elastic texture at room temperature.

In the revised manuscript (Lines 80 - 84): Alginate, a brown algae-derived polysaccharide, lacks cell adhesive capacity, but can strengthen scaffolds to support cells when crosslinked with cations. Particularly, the mechanical properties of alginate hydrogel can be significantly varied by varying the concentration of the crosslinked alginate. For this reason, alginate has also been widely used for scaffold component to regulate the physical properties of scaffold.

References

1. Kang, Dong-Hee, et al. "Engineered whole cut meat-like tissue by the assembly of cell fibers using tendon-gel integrated bioprinting." Nature Communications 12.1 (2021): 5059.

2. Ben-Arye, Tom, et al. "Textured soy protein scaffolds enable the generation of three-dimensional bovine skeletal muscle tissue for cell-based meat." *Nature Food* 1.4 (2020): 210-220.
3. Furuhashi, Mai, et al. "Formation of contractile 3D bovine muscle tissue for construction of millimetre-thick cultured steak." *npj Science of Food* 5.1 (2021): 6.
4. Jeong, Dayi, et al. "Efficient Myogenic/Adipogenic Transdifferentiation of Bovine Fibroblasts in a 3D Bioprinting System for Steak-Type Cultured Meat Production." *Advanced Science* 9.31 (2022): 2202877.
5. Chen, Yan, et al. "Gellan gum-gelatin scaffolds with Ca²⁺ crosslinking for constructing a structured cell cultured meat model." *Biomaterials* 299 (2023): 122176.
6. Liu, Ye, et al. "Engineered meatballs via scalable skeletal muscle cell expansion and modular micro-tissue assembly using porous gelatin micro-carriers." *Biomaterials* 287 (2022): 121615.
7. Lee, Milae, et al. "Tailoring a gelatin/agar matrix for the synergistic effect with cells to produce high-quality cultured meat." *ACS Applied Materials & Interfaces* 14.33 (2022): 38235-38245.
8. Guimarães, Carlos F., et al. "The stiffness of living tissues and its implications for tissue engineering." *Nature Reviews Materials* 5.5 (2020): 351-370.
9. Chaudhuri, Ovijit, et al. "Effects of extracellular matrix viscoelasticity on cellular behaviour." *Nature* 584.7822 (2020): 535-546.
10. Park, Sohyeon, et al. "Gelatin MAGIC powder as nutrient-delivering 3D spacer for growing cell sheets into cost-effective cultured meat." *Biomaterials* 278 (2021): 121155.
11. Schreuders, Floor KG, et al. "Texture methods for evaluating meat and meat analogue structures: A review." *Food Control* 127 (2021): 108103.
12. De Huidobro, F. Ruiz, et al. "A comparison between two methods (Warner–Bratzler and

texture profile analysis) for testing either raw meat or cooked meat." *Meat science* 69.3 (2005): 527-536.

13. Sun, Yuwei, Yu Zhang, and Huanlu Song. "Variation of aroma components during frozen storage of cooked beef balls by SPME and SAFE coupled with GC-O-MS." *Journal of Food Processing and Preservation* 45.1 (2021): e15036.

14. Jin, Yuxi, et al. "Identification of the main aroma compounds in Chinese local chicken high-quality meat." *Food chemistry* 359 (2021): 129930.

15. Sáez-Plaza, Purificación, et al. "An overview of the Kjeldahl method of nitrogen determination. Part II. Sample preparation, working scale, instrumental finish, and quality control." *Critical Reviews in Analytical Chemistry* 43.4 (2013): 224-272.

16. Enser, M., et al. "Fatty acid content and composition of English beef, lamb and pork at retail." *Meat science* 42.4 (1996): 443-456.

17. Su, Lingshan, et al. "3D-Printed prolamin scaffolds for cell-based meat culture." *Advanced Materials* 35.2 (2023): 2207397.

18. Jeong, Dayi, et al. "Efficient Myogenic/Adipogenic Transdifferentiation of Bovine Fibroblasts in a 3D Bioprinting System for Steak-Type Cultured Meat Production." *Advanced Science* 9.31 (2022): 2202877.

19. Kang, Dong-Hee, et al. "Engineered whole cut meat-like tissue by the assembly of cell fibers using tendon-gel integrated bioprinting." *Nature Communications* 12.1 (2021): 5059.

20. Li, Linzi, et al. "Chitosan-sodium alginate-collagen/gelatin three-dimensional edible scaffolds for building a structured model for cell cultured meat." *International Journal of Biological Macromolecules* 209 (2022): 668-679.

21. Chen, Yan, et al. "Gellan gum-gelatin scaffolds with Ca²⁺ crosslinking for constructing a structured cell cultured meat model." *Biomaterials* 299 (2023): 122176.

Comment 4. According to the added information regarding the performed SEM and porosity assessments, it is now clear that conceptually, it is incorrect to relate the porosity results of freeze-dried constructs, to the stiffness of the hydrogels on which cells were later seeded. This is because the freeze-drying process itself creates pores. As described in your work, both the compression tests and all later cell seeding experiments in the whole paper were done with non-lyophilized hydrogels. Proper porosity analysis of the wet hydrogels would need other methods such Environmental Scanning Electron Microscopy (ESEM) – where the samples aren't dried. Thus, you cannot link the porosity results of lyophilized constructs with the performance of the hydrogels used in the rest of this work. Please change all related sections in the manuscript accordingly.

Response: We thank the reviewer for this comment. As the reviewer pointed out, the scaffolds through out this study were non-lyophilized hydrogels and only the samples used for SEM imaging were freeze-dried. Therefore, rather than stating that the mechanical properties of the scaffolds are affected by the voids formed by the pores, we tried to relate the density of the alginate network with the stiffness of the hydrogels. Previous studies have confirmed the relationship between porosity and mechanical properties in non-lyophilized hydrogels^{1,2}. As the polymer density increases due to the increment of polymer mass or crosslinking degree, the water content in the hydrogel decreases which results lower porosity. The pores shown in the SEM image are spaces occupied by water in the hydrogel, and as the ratio of this space increases, the density of the polymer network that determines the stiffness of the hydrogel decreases. We confirmed the same trend in our work. In Figure 2d, we wanted to address that the various stiffness of LALC, LAHC, HALC, and HAHC hydrogels are caused by differences in alginate content and crosslinking degree of the polymer. However, we realized that the

description in our manuscript was insufficient. Therefore, we revised the manuscript to relate Figure 2c and Figure 2d more effectively.

In the revised manuscript (Lines 158 - 162): Pores are the spaces occupied by water in the hydrogel. Therefore, these results imply that the water content in the hydrogel is decreased. As the content of crosslinked alginate increased, the space occupied by water decreased and the density of the alginate network increased, resulting in higher stiffness of the scaffold.

References

1. Khoury, Luai R., and Ionel Popa. "Chemical unfolding of protein domains induces shape change in programmed protein hydrogels." *Nature communications* 10.1 (2019): 5439.
2. Rizwan, Muhammad, et al. "Sequentially-crosslinked bioactive hydrogels as nano-patterned substrates with customizable stiffness and degradation for corneal tissue engineering applications." *Biomaterials* 120 (2017): 139-154.

Comment 5. While the authors show results for control groups in R14, was the analysis done after performing an experiment with all 4 groups? If not, this was done incorrectly and therefore should be addressed again. The analysis of adipogenic behavior on both scaffold types with and without adipogenic induction, should have been repeated where all 4 groups were grown in-parallel, and later analyzed. Moreover, it is crucial to compare the negative control groups and their respective induced ones: LALC uninduced compared to LALC induced, and HALC uninduced compared to HALC induced. This is relevant both to Oil red O staining, and confocal imaging results in Figures R13 & R14. Additionally, please address in your manuscript the fact that confocal imaging results might show the presence of lipids, but do not show mature adipocytes (on the 1st review round, you were referred to another work in our comment, showing relevant staining as reference). What stain did you use for the confocal imaging?

Response: We thank the reviewer for this comment. As we added the undifferentiated groups data during the first revision period, we performed all the experiments for both undifferentiated groups (LALC and HALC with undifferentiated cells) and differentiated groups (LALC and HALC with differentiated cells) together. Therefore, the data for all the differentiated groups that were originally submitted were fully changed after the first revision period. All four groups were grown in-parallel in the same condition, then analyzed.

We also compared the groups before and after differentiation induction in the same scaffold for the pointed figures by the reviewer (Figure R1). By comparing the neutral lipid formation between the groups before and after inducing the adipogenic differentiation in the same scaffold stiffness, it was confirmed that the degree of adipogenesis is significantly higher in the scaffold having similar stiffness to adipose tissue (LALC). Through this comparison, we could

more clearly confirm that the degree of cell differentiation can vary depending on the scaffold stiffness, even if differentiation is induced for both scaffold types using the differentiation medium. This difference eventually affects the flavor of cultured meat through further experiments.

Figure R1: Neutral lipid contents of differentiated adMSCs and undifferentiated adMSCs on the (i) LALC and (ii) HALC are compared ($n = 3$).

For the confocal images, we used HCS LipidTOX™ Red Neutral Lipid Stain for cellular imaging (Invitrogen, H34476) as a dye molecule. This dye stains neutral lipid droplets in the fixed adipose cells which basically stains the same molecule as Oil Red O staining. Oil Red O staining assay has the advantage of being able to quantitatively compare the degree of lipid staining between experimental groups by dissolving the dye with isopropanol after staining, but it is difficult to clearly identify the stained lipid droplets around the cell nucleus like confocal images. Therefore, we included both the Oil Red O staining results and confocal images in Figure 4. Since Figure R1 showed the quantitative results of Oil Red O of neutral lipid formation before and after the differentiation induction on the same scaffold group, confocal imaging, which is the same staining method, was not performed further.

Also, we appreciate the reviewer's comment about the lipid staining of mature adipocyte.

Referred to the reviewer's comments with mature adipocyte staining, our lipid staining data in the confocal images are insufficient to say that it is a mature adipocyte stage. Therefore, in the revised manuscript, we included the description in the results section to indicate that our data cannot ensure the maturity of adipogenesis but only addresses the difference in the lipid droplet formation depending on the scaffold.

In the revised manuscript (Lines 315 - 320): The results revealed that a higher amount of neutral lipids was deposited around the nuclei of the cells in the LALC. **It was only confirmed that the difference in the lipid droplet formation occurred depending on the scaffold stiffness, rather than precisely evaluating the adipocyte maturity. Therefore, it cannot be said that the adipocytes are fully mature on the scaffold, but it was confirmed that the degree of adipogenic differentiation was regulated by the scaffold stiffness.**

Comment 6. The performed proteomics could indeed aid in achieving a more accurate evaluation of the state of the cells is (Figure R15 and Figure R16). However, the same comment arises here as before: did you analyze all 4 groups after being produced and grown in the same experiment? If not, the previous statement refers to this part as well. Moreover, your evaluation of the extent of the stiffness effect prior and after adipogenic induction is very important – please add all these new results and potential conclusions to the results and discussion sections properly.

Response: We thank the reviewer for this comment. We performed experiment for the undifferentiated control groups in-parallel with the differentiated groups. Reflecting the reviewer’s comments, we compared the protein expression results before and after differentiation induction in the same scaffold type (Figure R2). As a result, we confirmed that the expression of the adipogenic differentiation-related proteins was increased both in LALC and HALC by treating the differentiation medium. However, regarding to the Figure R1, we ensured that although adipogenic differentiation was induced in both scaffold types, the degree of differentiation varied depending on the scaffold stiffness, resulting in differences in lipid droplet formation.

Figure R2. Increase in expression of the adipogenic differentiation-related proteins was confirmed both in (i) LALC after differentiation induction compared to that of undifferentiated LALC group and (ii) HALC after differentiation induction compared to

that of undifferentiated HALC group ($n = 3$).

We also included the conclusion about the stiffness data of adMSCs in the results section of the revised manuscript. From the result of the stiffness change in the adMSC groups, we confirmed that cell cultivation and differentiation of adMSCs do not significantly affect the physical properties of cultured meat. We included this conclusion both in the results and discussion section of the revised manuscript.

In the revised manuscript (Lines 332 - 337): The change in the stiffness of the differentiated group was not significant compared to those of the bare and undifferentiated groups (Supplementary Fig. 15). **Therefore, we concluded that cell cultivation and differentiation of adMSCs do not significantly affect the physical properties of cultured meat.** However, the increment of the fatty flavor was higher in the LALC with differentiated adMSCs than in the HALC group (Fig. 4c and Supplementary Table 5). **These results implies that the adipogenic differentiation does not affect the physical properties of the cultured meat construct but only the flavors.**

In the revised manuscript (Lines 483 - 486): **Although cell cultivation and differentiation of adMSCs did not significantly affect the stiffness of cultured meat,** the results revealed that lipid droplet formation was significantly higher in the LALC scaffold, which has similar stiffness to adipose tissue.

Comment 7. Regarding Figures R21-25: While the authors show results for undifferentiated control groups in order to compare to differentiated samples, were all these analyses done after performing experiments with all 4 groups together? The analysis of cell behavior on both scaffold types with and without maturation induction, should have been repeated where all 4 groups were grown in-parallel, and later analyzed. It is crucial to compare the negative control groups and their respective induced ones: LALC uninduced compared to LALC induced, and HALC uninduced compared to HALC induced. Inaccuracies may have risen if results from different experiments were compared.

Response: We thank the reviewer for this comment. We cultured the undifferentiated control groups with the differentiated groups in-parallel for all experiments performed during the first revision. Therefore, it can be noticed that most of the experimental results of the differentiated groups were changed after the first revision due to the re-performing experiments in-parallel with the undifferentiated groups. For the Figures R21-25 (Supplementary Fig. 5-7, Supplementary Fig. 9-10), we also compared the LALC groups before and after the differentiation induction as well as the HALC groups before and after the induction (Figure R3-R7). In Figure R3, we confirmed that the expression of the proteins related to actin-binding, MHC1, and thick filament proteins were increased after differentiation induction. Therefore, we confirmed that myogenic differentiation was induced both in LALC and HALC after changing the medium to induction medium. Also, the total amount of proteins synthesized from myoblast was significantly increased both in LALC and HALC after cell maturation. However, significant increase in the amount of myosin heavy chain (MHC) after differentiation induction was only confirmed in HALC. Due to this difference in the amount of MHC, stiffness change

upon heating at 60 °C after differentiation induction was significantly higher in the HALC group. Furthermore, roasted beef-like flavor was only found in the HALC group after differentiation induction. By comparing the results of undifferentiated groups and differentiation groups which were cultured on the same scaffold type, we could ensure that although myogenic differentiation was induced in both LALC and HALC groups by changing the medium, the degree of differentiation varied depending on the scaffold stiffness.

Figure R3. Myogenic differentiation-related proteins of which expression increased in (i) LALC after differentiation induction and (ii) HALC after differentiation induction ($n = 3$).

Figure R4. Amount of total protein synthesized from myoblast was compared before and after differentiation induction in (i) LALC and (ii) HALC ($n = 3$).

Figure R5. Myosin heavy chain (MHC) synthesis of myoblast was compared before and after differentiation induction in (i) LALC and (ii) HALC ($n = 3$).

Figure R6. Increase in stiffness compared to bare scaffold when heated at 60 °C before and after differentiation induction in (i) LALC and (ii) HALC ($n = 3$).

Figure R7. Flavor analysis of the (i) LALC groups and (ii) HALC groups before and after myogenic differentiation induction ($n = 3$).

Comment 8. What is shown in R26 (i) are results of stiffness of bare scaffolds compared to ones seeded with cells, unclear at what state. For example, what is the meaning of “LALC” or HALC” in this graph? do you mean that there are induced cells there? If yes, then the part comparing the effect of adipogenic differentiation was assessed fully. Please address relevant results and conclusions in your manuscript.

Response: We thank the reviewer for pointing out this. In Figure R26 (Supplementary Fig. 15) (i), the name of the experimental groups can be confused. We meant the LALC and HALC as the groups with differentiated cells. To clarify the names, LALC and HALC are now changed to “LALC + differentiated adMSC” and “HALC + differentiated adMSC” (Figure R8). We apologize for the confusion. Since there was no significant change in stiffness before and after differentiation, we concluded that cell cultivation of adMSCs do not affect the physical properties of cultured meat. We also supplemented the description of these results in the revised manuscript by carefully reflecting the reviewer’s comments.

Figure R8 (Supplementary Fig. 15): Evaluation of the stiffness change according to adipogenic differentiation. Stiffness of each group was measured by frequency sweep at the angular frequency range from 0.1 to 1 (rad/s) ($n = 3$ independent experiments, two-tailed t-test). Error bars represent mean \pm s.d. Source data are provided a Source Data file.

In the revised manuscript (Lines 330 - 337) : Next, the changes in the stiffness and flavor upon cell differentiation were investigated. The change in the stiffness of the differentiated group was not significant compared to those of the bare and undifferentiated groups (Supplementary Fig. 15). Therefore, we concluded that cell cultivation and differentiation of adMSCs do not significantly affect the physical properties of cultured meat. However, the increment of the fatty flavor was higher in the LALC with differentiated adMSCs than in the HALC group (Fig. 4c and Supplementary Table 5). These results implies that the adipogenic differentiation does not affect the physical properties of the cultured meat construct but only the flavors.

In the revised manuscript (Lines 483 - 486): Although cell cultivation and differentiation of adMSCs did not significantly affect the stiffness of cultured meat, the results revealed that lipid droplet formation was significantly higher in the LALC scaffold, which has similar stiffness to adipose tissue.

Comment 9. Please check your description of the results and conclusions regarding potential ECM secretion: While a higher total protein content in differentiated myoblasts was shown, total protein content might be influenced both from cell maturation and (inner protein synthesis in adults muscle fibers) or from ECM secretion, or both. However, only the overall protein content was checked, while direct ECM secretion evaluation wasn't performed to unveil the exact reason. Only a general statement on higher protein content could be made, but not on higher ECM proteins.

Response: We appreciate the reviewer for this comment. As the reviewer pointed out, we stated that the ECM secretion is increased during the myogenic differentiation and affected the flavors of cultured meat in the discussion section of the manuscript. We included this statement according to the previous research reporting that the expression of proteins including ECM increases upon myogenic differentiation¹. However, we agree with the reviewer that this statement is not appropriate since we did not analyze the specific ECM secretion in our work. Therefore, we revised the sentence to not specify the ECM secretion. By reflecting the valuable comments from the reviewer, we could revise our discussion section once again and could reduce the overestimation and inappropriate assumptions.

In the revised manuscript (Lines 472 – 474): Because protein synthesis increased during the myogenic differentiation, we expected higher amounts of Maillard reaction products in the HALC than in the LALC.

Reference

1. Ojima, Koichi, et al. "Proteomic analysis of secreted proteins from skeletal muscle cells

during differentiation." *EuPA Open Proteomics* 5 (2014): 1-9.

REVIEWER COMMENTS

Reviewer #3 (Remarks to the Author):

Now that the abstract, introduction and discussion sections were revised, a new analysis of this paper is possible. In this second revision of this paper, the authors provided the previously demanded clarifications regarding the exact topic they wished to investigate, and what previous research was already conducted in that field. Additionally, the authors performed further analysis of the results, to provide a clarified version of the performed research.

As now understood by us, the paper does not show the development of novel fabrication processes, novel materials, novel cell differentiation induction or novel assessment techniques. It does, however, attempt to investigate the relationship between the maturation state of induced muscle and adipose progenitor cells on assembled 2D constructs, and several organoleptic assessments (available under laboratory-scaled research). This is an interesting aspect which should be investigated in developing cultured meat products. Importantly, however, high differentiation of adipose cells wasn't reached in this research, therefore extremely weakening related claims regarding the effects adipogenesis on the resultant construct characteristics. As this is one of the main concepts potentially bringing novelty to this work, additional experiments and stronger results are highly advised.

In conclusion, both the novelty and scientific claims made here are still questionable. Thus, this reviewer agrees with the 1st reviewer's recommendation to send this manuscript for further corrections, assessment, review and potential publication to a different, specific journal.

specific comments:

1. The authors should change the title of this paper, to convey that the topic relates to investigating the effect of cell maturation state on the resultant organoleptic attributes. The current one does not focus on this aspect – which is the main (perhaps only) novel concept in this work.
 2. The author should correct throughout the paper the aim of the study. It is not a novel fabrication technique, maturation induction process or new materials that was offered. It is a new understanding on how cell behavior relates to several different resultant organoleptic attributes.
 3. High maturation of cells into adipocytes wasn't observed in this work. Thus, all claims regarding the effect of adipogenesis on the resultant construct characteristics, should be carefully and thoroughly re-evaluated, by performing further research. This is relevant both to constructs with just adipogenesis-induced cells, and complex constructs after assembly with myogenic-induced ones. Overall, to strengthen this work, similar experiments with more evident adipogenesis are highly advised, to single out the extent of this process' effect on the resultant organoleptic characteristics.
- For example, the author have now showed that the expression of adipogenic differentiation-related proteins was increased both in LALC and HALC after differentiation induction (Figure R2), and that in Figure R1, the degree of differentiation varied depending on the scaffold stiffness, resulting in differences in 'lipid droplet' formation. However, no large lipid droplets were formed, but only a slightly elevated lipid synthesis was observed. Moreover, the author then stated that "Therefore, we concluded that cell cultivation and differentiation of adMSCs do not significantly affect the physical properties of cultured meat. However, the increment of the fatty flavor was higher in the LALC with differentiated adMSCs than in the HALC group (Fig. 4c and Supplementary Table 5). These results implies that the adipogenic differentiation does not affect the physical properties of the cultured meat construct but only the flavors."

Such strong assumptions are misleading, as the results did not provide a solid basis for them. At such early adipogenesis stage, without significant morphological changes in adipogenic-induced cells, one might not expect a significant difference in the resultant mechanical attributes with/without your used maturation medium.

In short, based on these results, one cannot deduce the effect of proper adipogenesis on the organoleptic properties of CM. As this is one of the only potential concepts bringing novelty to this

work, it should be investigated properly and be based on strong results.

4. The reviewed conclusions made by the authors regarding porosity from the performed SEM, are incorrect. This is because the freeze-drying process itself, to prepare samples for regular SEM (as you performed), creates pores. In the freezing process, ice crystals form, which are larger than the voids occupied by the previously liquid water. Then, sublimation occurs during lyophilization, resulting in these larger pores. Thus, the pores that were imaged probably resulted from this processes, and cannot be attributed to the difference in polymer concentration, or incorporated in the paper. one cannot state that the observed pores are the spaces the water occupied.

As described in this work, both the compression tests and all later cell seeding experiments in the whole paper were done with non-lyophilized hydrogels. without performing alternative, suitable porosity assessments – as previously suggested- one can't state any claim regarding porosity. One could only hypothesize that the differences in mechanical stiffness potentially rose from different polymer concentrations.

5. All used dyes and reagents used in the study, such as the LipidTOXTM Red Neutral Lipid for the confocal imaging, need to specifically stated in the manuscript .

Response letter

Journal: *Nature Communications*

Manuscript ID: NCOMMS-22-41801-T

Original Title: Block-Assembled Edible Scaffold for Cultured Meat

Revised Title: Cultured Meat with Enriched Organoleptic Properties by
Regulating Cell Differentiation

Authors: Milae Lee, Sohyeon Park, Bumgyu Choi, Woojin Choi, Hyun Lee,
Jeong Min Lee, Seung Tae Lee, Ki Hyun Yoo, Dongoh Han, Geul Bang, Heeyoun
Hwang, Won-Gun Koh, Sangmin Lee, Jinkee Hong

Response to Reviewer #3

Reviewer(s)'s Comments to Author:

Reviewer 3.

Comments

Now that the abstract, introduction and discussion sections were revised, a new analysis of this paper is possible. In this second revision of this paper, the authors provided the previously demanded clarifications regarding the exact topic they wished to investigate, and what previous research was already conducted in that field. Additionally, the authors performed further analysis of the results, to provide a clarified version of the performed research.

As now understood by us, the paper does not show the development of novel fabrication processes, novel materials, novel cell differentiation induction or novel assessment techniques. It does, however, attempt to investigate the relationship between the maturation state of induced muscle and adipose progenitor cells on assembled 2D constructs, and several organoleptic assessments (available under laboratory-scaled research). This is an interesting aspect which should be investigated in developing cultured meat products. Importantly, however, high differentiation of adipose cells wasn't reached in this research, therefore extremely weakening related claims regarding the effects adipogenesis on the resultant construct characteristics. As this is one of the main concepts potentially bringing novelty to this work, additional experiments and stronger results are highly advised.

In conclusion, both the novelty and scientific claims made here are still questionable. Thus, this reviewer agrees with the 1st reviewer's recommendation to send this manuscript for

further corrections, assessment, review and potential publication to a different, specific journal.

Response: We sincerely appreciate the reviewer for reviewing our manuscript and providing truthful comments about the manuscript. We also thank the reviewer for another chance to develop the quality of our manuscript. Comments from the reviewer in the first and second revisions allowed us to refine the logic of our study. Also, we could supplement the evident results to support our hypothesis though performing additional experiments followed by the reviewer's grateful comments. We sincerely appreciate the opportunity that the reviewer has given to us. In this third revision, we have written a more complete manuscript with supplemented results reflecting the reviewer's comments. We also performed additional experiments based on the reviewer's valuable comments to supplement the adipogenic differentiation data. We also changed the title and scheme of the manuscript to convey the main concept of our study. We appreciate the reviewer for pointing the shortcomings in our manuscript and for providing comments for improvement. From the reviewer's comments, we carefully revised our manuscript point-by-point as presented below. Changes made according to the reviewer's comments are highlighted in blue in the point-by-point responses.

Comment 1: The authors should change the title of this paper, to convey that the topic relates to investigating the effect of cell maturation state on the resultant organoleptic attributes. The current one does not focus on this aspect – which is the main (perhaps only) novel concept in this work.

Response: We appreciate the reviewer's comment. We agree with the reviewer that the current title does not convey the concept of this work. Therefore, we changed the title of the paper to **“Cultured Meat with Enriched Organoleptic Properties by Regulating Cell Differentiation”**. As the reviewer pointed out, block-assembly is not the novelty of this work because we do not address any new assembly method or new assemblable materials. Since we focus of investigating the effect of different degree of cell differentiation of myoblasts and adipogenic cells on various organoleptic properties, we changed the title to better reveal the keywords “cell differentiation” and “organoleptic properties of cultured meat”. We sincerely thank the reviewer for pointing out this important point.

Comment 2: The author should correct throughout the paper the aim of the study. It is not a novel fabrication technique, maturation induction process or new materials that was offered. It is a new understanding on how cell behavior relates to several different resultant organoleptic attributes.

Response: We thank the reviewer for this comment. We sincerely agree with the reviewer that the aim of this study is on the effect of cell differentiation on organoleptic properties of cultured meat. Therefore, we revised the introduction to focus on our main concept. Rather than starting with the explanation on difficulties of developing large size of cultured meat, we revised the sentences to describe that there is only a few studies that focus on mimicking the food-related characteristics of slaughtered meat. Then, we explained in the introduction that the goal of our study is developing cultured meat with beef-like organoleptic properties by regulating cell differentiation. Also in the introduction, we stated that the investigation on the effect of cell differentiation on the sensorial properties of cultured meat was conducted via scaffold engineering throughout the paper. We also changed the scheme of this work (Figure R1). Before reflecting the reviewer's valuable comment, the original scheme was focused on "block-assembly" of the cultured meat. However, as the reviewer pointed out, the main concept and new understanding of this work is investigating how cell differentiation affects organoleptic properties. Therefore, the scheme is now changed. Figure R1a describes that the organoleptic properties of slaughtered beef are determined by the biological characteristics of muscle and fat tissues. Figure R1b then shows that we regulate cell differentiation of muscle and fat cells by scaffold engineering to control the sensorial properties of cultured meat. Figure R1c describes that the 2D blocks with cells of which differentiation degree regulated in Figure R1b can be assembled together to form a bulk cultured meat that has beef-like organoleptic properties. The description of Fig. 1 in the introduction is also changed.

We also removed the descriptions such as "The marbled structure of meat, in which muscle

and fat tissues are blended, can be mimicked by implementing this strategy.”, “The most significant advantage of the ACM is the ability to control its size and shape using a simple assembly process.”, “Thus, we confirmed that different muscle to fat ratios can be implemented for mimicking each meat part through a simple assembly of the meat blocks.”, and “Furthermore, we produced the cultured meat with different sizes and shapes, confirming the potential of our strategy to mimic the structural and sensory characteristics of various cuts of slaughtered meat.” in the result section. Rather, we inserted a sentence like “After optimizing the muscle and fat blocks, we assembled the blocks to produce a small-sized cultured beef **that embodies the beef-like sensorial properties derived from the differentiated cells.**” in the result section (Lines 349 – 351) to explain that the purpose of assembly is not on mimicking the marbled structure of meat but on mimicking the sensorial properties of meat. Also, we revised the description in Lines 413 – 417 to explain that the fabrication of bulk ACM is not for showing the novelty of the assembly method, but for showing the potential of our strategy to mimic both structural and sensorial characteristics of various cuts of slaughtered beef by assembling our 2D meat blocks.

In the revised manuscript (Lines 39 - 44): However, research on cultured meat is still in its infancy because of the challenges on mimicking the biological and physical properties of slaughtered meat in vitro. Furthermore, cultured meat research on mimicking the food-related characteristics such as sensorial properties and nutritional value of meat is even fewer. To overcome these issues, cells can be supported with scaffolds, which provide them with biological or physical environment of natural tissues of animals.

In the revised manuscript (Lines 82 - 86): In this study, we suggest a strategy of developing cultured meat that embodies the organoleptic properties of conventional beef by only regulating

cellular differentiation. To investigate the effect of cellular differentiation on the sensorial characteristics of cultured meat, we firstly fabricate 2D hydrogel scaffolds composed of fish gelatin and alginate, and the stiffness of the hydrogels is controlled by varying the content of the crosslinked alginate.

In the revised manuscript (Lines 49 - 52): For example, the biological characteristics of muscle and fat tissues, such as myofiber dimensions and lipid content, determine the organoleptic properties of slaughtered meat, and these tissue characteristics can be affected by the differentiation of cells (Fig. 1a).

In the revised manuscript (Lines 88 - 89): We verify the variation in the sensory properties with a change in the cellular differentiation degree (Fig. 1b).

In the revised manuscript (Lines 89 - 91): Using this approach, we develop 2D muscle blocks and fat blocks. Lastly, the two meat block types are assembled to produce a cultured meat with beef-like organoleptic characteristics (Fig. 1c).

In the revised manuscript (Lines 349 – 351): After optimizing the muscle and fat blocks, we assembled the blocks to produce a small-sized cultured beef that embodies the beef-like sensorial properties derived from the differentiated cells.

In the revised manuscript (Lines 413 – 417): Thus, we confirmed that the cultured meat that embodies the food-related properties of conventional beef can be fabricated with our 2D meat blocks. We also produced the bulk assembled cultured meat mimicking the different cuts of slaughtered beef, confirming the potential of our strategy to mimic both structural and sensorial

characteristics of various cuts of slaughtered meat.

Figure R1 (Fig. 1): Schematic illustration of the proposed strategy. a Muscle and fat tissues consisting slaughtered beef determine the organoleptic properties of beef. **b** Scaffold engineering for controlling myogenesis and adipogenesis affects the nutritional and sensorial properties of cultured meat. **c** Assembled cultured meat composed of the 2-dimensional meat blocks with highly differentiated cells owns beef-like organoleptic properties.

Comment 3: High maturation of cells into adipocytes wasn't observed in this work. Thus, all claims regarding the effect of adipogenesis on the resultant construct characteristics, should be carefully and thoroughly re-evaluated, by performing further research. This is relevant both to constructs with just adipogenesis-induced cells, and complex constructs after assembly with myogenic-induced ones. Overall, to strengthen this work, similar experiments with more evident adipogenesis are highly advised, to single out the extent of this process' effect on the resultant organoleptic characteristics.

For example, the author have now showed that the expression of adipogenic differentiation-related proteins was increased both in LALC and HALC after differentiation induction (Figure R2), and that in Figure R1, the degree of differentiation varied depending on the scaffold stiffness, resulting in differences in 'lipid droplet' formation. However, no large lipid droplets were formed, but only a slightly elevated lipid synthesis was observed. Moreover, the author then stated that "Therefore, we concluded that cell cultivation and differentiation of adMSCs do not significantly affect the physical properties of cultured meat. However, the increment of the fatty flavor was higher in the LALC with differentiated adMSCs than in the HALC group (Fig. 4c and Supplementary Table 5). These results implies that the adipogenic differentiation does not affect the physical properties of the cultured meat construct but only the flavors."

Such strong assumptions are misleading, as the results did not provide a solid basis for them. At such early adipogenesis stage, without significant morphological changes in adipogenic-induced cells, one might not expect a significant difference in the resultant mechanical attributes with/out your used maturation medium.

In short, based on these results, one cannot deduce the effect of proper adipogenesis on the organoleptic properties of CM. As this is one of the only potential concepts bringing novelty to this work, it should be investigated properly and be based on strong results.

Response: We thank the reviewer for this comment. We acknowledge the reviewer's concerns. To supplement the adipogenic differentiation data, we performed additional experiments on adMSC differentiation. We replaced the lipid-stained confocal images to the one with higher magnification to show the lipid droplets more precisely (Figure R2). Also, we removed the lipid droplet intensity data. Instead, we added the results analyzing the lipid droplet coverage percent and the number of lipid droplet per cell based on the confocal images using Image J software (Figure R3). For the lipid droplet coverage data, we calculated the area percentage of the LipidTOXTM stained lipid droplets in the total image area. The number of lipid droplet per cell was calculated by dividing the number of the LipidTOXTM stained lipid droplets by the number of DAPI. Compared to HALC, we confirmed that both the lipid droplet content per cell and the lipid droplet coverage (%) were significantly higher in LALC. The description of these results is also added in the revised manuscript.

Figure R2 (Fig. 4b(i)). Lipid droplets and nuclei of differentiated adMSCs were stained with LipidTOXTM (red) and DAPI (blue).

Figure R3 (Fig. 4b(ii)). Quantitative assessments of the lipid droplet formation are shown in the graph based on the confocal images using Image J software ($n = 3$ independent experiments, two-tailed t-test). Error bars represent mean \pm s.d. Source data are provided as a Source Data file.

In the revised manuscript (Lines 304 - 305): The lipid droplet coverage and number of lipid droplet per cell were significantly higher in LALC.

In the revised manuscript (Lines 620 - 622): For lipid droplet coverage data, the area percentage of the LipidTOX™ stained lipids in the total image area was calculated. The number of lipid droplet per cell was calculated by dividing the number of the LipidTOX™ stained lipid droplets by the number of DAPI.

We also performed specific analysis on the protein expression results from our LC-MS/MS proteomics data to compare the expression of the adipogenesis-related proteins of the adMSCs depending on the scaffold (Figure R4). In LALC, we could confirm that various adipogenesis-related proteins such as fat storage inducing transmembrane protein 2 (FITM2) and adiponectin

(ADIPOQ) were expressed significantly higher than that of HALC. Specifically, ADIPOQ is the representative marker for adipocyte differentiation[1]. In LALC, the expression of ADIPOQ of the differentiated adMSC was approximately 1.5 folds higher than that of HALC sample. This suggests that the degree of differentiation of adMSC into adipocytes was higher in LALC. Glycerol-3-phosphate dehydrogenase 2 (GPD2), CCAAT enhancer binding protein beta (CEBPB), glutamate-ammonia ligase (GLUL), acyl-CoA dehydrogenase medium chain (ACADM), LETM-domaining containing 1 (LETMD1), aldehyde dehydrogenase 6 family member A1 (ALDH6A1), and acyl-CoA dehydrogenase short/branched chain (ACADSB) are also known to be up-regulated during the adipogenic differentiation[2], [3], [4]. Expression of these proteins were also significantly higher in the LALC sample compared to HALC, confirming the higher adipogenic differentiation degree in the scaffold with softer stiffness, LALC.

Also, we analyzed the expression of the identical proteins in the undifferentiated groups to verify that this difference in protein expression levels are due to different adipogenic differentiation (Figure R5). In the results, we could confirm that the expression of the adipogenesis-related proteins were up-regulated in differentiated cells both in HALC and LALC compared to the undifferentiated groups, indicating that differentiation was induced in both scaffolds. Combining the two results in Figure R4 and Figure R5, we verified that the adipogenesis-related protein expressions were regulated depending on the scaffold stiffness. We added the description on Figure R4 in the revised manuscript.

In the revised manuscript (Lines 314 - 318): Furthermore, the expression of the adipogenesis-related proteins was confirmed to be higher in the LALC compared to that of HALC (Supplementary Fig. 14). Specifically, the representative marker for adipocyte differentiation, ADIPOQ, was expressed significantly higher in LALC. These results indicate the significantly

different degree of adipogenesis depending on the scaffolds.

Figure R4 (Supplementary Fig. 14): Proteins related to adipogenesis which are upregulated in the differentiated adMSCs in LALC. Expression of the adipogenesis-related proteins in the differentiated adMSCs in LALC were normalized to that of HALC ($n = 3$ independent experiments, two-tailed t-test).

Figure R5: Expressions of the adipogenesis-related proteins in the differentiated adMSC samples (HALC diff. and LALC diff.) were normalized to that of undifferentiated adMSC samples (HALC undiff. and LALC undiff.) ($n = 3$ independent experiments, two-tailed t-test).

As we supplemented data for lipid droplet formation in Figure R3, it can be seen that the lipid droplet formation was significantly higher in LALC compared to that of HALC. We acknowledge the reviewer's point that the size and number of lipid droplets are small compared to the lipid droplets of the previous papers reporting adipocyte differentiation. We also remember that the reviewer had kindly provided us a previous study which addresses successful

LipidTOXTM staining for adipocyte maturation during the first revision[5]. Many previous research show large size of lipid droplets when evaluating adipogenic differentiation of the cells[6], [7]. However, these studies mostly use human adipose stem cells or murine-derived cells of which adipogenic differentiation protocol has already been established from many previous papers. In this study, we use primary bovine adipose-derived mesenchymal stem cells of which differentiation protocol has rarely been reported. In the most recent paper on developing cultured fat from ruminant species-derived adipogenic cells, they explain the lack of research on adipogenic cells from agricultural species[8]. This difficulty is because of the different metabolism between the ruminant species and human or rodent species. Unlike humans or murine, ruminants such as cattle are known to be less responsive to the insulin pathway, thus the differentiation protocol other than the insulin-based adipogenesis induction method is required. However, since a precise adipogenesis induction method not using insulin-based medium is not defined yet, many research on bovine adipogenesis still use insulin-based method. The composition of the differentiation medium and the degree of adipogenic differentiation vary in papers addressing the adipogenesis using bovine cells. Due to these limitations, there are only few reports of achieving adipogenic differentiation of human or mouse cells using bovine-derived cells. Even recent papers on cultured meat addressing adipogenic differentiation do not reach such mature stage of adipogenesis[9],[10]. In our manuscript, the lipid droplet size is also small compared to conventional research on adipogenesis. However, we confirmed the significant difference in the lipid droplet formation and expression of the adipogenesis-related proteins depending on the scaffold. Then, we assessed the difference in the amount of fatty flavor compounds depending on this different degree of adipogenic differentiation. These compounds are known as volatile compounds resulting from lipid oxidation in meat fat[11],[12],[13]. As the reviewer pointed out, we show a small amount of lipid droplets, but nevertheless, it was confirmed that there was a significant

difference in fatty flavor as the formation of lipid droplets was differed. From these results, we want to suggest that if a differentiation protocol that can maximize the adipogenic differentiation of bovine adipogenic cells is developed in the future, cultured meat that is even more enriched in fatty flavors can be obtained through the strategy in this study.

We also appreciate the reviewer's comment on our description explaining the non-significant effect of adipogenesis on mechanical properties of cultured meat. From the reviewer's valuable comments, we realized that the description can be misleading since the adipogenic differentiation is not at the mature stage. We deeply agree with the reviewer's comments that the mechanical property of cultured meat is not addressed after the maturation of adipocytes thus cannot be concluded in this study. Therefore, we supplemented the description of this part in the main context as well as in the discussion section to explain that the effect of adipogenesis on the stiffness of cultured meat should be verified in further studies since the adipogenic differentiation is not yet at the mature stage in our work. We are grateful to the reviewer for the comments so that we could revise the incorrect assumptions.

In the revised manuscript (324 – 328): The change in the stiffness of the differentiated group was not significant compared to those of the bare and undifferentiated groups (Supplementary Fig. 16). However, the effect of adipogenesis on the physical properties of cultured meat should be investigated in further studies since adipocyte maturation was not evaluated at this stage.

In the revised manuscript (Lines 454 - 459): It was confirmed that the degree of adipogenesis was regulated depending on the stiffness of scaffold. However, the maturity of adipocyte was not specifically evaluated in this study. Therefore, the results showing non-significant effect of adipogenic differentiation on the stiffness of cultured meat should be further verified in future studies with fully matured adipocytes. Although stiffness was not dependent on the

differentiation of adMSCs at such early differentiation stage, the formation of fatty flavor molecules was affected.

References

- [1] Ando, Yusuke, et al. "Placental extract suppresses differentiation of 3T3-L1 preadipocytes to mature adipocytes via accelerated activation of p38 MAPK during the early phase of adipogenesis." *Nutrition & metabolism* 16.1 (2019): 1-13.
- [2] Choi, Sunkyu, Neha Goswami, and Frank Schmidt. "Comparative proteomic profiling of 3T3-L1 adipocyte differentiation using SILAC quantification." *Journal of Proteome Research* 19.12 (2020): 4884-4900.
- [3] Tang, Qi-Qun, Tamara C. Otto, and M. Daniel Lane. "CCAAT/enhancer-binding protein β is required for mitotic clonal expansion during adipogenesis." *Proceedings of the National Academy of Sciences* 100.3 (2003): 850-855.
- [4] Swierczynski, Julian, et al. "Enhanced glycerol 3-phosphate dehydrogenase activity in adipose tissue of obese humans." *Molecular and cellular biochemistry* 254 (2003): 55-59.
- [5] Mandl, Markus, et al. "An organoid model derived from human adipose stem/progenitor cells to study adipose tissue physiology." *Adipocyte* 11.1 (2022): 164-174.
- [6] Batrakou, Dzmitry G., et al. "TMEM120A and B: nuclear envelope transmembrane proteins important for adipocyte differentiation." *PloS one* 10.5 (2015): e0127712.
- [7] Sanjabi, Bahram, et al. "Lipid droplets hypertrophy: a crucial determining factor in insulin regulation by adipocytes." *Scientific reports* 5.1 (2015): 8816.
- [8] Louis, Fiona, et al. "Mimicking Wagyu beef fat in cultured meat: Progress in edible bovine adipose tissue production with controllable fatty acid composition." *Materials Today Bio* 21 (2023): 100720.
- [9] Jeong, Dayi, et al. "Efficient Myogenic/Adipogenic Transdifferentiation of Bovine

Fibroblasts in a 3D Bioprinting System for Steak-Type Cultured Meat Production." *Advanced Science* 9.31 (2022): 2202877.

[10] Kang, Dong-Hee, et al. "Engineered whole cut meat-like tissue by the assembly of cell fibers using tendon-gel integrated bioprinting." *Nature Communications* 12.1 (2021): 5059.

[11] Watanabe, Akira, et al. "Analysis of volatile compounds in beef fat by dynamic-headspace solid-phase microextraction combined with gas chromatography–mass spectrometry." *Journal of Food Science* 73.5 (2008): C420-C425.

[12] Bi, Jicai, et al. "Effect of different cooking times on the fat flavor compounds of pork belly." *Journal of Food Biochemistry* 46.8 (2022): e14184.

[13] Bleicher, Julian, Elmar E. Ebner, and Kathrine H. Bak. "Formation and analysis of volatile and odor compounds in meat—a review." *Molecules* 27.19 (2022): 6703.

Comment 4: The reviewed conclusions made by the authors regarding porosity from the performed SEM, are incorrect. This is because the freeze-drying process itself, to prepare samples for regular SEM (as you performed), creates pores. In the freezing process, ice crystals form, which are larger than the voids occupied by the previously liquid water. Then, sublimation occurs during lyophilization, resulting in these larger pores. Thus, the pores that were imaged probably resulted from this processes, and cannot be attributed to the difference in polymer concentration, or incorporated in the paper. one cannot state that the observed pores are the spaces the water occupied.

As described in this work, both the compression tests and all later cell seeding experiments in the whole paper were done with non-lyophilized hydrogels. without performing alternative, suitable porosity assessments – as previously suggested- one can't state any claim regarding porosity. One could only hypothesize that the differences in mechanical stiffness potentially rose from different polymer concentrations.

Response: We thank the reviewer for this comment. We understand the reviewer's concerns about the incorrect statement relating the porosity and mechanical properties of the hydrogels. We acknowledge that our porosity data are insufficient to support such relationship in our study. We agree with the reviewer's valuable comment that we could only relate the polymer concentration with the mechanical properties of the hydrogels. To remove the exaggerations and leaps in the manuscript, we excluded the SEM data and the explanation relating the porosity with the stiffness of the hydrogels. Thus, the description in Fig. 2d was removed and Fig. 2 is now changed (Figure R6). We sincerely thank the reviewer again for pointing out this so that we could revise the misinterpretation in the manuscript.

Figure R6 (Fig. 2): Characterization of the gelatin/alginate scaffold. **a** Illustration of the change in the alginate concentration and crosslinking degree of the alginate network of the scaffolds. **b** (i) Raman spectra of the LALC, LAHC, HALC, and HAHC, (ii) Percentage of the calcium-crosslinked alginate structures calculated from the Raman shifted peak area between 1390 and 1420 cm^{-1} ($n = 3$ independent experiments, two-tailed t-test). **c** (i) Stress–strain curves obtained during the compression test of the scaffolds, (ii) Young’s modulus of each scaffold at a compressive strain of 0–50 % ($n = 3$ independent experiments, two-tailed t-test). Error bars represent mean \pm s.d. Source data are provided as a Source Data file.

Comment 5: All used dyes and reagents used in the study, such as the LipidTOX™ Red Neutral Lipid for the confocal imaging, need to specifically stated in the manuscript.

Response: We thank the reviewer for pointing out this. We added the description for indicating the MHC dye and lipid dye in the confocal images in Fig. 3a and Fig. 4b (Lines 171 – 173 and Lines 280 – 282 in the revised manuscript). In addition, we clarified the dyes used in the myosin heavy chain, lipid, and nuclei staining in the method section by carefully reflecting the reviewer's comments.

In the revised manuscript (Lines 171 - 173): a Myosin heavy chains (MHC) and nuclei were immunostained with MF20 and DAPI on differentiation day 5 and day 8. The nuclei stained with DAPI (blue), and the MHCs stained with MF20 (red).

In the revised manuscript (Lines 280 - 282): b (i) Confocal images showing the lipid droplet formation analyzed by immunostaining using HCS LipidTOX™ Red Neutral Lipid Stain (red) and DAPI (blue).

In the method section (Lines 604 - 615): Subsequently, the samples were treated overnight to inhibit any non-specific antibody binding using a blocking solution containing 2% (w/v) bovine serum albumin (BSA, Sigma Aldrich), 0.3% (v/v) triton X-100 (Triton™ X-100 solution, Sigma Aldrich), and 10% (v/v) horse serum in 1X PBS. The MHC antibody (MF 20, DSHB) was diluted 100 folds in the diluent solvent composed of 10% (v/v) horse serum and 2% (w/v) BSA and was used to treat the samples for 2 h at room temperature. The samples were washed twice with 1X PBS and once with 0.025% (v/v) triton X-100. Thereafter, the samples were treated with the secondary antibody (Donkey anti-mouse Alexa flour 594, Thermo Fischer), which was diluted to 400 folds in the same diluent solvent of MF 20, for 1 h at room

temperature. After the washing procedure, the sample was treated with DAPI (Sigma Aldrich), diluted to 250 folds in 1% (w/v) BSA solution, for 30 min to stain the cell nuclei in the scaffolds. The stained cells were observed using a confocal laser scanning microscope (LSM 980, Carl Zeiss).

In the method section (Lines 616 - 619): For the adipogenic differentiation analysis, lipid staining was performed using the Oil Red O stain kit (abcam) and the associated staining protocol. The fluorescent staining of the lipid droplets was performed using the 200-folds diluted LipidTOX™ (HCS LipidTOX™ Red Neutral Lipid Stain, Invitrogen).

REVIEWERS' COMMENTS

Reviewer #2 (Remarks to the Author):

I am satisfied with the author's responses.